# Evaporation from a large lowland reservoir – observed dynamics and drivers during a warm summer

Femke A. Jansen[1], Remko Uijlenhoet[2], Cor M.J. Jacobs[3], and Adriaan J. Teuling[1]

[1]Hydrology and Quantitative Water Management Group, Wageningen University, Wageningen, The Netherlands
[2]Department of Water Management, Delft University of Technology, Delft, The Netherlands
[3]Wageningen Environmental Research, Wageningen University and Research, Wageningen, The Netherlands

**Correspondence:** Femke Jansen (femke.jansen@wur.nl)

**Abstract.** We study the controls on open water evaporation of a large lowland reservoir in the Netherlands. To this end, we analyse the dynamics of open water evaporation at two locations, i.e. Stavoren and Trintelhaven, at the border of Lake IJssel (1,100 km$^2$) where eddy covariance systems were installed during the summer seasons of 2019 and 2020. These measurements were used to develop data-driven models for both locations. Such a statistical model is a clean and simple approach that can provide a direct indication and insight of the most relevant input parameters involved in explaining the variance of open water evaporation, without making a prior assumptions on the process itself. This way, we find that a combination of wind speed and the vertical vapour pressure gradient can explain most of the variability of observed hourly open water evaporation. This is in agreement with Dalton's model which is a well-established model often used in oceanographic studies for calculating open water evaporation.

Validation of the data-driven models demonstrates that a simple model using only two variables yields satisfactory results at Stavoren, with R$^2$ values of 0.84 and 0.78 for hourly and daily data, respectively. However, the validation results for location Trintelhaven fall short (R$^2$ values of 0.67 and 0.65 for hourly and daily data, respectively). Validation of the simple models that are using only routinely measured meteorological variables perform adequately at hourly timescale (R$^2$ = 0.78 at Stavoren, and R$^2$ = 0.51 at Trintelhaven) and at daily timescale (R$^2$ = 0.82 at Stavoren, and R$^2$ = 0.87 at Trintelhaven) timescales. These results for the summer periods show that global radiation is not directly coupled to open water evaporation at the hourly or daily timescale, but it rather is a combination of wind speed and vertical gradient of vapour pressure. We would like to stress the importance of including the correct drivers of open water evaporation in the parametrization in hydrological models to adequately represent the role of evaporation in the surface-atmosphere coupling of inland water bodies.

## 1 Introduction

Inland water bodies are known to interact with the local, regional and even global climate and are therefore highly sensitive to climate change (Adrian et al., 2009; Liu et al., 2009; Le Moigne et al., 2016; Wang et al., 2018; Woolway et al., 2020). Evaporation is a sink in the water balance of inland water bodies and therefore it becomes most critical to understand how open water evaporation ($E_{\text{water}}$) will respond to these changing conditions. It is expected that changes in longwave radiation, Bowen ratio, ice cover, and stratification will affect the dynamics of $E_{\text{water}}$ at the long-term (Wang et al., 2018; Woolway et al.,

2020). Whereas at the shorter decadal timescale, a contribution to trends and variations in $E_{\text{water}}$ is expected as a result from changes in wind speed, humidity, and through global and regional solar dimming and brightening and its effect on water surface temperature (Desai et al., 2009; McVicar et al., 2012; Schmid and Köster, 2016; Wang et al., 2018; Woolway et al., 2020). During the summer season evaporation rates are highest and, depending on the functions of the water body, the water demand is largest for other purposes such as drinking water extraction and agricultural irrigation practices. Summer seasons are projected to become warmer in the Netherlands, with more severe and prolonged periods of drought (Seneviratne et al., 2006, 2012; KNMI, 2015; Teuling, 2018; Christidis and Stott, 2021). Only if we are able to correctly parameterize $E_{\text{water}}$, implying that the employed model is right for the right reasons, it is possible to make well supported short-term predictions and long-term projections of $E_{\text{water}}$ during these critical summer periods. These predictions and projections could assist water managers to make appropriate decisions to guarantee ample access to freshwater.

In terms of thermodynamics, a shallow inland water body of only a few meters deep can be considered as a system that can be placed somewhere in between an ocean system, or another deep water body, and an infinite shallow water surface that behaves almost similar to a land surface. An important difference between these two systems at both ends of the spectrum is the location where heat is stored (Brutsaert, 1982; Kleidon and Renner, 2017). In the case of water bodies heat storage takes place below the atmosphere-water interface and is generally mixed away from the surface. This is different for a land surface, where heat is stored in the lower atmosphere, vegetation and the upper soil layers. This leads to larger temperature amplitudes in sunny conditions, with strongly increasing surface temperatures and warming of the lower atmosphere during daytime, and strong decreases during nighttime. This difference is rooted in the distinct surface properties and heat capacity of a water body and a land surface, which is leading to different dynamics of turbulent exchange with the atmosphere and is reflected at both the seasonal and daily cycle of latent heat flux (Brutsaert, 1982). In contrast to a land surface, solar radiation is able to penetrate through the water surface, thereby delivering and storing its energy down to deeper water layers, depending on the light absorption characteristics of the water. There, subsurface redistribution of energy can take place through turbulent mixing and non-turbulent flow of the water, and the energy can be released back into the atmosphere through sensible and latent heat fluxes. The subsurface energy budget implies that lake depth controls the dynamical range of lake temperature amplitudes on a diurnal timescale. Thus, instead of focusing at the surface only, rather the whole volume of the system should be considered. It is essential to understand how differences in properties of a system result into distinct drivers of evaporation and to include and to represent those in the parameterization of evaporation in hydrological models.

The frequently used method of Penman (1948) is widely recognized as the standard for calculating both terrestrial evaporation and $E_{\text{water}}$ for shallow water surfaces for which the model was originally developed. Penman (1948) based his model on the historical model originally developed by Dalton in 1802. The latter model, and variations of it in the form of bulk transfer models, has been adopted and reviewed by many oceanographic studies and was found to perform well in estimating $E_{\text{water}}$ from oceans (Brutsaert, 1982; Josey et al., 2013; Pinker et al., 2014; Bentamy et al., 2017; Cronin et al., 2019). Dalton (1802) recognized the importance of using the difference of vapour pressure at the water-air interface, where the exchange of water takes place, to model $E_{\text{water}}$. This difference is subject to change when energy entered the water body, is stored, and released again, thereby changing the temperature and thus the vapour pressure at the water surface. Dalton (1802) proposed

that $E_{\text{water}}$ can best be described by the product of a wind function, acting as transport mechanism, and the difference between the saturation vapour pressure at the water surface and the vapour pressure at 2 metres above the water surface. Penman (1948) eliminated the surface temperature, which is often difficult to determine, by assuming that it could be replaced by temperature and vapour pressure at reference height through linearisation of the vapour pressure curve (Brutsaert, 1982). This assumption results in the essential difference between the models of Dalton and Penman, where Dalton uses the vertical difference of vapour pressure, while Penman uses the vapour pressure deficit at 2 metres height (Brutsaert, 1982). Omitting the water heat flux (G) for infinitely shallow water surfaces reduces Penman's model to a combination of a radiation term that is driven by net radiation and an aerodynamic term.

Most studies in the past have been dedicated to measuring terrestrial evaporation to understand its driving variables. However, comparably significant less studies performed measurements of $E_{\text{water}}$ from inland water bodies. This can partly be attributed to practical difficulties when measuring above or close to a water body. There are numerous methods available to measure $E_{\text{water}}$ either through indirect estimations (e.g. water balance method, energy budget approach, bulk transfer method, complementary approaches) or through more direct measurements (e.g. scintillometry, eddy covariance technique, evaporation pan method) (Finch and Calver, 2008; Abtew and Melesse, 2013). Historically, evaporation pans have been widely used because of their relatively simple use and moderate data and installation requirements. However, depending on the installation method of the pan drawbacks that might be encountered are: adverse effects of heat exchange through the side walls, incomparable heat storage properties of the pan and a lake, limited temporal resolution, and splashing in or out of water caused by wind or rain (Allen et al., 1998; Sumner and Jacobs, 2005; Masoner and Stannard, 2010). Scintillometry, a technique that was developed more recently, enables us to quantify $E_{\text{water}}$ integrated over larger surfaces. Scintillometers therefore offer the possibility to account for spatial variability and comparisons with data obtained from satellite images (McJannet et al., 2011). However, scintillometers only indirectly measure the turbulent fluxes through the use of the Monin-Obukhov Similarity Theory (MOST) of which the assumptions do not always hold (Beyrich et al., 2012). In general, the eddy covariance technique is considered to be the most accurate method to quantify $E_{\text{water}}$ (Lenters et al., 2005b). In contrast to scintillometry, eddy covariance is based on a point measurement with a smaller footprint at the hectare to square kilometer scale, depending on the meteorological conditions and the height of the sensor. It measures the vertical moisture flux through the covariance of the vertical wind speed and the concentration of water vapour. This concept renders it the most direct flux measurement technique available and it provides continuous observations suitable for studying the evaporation process.

In the past, a number of studies reported measurements of $E_{\text{water}}$ from which modelling concepts to estimate $E_{\text{water}}$ were developed. Some of these concepts are based on different drivers of evaporation and they disagree on the meteorological variables to be included. Some studies have for instance found that global radiation is not a direct driver of $E_{\text{water}}$ at shorter timescales and should therefore not be included in the parameterization (Venäläinen et al., 1999; Blanken et al., 2011; Kleidon and Renner, 2017). Rather, the product of vapour pressure deficit (VPD) and wind speed should be used as argued by Blanken et al. (2000) and Granger and Hedstrom (2011). At larger timescales a spatial coupling was found between $E_{\text{water}}$ and precipitation minus $E_{\text{terrestrial}}$ (Zhou et al., 2021). Jansen and Teuling (2020) studied the (dis)agreement among a number of concepts that are commonly used. They found that the models of Penman (1948), Makkink (1957), De Bruin and Keijman (1979), Granger and

Hedstrom (2011), Hargreaves (1975), and Mironov (2008) result in different representations of especially the diurnal cycle of evaporation. Additionally, at the yearly timescale the methods disagree on the average increasing historical trend of the evaporation rate, as well as for the projected future trends. At longer timescales (i.e. seasonal and yearly timescales) it is important to include the interdependency between lake temperature and evaporation (Woolway et al., 2018; Wang et al., 2018). This requires a concept in which the water body energy balance is to be represented adequately, for the correct modelling of the $E_{water}$ process.

In the Netherlands measurements of $E_{water}$ have been under-represented as well, although measured by their extent ($\sim$17% of the total area (Huisman, 1998)), inland water bodies form a crucial element in its water management system (Buitelaar et al., 2015). Adequate estimations of $E_{water}$ are important in this context because there is a strong coupling between $E_{water}$ and for instance lake level and extent, the lake ecosystem, and lake stratification and mixing regimes (Woolway et al., 2020; Jenny et al., 2020). Lake IJssel is the largest freshwater reservoir in the Netherlands fulfilling crucial hydrological functions, both in flood prevention and freshwater supply for agricultural irrigation and drinking water extraction. The water level of the lake is managed to have a distinct summer and winter level. This flexibility provides the opportunity to raise the water level before the start of the summer with typically higher evaporation rates. In this way, a buffer can be created to ensure that its functions can be fulfilled continuously throughout the summer season. Currently, the Dutch operational hydrological models use Makkink's equation (Makkink, 1957) to quantify $E_{water}$ for Lake IJssel. Makkink is a radiation-based model which finds its origin in Penman's equation through the Priestley-Taylor equation (explained in Sect.2.5) and has been developed to estimate evapotranspiration over well-watered grasslands at daily timescale. Although a correction factor is applied to account for the difference between terrestrial evaporation and $E_{water}$, Makkink's equation is not able to capture the dynamics of $E_{water}$ compared to what is found by aforementioned observational studies on $E_{water}$ and compared to estimations with physically-based lake models such as FLake (Jansen and Teuling, 2020). This calls for improving and implementing our understanding of the driving process of $E_{water}$ by building on previous studies about $E_{water}$ of Lake IJssel (Keijman and Koopmans, 1973; De Bruin and Keijman, 1979; Abdelrady et al., 2016). The goal of our study is therefore to analyse the dynamics of $E_{water}$ of Lake IJssel supported by a data-driven analysis with the aim to parameterize $E_{water}$ based on its main drivers. To this end, we performed a long-term measurement campaign focusing on two summer periods (2019 – 2020) at two locations using the eddy covariance technique to measure $E_{water}$, along with observations of related meteorological variables over Lake IJssel in the Netherlands.

## 2 Data, Materials and Methods

### 2.1 Study area

In this research study the latent heat flux of Lake IJssel was analysed. Lake IJssel, also referred to as IJsselmeer in Dutch, is the largest freshwater lake in the Netherlands, bordering the provinces of Flevoland, Friesland, and Noord-Holland (see Fig. 1a). The lake covers an area of 1,100 km$^2$ and is enclosed by the Afsluitdijk embankment in the north and by the Houtribdijk embankment in the south-west. With an average depth of 5.5 m and a maximum depth of 7 m, the lake can be considered a large shallow lake. The river IJssel is the main vein that supplies the lake with freshwater. Together with the inflow from

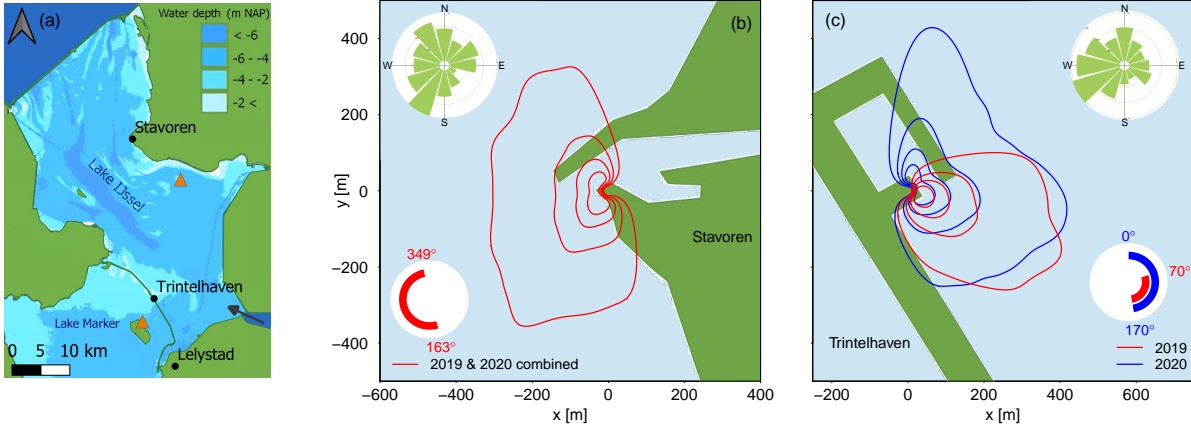

**Figure 1.** Map of the study region and location of the measurement sites. The black dots in panel (**a**) represent the locations of the turbulent flux observations (Stavoren and Trintelhaven) and the locations where supplementary meteorological data was gathered by the Royal Netherlands Meteorological Institute (Stavoren and Lelystad). The orange triangles are the locations of water temperature measurements performed by Rijkswaterstaat. Black arrow indicates the location where water from the IJssel river enters Lake IJssel. Panels (**b**) and (**c**) illustrate the sampling area that is measured by the flux tower for onshore wind conditions by contour lines (20, 40, 60, and 80% from inside to outside). This was based on a flux footprint analysis using the model of Kljun et al. (2015). The circular inset on the bottom side of both panels indicates the wind directions that are included in the analysis of open water evaporation. At location Trintelhaven (**c**) this angle has shifted from the year 2019 to 2020 following the change of direction of the eddy-covariance instrument (see text). The circular inset at the top of both panels represents the average wind conditions as illustrated by a windrose.

the neighbouring polder systems the lake receives on average 340 m$^3$s$^{-1}$. Its main outflow occurs under gravity at the sluices of the Afsluitdijk where water is discharged to the Waddenzee. During summertime a flexible water level is used, which can vary between $-0.10$ m NAP (Normaal Amsterdams Peil, the local sea level reference) and $-0.30$ m NAP. During wintertime the lake level should at least be kept at $-0.40$ m NAP. Lake IJssel fulfills an important hydrological role in the low-lying

5 Netherlands, both in flood mitigation and in freshwater supply for agricultural and drinking water purposes. The flexible lake level management during the year provides the water managers with a tool to respond to the meteorological conditions and the need for fresh water.

## 2.2 Site description, instrumentation and data

An eddy covariance (EC) measurement system was mounted in a telecommunication tower that is located at the shoreline

10 in the city of Stavoren at the north-east coast of the lake (see Fig. 1a). Its favourable position in relation to the predominant south-westerly wind direction in combination with an already existing telecommunication tower renders this location suitable for the measurements needed to analyse the dynamics of open water evaporation. An additional benefit of this location is that it allows a comparison of the dynamics of terrestrial evaporation and open water evaporation by selecting time intervals based

on the footprint of the flux tower. An open path-integrated gas analyser and sonic anemometer (IRGASON) from Campbell Scientific was installed at a height of 7.5 m above the land surface and was pointed towards 220°. The IRGASON measures the water vapour and CO2 concentration, air temperature by the sonic anemometer, barometric pressure, and the three wind components at a sampling frequency of 20 Hz. In addition, air temperature and relative humidity were both measured at 5.9 m
and 7.4 m height using HMP155A sensors (Campbell Scientific).

In the harbour Trintelhaven located in the middle of the Houtribdijk embankment another telecommunication tower was equipped with the same EC system, installed at a height of 10.8 m above the surface. This location is surrounded by water with Lake IJssel on the east side and lake Marker on the west side of the embankment. The IRGASON pointed in a 240° direction for the summer period of 2019, and to 92° as of January 2020. The latter change maximized the suitable viewing angle of
the IRGASON, taking into account the dominant wind direction and the position of the telecommunication tower and Lake IJssel. HMP155A sensors were used to measure the air temperature and relative humidity at two heights, namely 9.1 m and 10.9 m. The measurement height at the two location Stavoren and Trintelhaven differ. In our analysis we have not adjusted the measurements to an equivalent height. In theory, the small height difference will not affect the heat fluxes under the assumption of a constant turbulent flux layer.

Practical issues precluded observations of the four radiation components and water temperature at the sites. Therefore, observations of global radiation were obtained from the automated weather stations in Stavoren and Lelystad employed by the Royal Netherlands Meteorological Institute (KNMI). The KNMI weather station in Lelystad is assumed to be representative for Trintelhaven. The sub-skin water temperature, used to estimate the water vapour pressure at the air-water surface, was retrieved from the hourly sub-skin Sea Surface Temperature product with 0.05° spatial resolution derived from the Meteosat-11 satellite.
Product specification describes a target accuracy with a bias of 0.5°C and a standard deviation of 1.0°C. From this product, the grids belonging to the locations of Stavoren (52°53'06.2"N 5°21'04.1"E) and Trintelhaven (52°38'03.8"N 5°25'03.8"E) were retrieved. Data were only available during cloudless days. Furthermore, routinely measured water temperatures of Lake IJssel by the Directorate-General for Public Works and Water Management (Rijkswaterstaat) were retrieved. For this, the stations Friese Kust and Marker Wadden (orange triangles in Fig. 1a) were used, where the water temperature is measured at a depth
of 1.5 m and 1.2 m below NAP, respectively. Although typically a strong vertical water temperature gradient exists near the water surface, a good correlation ($R^2 = 0.71$ and $R^2 = 0.94$, for the summer period of 2019 at Stavoren and Trintelhaven, respectively) was found between the sub-skin water temperature from the satellite product and the water temperature measured at larger depths.

## 2.3   Data processing

The analysis in this study is focusing on the data collected during the summer periods of 2019 and 2020. Latent heat flux (LE) and sensible heat flux (H) can be calculated using the covariance of vertical wind speed and specific humidity or temperature, respectively, as follows:

$$\text{LE} = \rho_a L_v \overline{w'q'}, \tag{1}$$

$$H = \rho_a c_p \overline{w'T'}, \tag{2}$$

where $\rho_a$ [kg m$^{-3}$] is the air density, $L_v$ [J kg$^{-1}$] is the latent heat of vaporization, $c_p$ [J K$^{-1}$ kg$^{-1}$] the specific heat of air at constant pressure, $w$ [m s$^{-1}$] the vertical wind speed, $q$ [kg m$^{-3}$] the specific humidity, and $T$ [K] the air temperature.

For this, raw EC data were processed according to Foken et al. (2012). The processing steps were performed using the EddyPro software (EddyPro, 2021). This software package was chosen because it is widely used for processing eddy covariance measurements. The results compare well with the results directly obtained through the incorporated EasyFlux DL software made for the IRGASON (EasyFlux, 2017).

Firstly, the raw data were quality-controlled using several criteria in order to remove faulty or corrupted data. This included testing on completeness. If more than 5% of the high-frequency data of what is expected within the chosen averaging interval was missing it was flagged. Unrealistic values for each variable based on fixed individual thresholds were removed. Spikes were detected and eliminated in accordance with the algorithm of Mauder et al. (2013). Furthermore, according to the approach of Vickers and Mahrt (1997) the data was screened on too many bad resolution records, and too many so-called dropouts referring

to jumps in the data that continue over a longer period and therefore are not recognized as spikes. Density fluctuations were compensated using the WPL approach (Webb et al., 1980). If the signal strength of the gas analyzer falls below 70% the data was removed as well. Secondly, this quality checked dataset was further processed to obtain calculated raw fluxes. Next, coordinate rotation of the sonic anemometer using the double rotation method (Wilczak et al., 2001) was applied to correct for a not perfectly levelled sonic anemometer, and trends were removed using block averaging.

From this point, the final covariances were calculated, which were subject to two essential tests as described by Foken et al. (2004), namely (i) to test stationarity during the averaging interval (30 min) and (ii) whether there are well-developed turbulent conditions such as required for proper usage of the Monin-Obukhov similarity theory. This yielded time series containing raw fluxes. As a final step, spectral corrections were applied to account for high and low frequency losses (Massman, 2000; Moncrieff et al., 2004), as well as the correction developed by Schotanus et al. (1983) to account for the humidity effect

on the sonic temperature, which becomes specifically important for locations near water bodies. After iterating the last steps and incorporating the quality tests, a fully quality checked half-hourly flux dataset is obtained. Within this dataset, gaps of maximum one data point (i.e. half an hour) were linearly interpolated. Larger gaps were intentionally not gap filled, because this would create too much synthetic results interfering with our aim to perform a process-oriented study. Hourly data were obtained by aggregating the half-hourly dataset with no further gap filling actions taken. Daily averages were calculated from

the half-hourly dataset only if valid data were available for at least 66% of the time.

## 2.4  Flux footprint analysis

The flux footprint is computed to quantify the sampling area which contains the sinks and sources contributing to the measurement point. Additionally, the relative contribution of each upwind location to the measured flux is quantified. In figures 1b and 1c the contour lines represent 20%, 40%, 60%, and 80% of the footprint area, where the 20% line is located closest to the measurement tower. This footprint analysis helps to make decisions on which wind directions to include in the analysis based on the area of interest. The size of the footprint depends on the measurement height, atmospheric stability and surface roughness (McGloin et al., 2014). In this study we used the footprint model developed by Kljun et al. (2004) using the Flux Footprint Prediction (FFP) R code (Kljun et al., 2015). This footprint analysis showed that in Stavoren the flux data for wind directions between 163° and 349° are available for analysis of $E_{\text{water}}$, while the remaining wind directions represent the fetch over land and could therefore be used for comparison with terrestrial evaporation. At the Trintelhaven we were only interested in the fetch above Lake IJssel. Therefore, the flux data in Trintelhaven were removed for wind directions between 170° and 70° for the summer of 2019. After changing the direction of the IRGASON this yielded a larger angle from which data could be used during the summer of 2020, namely for wind directions between 0° and 170°. Given the dominant south-westerly wind direction, visualized with a windrose (inset at the top of both panels (**b**) and (**c**) in Fig. 1), this means that unfortunately a large part of the data had to be rejected at location Trintelhaven.

## 2.5  Regression model

A regression analysis was performed to explore which variable, or combination of variables, can best explain the dynamics of $E_{\text{water}}$. The 'leaps' package in R has been used to identify the best regression model, where the residual sum of squares was used as a metric to find the best model given the predictors. Variables included in this analysis were wind speed, VPD, global radiation, vertical vapour pressure gradient, and water temperature. These variables included in the analysis are generally considered to be important in describing $E_{\text{water}}$ and are partially included in the models of Dalton and Penman. To be specific, the choice of including VPD and vertical vapour pressure gradient in the regression analysis was motivated by the apparent drivers of the Dalton and Penman equations. It was decided to give preference to the use of VPD over air temperature as dependent variable in the regression analysis due to its explicit mention in the Penman equation, whereas air temperature only features implicitly in the definition of the slope of the vapour pressure gradient (s), and VPD. From the regression analysis a data-driven model was developed to estimate $E_{\text{water}}$ of Lake IJssel. This was done for both locations, Stavoren and Trintelhaven. For each individual variable, as well as for all combinations of variables, both the sum and product, a regression model was created. In the regression analysis simple linear regression models (Eq. 3), multiple linear regression models (Eq. 4), and quadratic regression models (Eq. 5) were considered. The equations of these models are prescribed as:

$$Y = \beta_0 + \beta_1 X_1 + \epsilon, \tag{3}$$

$$Y = \beta_0 + \beta_1 X_1 + \beta_2 X_2 + ... + \beta_i X_i + \epsilon, \tag{4}$$

$$Y = \beta_0 + \beta_1 X_1 + \beta_2 X_1^2 + \epsilon, \tag{5}$$

where $Y$ is the dependent variable, $X_i$ the explanatory variable(s), $\beta_0$ the intercept, $\beta_i$ the parameter(s), and $\epsilon$ the error term. The explanatory variable(s) $X_i$ was prescribed to be either a single variable or the product of multiple variables, except for equation 4, where $X_i$ can only be a single variable. Statistical significance (P < 0.05) was tested. From the multitude of regression models that is resulting from the regression analysis the best model was chosen, using the metrics adjusted $R^2$ and

RMSE to evaluate the fit of each regression model. We not only aimed for finding the best model, but we were also interested to find the best simple model that uses maximum two variables, while still able to explain the dynamics of $E_{\text{water}}$ well. The summer season of 2019, here taken as 01/05 – 31/08, was used as the training dataset to calibrate the data-driven model, and the dataset of the summer of 2020 was used for validation. The analysis was performed at hourly and daily timescales.

The same procedure as described above was repeated but now solely using routinely measured observations. This was

done to explore the possibility of using routine observations to make accurate estimations of $E_{\text{water}}$, instead of continuing the labour intensive and expensive measurements with the eddy covariance systems. As described previously (see Sect. 2.2), data of automatic meteorological stations of the KNMI were used to obtain data of global radiation, complemented with air temperature, wind speed, and relative humidity that are routinely measured at these stations. There are no routine observations available of the skin water temperature of the lake. As an alternative, the use of water temperature data routinely measured by

Rijkswaterstaat at depths ranging from 1.2 to 1.5 m was explored.

The resulting regression model was compared to the models of Dalton, Penman and Makkink (see Eq. 6, 8 and 12) to give an indication on the (dis)agreement of the variables involved in explaining the dynamics of $E_{\text{water}}$ and its form. Dalton's model is based on the empirical relationship that was found between evaporation and the product of a wind function and the vertical vapour pressure difference, which can be written as:

$$\text{LE}_{\text{Dalton}} = f(u)(e_s(T_0) - e_2), \tag{6}$$

in which $e_2$ [kPa] is the vapour pressure at 2 m height, $e_s(T_0)$ [kPa] is the saturation vapour pressure at the surface, and $f(u)$ [W m$^{-2}$ kPa$^{-1}$] is the wind function which takes the following form (Penman, 1956; De Bruin, 1979):

$$f(u) = 37 + 40u_2, \tag{7}$$

in which $u_2$ [m s$^{-1}$] is the wind speed at 2 m height. Although the representation of the Dalton models may seem simple,

obtaining reliable measurements of surface temperature is challenging.

Similarly, Penman's equation, which is derived from Dalton's equation, can be written as:

$$\text{LE}_{\text{Penman}} = \frac{\gamma}{s+\gamma} f(u)(e_s(T_2) - e_2) + \frac{s}{s+\gamma} Q^*, \tag{8}$$

in which $s$ [kPa $°C^{-1}$] is the slope of the saturated vapour pressure curve at air temperature, $\gamma$ [kPa $°C^{-1}$] the psychrometric constant, $e_s(T_2) - e_2$ [kPa] the vapour pressure deficit (VPD) at 2 m height, and $Q^*$ [W m$^{-2}$] the available energy at the surface

which can be defined as $R_n - G$, where $R_n$ [W m$^{-2}$] is net radiation and $G$ [W m$^{-2}$] is the downward heat flux from the water surface. Net longwave radiation was calculated following equations 2.24 and 2.28 from Moene and van Dam (2014):

$$L_{\text{in}} = \epsilon_a \sigma T_a^4, \tag{9}$$

$$L_{\text{out}} = L_{e,out} + (1 - \epsilon_s)L_{in}, \tag{10}$$

in which $L_{in}$ [W m$^{-2}$] is the incoming longwave radiation, $L_{out}$ [W m$^{-2}$] is the outgoing longwave radiation, $\epsilon_a$ [–] is the apparent emissivity that is a function of the fraction of cloud cover, $\sigma$ [= 5.67·10$^{-8}$ W m$^{-2}$ K$^{-4}$] the Stefan-Boltzmann constant, $T_a$ [K] the air temperature, $L_{e,out}$ [W m$^{-2}$] is the emitted longwave radiation, and $\epsilon_s$ the surface emissivity. Net shortwave radiation was calculated as (Allen et al., 1998):

$$K^* = (1 - \alpha)K_{in}, \tag{11}$$

in which $K^*$ [W m$^{-2}$] is the net shortwave radiation, $K_{in}$ [W m$^{-2}$] the global radiation, and $\alpha$ [–] the albedo values for which monthly values were calculated as a function of latitude (Cogley, 1979).

Priestley and Taylor (1972) found that the aerodynamic term of Penman's equation is approximately one-fourth of the radiation term. Makkink (1957) found that this equation could be simplified even more for estimating daily evapotranspiration from well-watered surfaces. Under these conditions $G$ is assumed to be negligible and a constant ratio between net radiation and global radiation of on average 0.5 can be assumed, which results in the following equation (Makkink, 1957):

$$\text{LE}_{\text{Makkink}} = 0.65\frac{s}{s+\gamma}K_{in}, \tag{12}$$

According to Penman's derivation $G$ is assumed to be negligible for short timescales of a day to several days for shallow water surfaces, similar to land surfaces, and the term is often ignored because of the difficulty of measuring it. However, for water bodies of several metres depth the impact on the energy balance by neglecting $G$ can be considerable (Keijman, 1974; de Bruin, 1982; Tanny et al., 2008; van Emmerik et al., 2013). For these water bodies $G$ should be considered as a result of temperature changes integrated over the volume of the water column in contrast to a land surface where the impact of $G$ is more superficial. It should be clearly noted, that although Lake IJssel is a lake of several metres depth, we have neglected $G$ in the following analyses because (1) we have not been able to measure it, and (2) in order to adhere to how Penman's equation is typically employed for shallow water surfaces.

## 3 Results

### 3.1 Data quality and quantity

To guarantee data quality of the flux measurements quality checks were performed as described in section 2.3. After the quality control 66% and 64% of the latent heat flux data were available in 2019 for Stavoren and Trintelhaven locations, respectively. In 2020 this number was lower, with 49% and 59% for Stavoren and Trintelhaven, respectively. Part of the available quality checked data needed to be rejected based on the flux footprint analysis. This led to a reduced number of available data of 42% and 13% of total data in 2019 for Stavoren and Trintelhaven. In 2020 the total available latent heat flux data was 33% and 19% for Stavoren and Trintelhaven, respectively. The reduction of available data at Trintelhaven location is larger given the

combination of the dominant south-westerly winds and the location of the instrument at the south-west border of Lake IJssel (see Fig. 1a). The number of total available flux data are at the lower end, but not unusual, of the data availability reported in other studies on lakes which is typically in the range of 16% – 59% (Vesala et al., 2006; Nordbo et al., 2011; Bouin et al., 2012; Mammarella et al., 2015; Metzger et al., 2018).

5     No clustering has been found in the availability of latent heat flux data during daylight hours (6hr – 21hr) compared to the night (21hr – 6hr). In the final half-hourly dataset, latent heat flux data were available during 56% of the total daytime half-hours in Stavoren in the summer of 2019, and 49% during night-time. In the summer of 2020 this was 49% and 40% for daytime and night-time, respectively. For location Trintelhaven the corresponding fractions were 10% and 18% during daytime and night-time in the summer of 2019, respectively. The difference between daytime and night-time fractions was smaller during the summer of 2020, with 18% and 21% of data available, respectively.

## 3.2   Meteorological conditions

Similar dynamics were observed at both locations Stavoren and Trintelhaven. Here we only show the meteorological conditions observed in Stavoren during the period 01/05/2019 – 30/09/2019 (Fig. 2; see Appendix A for the meteorological conditions in Stavoren in the summer period of 2020). This figure illustrates the dynamics and trends of the meteorological variables and the heat fluxes before, during and after the summer period and to explore if any visible lags for instance would occur at this timescale. Table 1 provides an indication of the source of the data for the variables that will be elaborated on in this section. According to the measurements of the KNMI both measurement stations received on average the same amount of global radiation for the period 01/05/2019 – 30/09/2019, namely: 208 W m$^{-2}$ in Stavoren, and 204 W m$^{-2}$ at Lelystad, the latter assumed to be representative for Trintelhaven. The average air temperature that we measured with the HMP155A sensor is lower in Stavoren (16.4 °C), than in Trintelhaven (18.0 °C). These temperatures are higher than the climatological mean observed by the KNMI (period 1991 – 2020) for the same months: 15.9 °C in Stavoren, 16.1 °C in Lelystad (KNMI, 2021a). The water temperature measured by Rijkswaterstaat in the vicinity of Stavoren was on average 17.9 °C, and the water temperature at location Marker Wadden close to the Trintelhaven was on average 18.3 °C. The time series of water temperature shows a more smooth, attenuated and lagged signal compared to the air temperature. At both measurement locations the measured wind speed observed with the IRGASON is similar, with an average wind speed of 5.8 m s$^{-1}$ in Stavoren and 5.6 m s$^{-1}$ at Trintelhaven, without a distinct seasonal pattern. The vapour pressure that we measured follows the seasonal cycle of the air temperature, and has a mean value of 1.4 kPa in Stavoren and 1.6 kPa at Trintelhaven. In the bottom panel the observed turbulent fluxes are shown (Fig. 2e). The sensible heat flux remains consistently low throughout the summer period, with an average value of 17 W m$^{-2}$ in Stavoren and 25 W m$^{-2}$ at Trintelhaven. The latent heat flux is more than four times as high on average, with values of 88 W m$^{-2}$ on average in Stavoren and 91 W m$^{-2}$ in Trintelhaven. Basic statistics on the observed latent heat flux can be found in Appendix B. The latent heat flux displays similar trends as the measured wind speed, indicating that the two variables are correlated (R$^2$ = 0.61). Based on the average rates of sensible and latent heat flux the Bowen ratio is 0.19 in Stavoren and 0.27 at Trintelhaven.

**Table 1.** Sources of data used in this study. The variables measured at our locations in Stavoren at 7.5 m and Trintelhaven at 10.8 m were sampled at high frequency (20 Hz), and aggregated to hourly data. The data retrieved from KNMI stations in Stavoren and Lelystad are provided at hourly timescales and were measured at 1.5 m height above land surface, and 10-minute water temperature data measured by Rijkswaterstaat at locations Friese Kust at -1.5 m NAP and Marker Wadden at -1.2 m NAP were aggregated to hourly timescales.

|  | Own observations | KNMI | Rijkswaterstaat |
|---|---|---|---|
| $K_{in}$ |  | X |  |
| $T$ |  |  |  |
|   - $T_{air}$ | X |  |  |
|   - $T_{air,climatology}$ |  | X |  |
|   - $T_{water}$ |  |  | X |
| $u$ | X |  |  |
| $e_z$ | X |  |  |
| LE | X |  |  |
| $H$ | X |  |  |

### 3.3 Diurnal and intra-seasonal variability of latent heat flux

The monthly average diurnal variability of observed LE, based on hourly data, are shown in the top panels of figure 3 for location Stavoren for the same period as in figure 2 (i.e. 01/05/2019 – 30/09/2019). The diurnal variability of LE does not have a strong diurnal cycle, but is rather constant throughout the day and night, which is in contrast to terrestrial evaporation which typically peaks during the day. Only in August the LE signal shows a distinct peak during the late afternoon and lower values during the night and early morning. The highest average diurnal LE is reached in July as indicated by the number in the top left-hand corner of each panel. Global radiation measured at the KNMI meteorological stations of Stavoren is shown in the middle panels. A clear distinctive diurnal cycle is visible with a peak in the afternoon, and the highest average value is found in June at both locations. The global radiation has served as input for the commonly used radiation-based models of Penman (1948) and Makkink (1957), of which the average diurnal cycles are shown in the lower panels of the same figures. Recall that G was omitted in Penman's model in this analysis (see Sect.2.5). They closely follow the pattern of the global radiation but with a lower amplitude. Highest average LE values are found in June for these models, in contrast to observed LE which is found to be highest a month later. In the lower panels the average diurnal cycle of LE that follows from Dalton's model is shown as well. There is no strong diurnal cycle visible, similar to the observed LE, but generally rates are highest during daytime. Highest average LE values are found in August. The observed monthly average diurnal dynamics observed were found similar in Trintelhaven (see Appendix C).

Extension of the time series to a complete year in order to visualize the seasonal variability, shows a clear seasonal cycle that is reminiscent of the influence of a radiation component on $E_{water}$ (Fig. 4). The bars represent the monthly average $E_{water}$ rate based on half-hourly data. For the year 2019 the evaporation rate is highest in July for both locations: 4.3 mm d$^{-1}$ in Stavoren and 3.8 mm d$^{-1}$ in Trintelhaven. Lowest values are 0.2 mm d$^{-1}$ in Stavoren (February) and 0.6 mm d$^{-1}$ in Trintelhaven (December, note: data of January/February of 2019 are lacking). In 2020 similar rates are found in winter: 0.1 mm d$^{-1}$ in

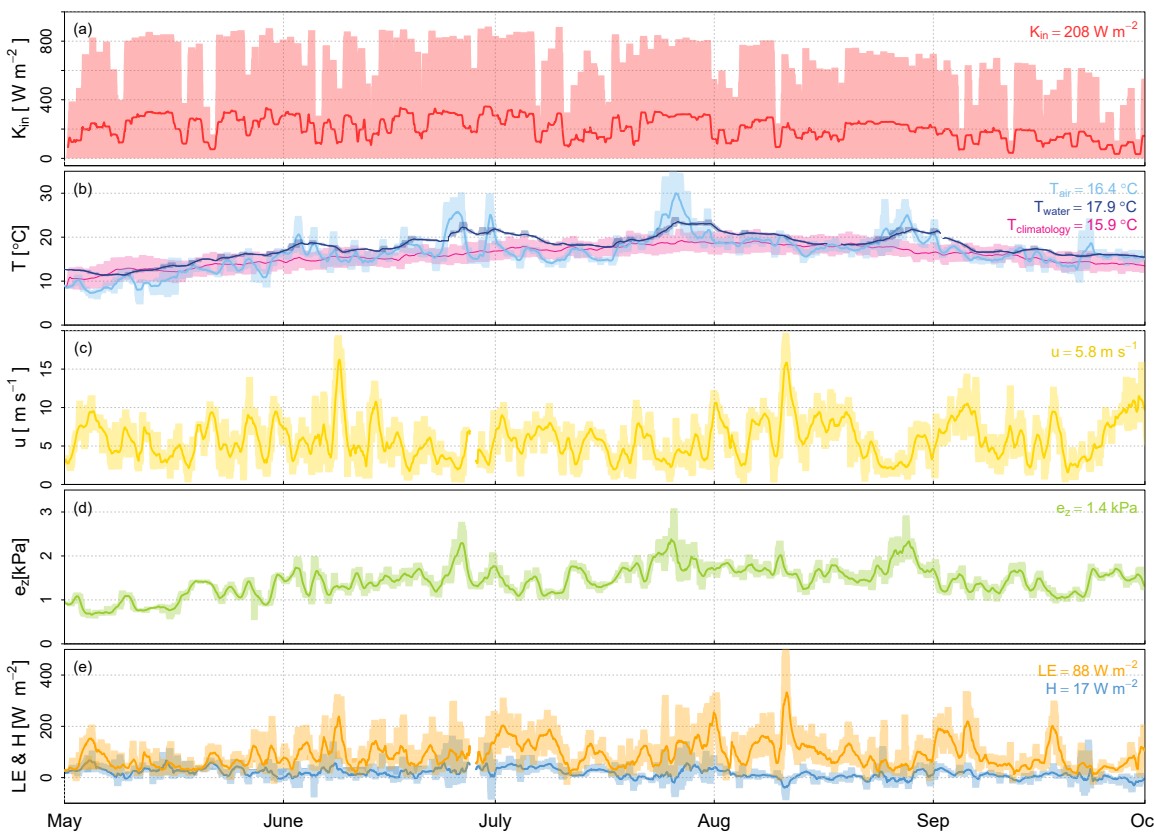

**Figure 2.** Meteorological conditions in Stavoren in 2019 showing running daily means of global radiation (**a**), air temperature (current and climatology) and water temperature (**b**), wind speed (**c**), vapour pressure (**d**), and turbulent fluxes (**e**). The shaded area represents the range between minimum and maximum observed values and the numbers reported at the top right of each panel provide the average values of the respective variables during the presented months.

Stavoren (December) and 0.5 mm d$^{-1}$ in Trintelhaven (November), while the summer of 2020 now has a dip in July instead of being the peak.

### 3.4 Drivers of open water evaporation

Based on the historical theory it is known that governing factors of $E_{water}$ include the gradient of vapour pressure above the water surface and some measure for the strength of the turbulence (Dalton, 1802; Penman, 1948; De Bruin, 1979; Brutsaert, 1982). These variables form the ingredients of the so-called aerodynamic method or mass transfer approach (Brutsaert, 1982). Here we tested which variable or combination of variables can best explain the dynamics of observed $E_{water}$ at Lake IJssel at both hourly and daily temporal resolution. The variables included in this analysis are global radiation, vertical gradient of vapour pressure, vapour pressure deficit, sub-skin water temperature, and wind speed. It should be reminded that air temperature

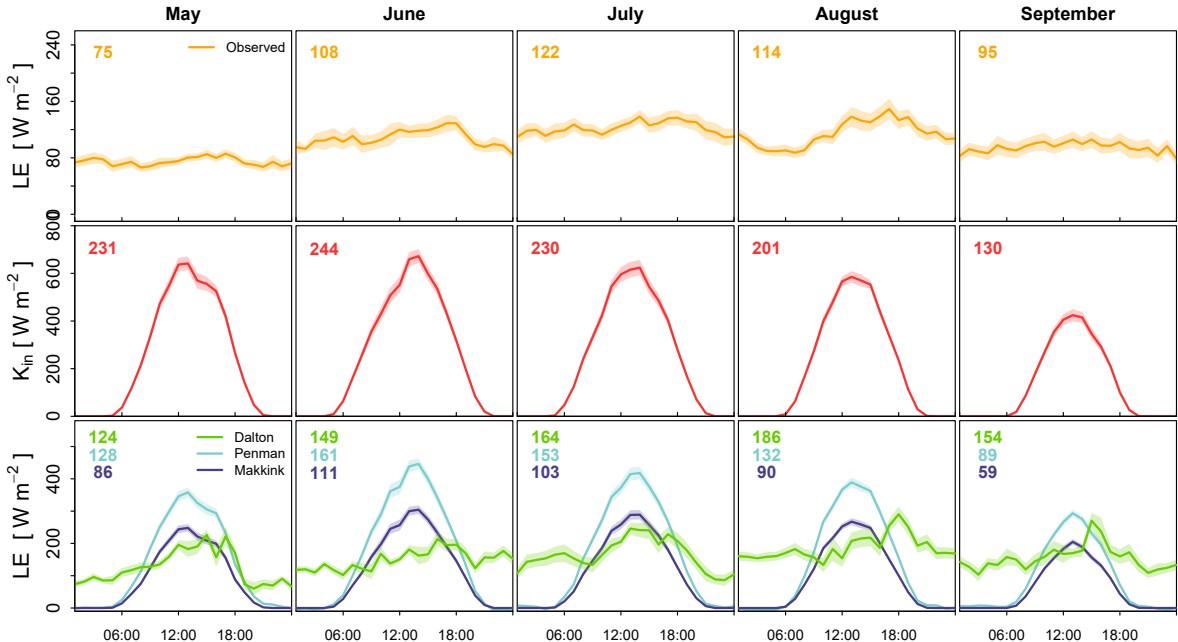

**Figure 3.** Illustration of the decoupling in Stavoren in 2019 between monthly average diurnal cycles of observed latent heat flux (top) and global radiation (middle), the latter forming the basis of the frequently used evaporation models of Penman (1948) and Makkink (1957). These models are shown together with the model of Dalton (1802) at the bottom panels. Note that some variables included in the evaporation models are measured at larger heights than the 2 m that are prescribed (see Eq. 6 – 12). Additionally, all three models are generally used on a daily basis, but they are presented here to show the underlying daily cycle. The shaded area represents the uncertainty, which is defined as the standard deviation divided by the square root of the number of observations. Average daily means of the respective variables are indicated by the number at the top left of each panel displaying the average course over the summer period.

was not explicitly included in the regression analysis as explained in Section 2.5. We expect that including air temperature as a separate dependent variable might have explained a part of the evaporation dynamics, since air temperature affects surface temperature through the sensible heat flux. In turn, the surface temperature affects the vapour pressure gradient and thus evaporation. However, due to the large thermal buffer of a water body we expect that there is a less direct coupling between 5  the sensible heat flux and the latent heat flux at short timescales.

The proportion of the dynamics of $E_{water}$ that can be explained by the variable or combination of variables is shown in Venn diagrams with the adjusted coefficient of determination ($R^2$) written inside (see Fig. 5). The higher the adjusted $R^2$, the more blue the colour is. Adding a variable will not always result in a higher adjusted $R^2$ value, because the adjusted $R^2$ takes into account the degrees of freedom. Therefore, it can lead to a decrease in adjusted $R^2$ if the added variable only slightly correlates 10  with the dependent variable. Values of adjusted $R^2$ were removed if the model fit was found to be insignificant ($p < 0.05$). Venn diagrams (**a**) and (**c**) on the left-hand side illustrate the analysis based on hourly data, and the diagrams (**b**) and (**d**) on the right-hand side are based on daily data. The outer 'leaves' of the diagram represent the single variables, while moving towards

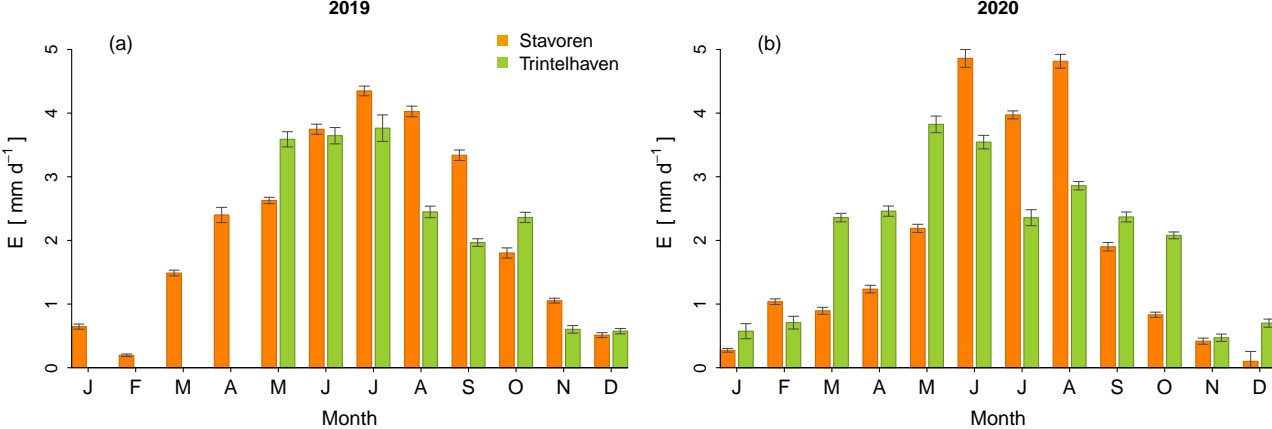

**Figure 4.** Yearly cycle of observed open water evaporation in Stavoren (orange) and Trintelhaven (green) based on half-hourly data for both years 2019 and 2020. The bars indicate the monthly average evaporation and the whiskers represent the uncertainty, which is defined as the standard deviation divided by the square root of the number of observations.

the centre of the diagram combinations of variables are taken into account to explain the dynamics of $E_{\text{water}}$. Both the sums and the products of the combined variables are analysed (Fig. 5 and Appendix D, respectively). Based on these diagrams, the decision was made which variables to include in the data-driven model to estimate $E_{\text{water}}$. The prominent blue colour connected to wind speed already tells us that this is an important variable to include, which is in agreement with Dalton's model and which was visible from figure 2 ($R^2 = 0.61$). Global radiation has the lowest adjusted coefficient of determination, which agrees with our findings in figures 2 and 3.

### 3.4.1 Calibration

For both locations two models were developed, namely: (i) a model that includes the variable(s) that explain(s) most of the variability of $E_{\text{water}}$, thus with the highest adjusted $R^2$, and (ii) a model that only uses one or two variables which are still able to explain a significant portion of the variability of $E_{\text{water}}$ (number depicted in red in the Venn diagrams Fig. 5 and D1). At the hourly timescale the best model fit, indicated by the highest $R^2$ value, is reached when the sum of (almost) all five variables are included ($R^2$=0.74 and $R^2$=0.69 at Stavoren and Trintelhaven, respectively). Moving from the outer leaves towards the centre of the diagram, we find that the most simple hourly model that still explains a large portion of the variance (i.e. the red numbers) includes only wind speed and vertical gradient of vapour pressure ($R^2$ is 0.70 in Stavoren (Fig. 5), and 0.69 in Trintelhaven (Fig. D1)). This is in agreement with Dalton's model. At the daily timescale in Stavoren we see a shift in which variables are included in the 'simple' model. The sum of wind speed and water temperature reaches the highest $R^2$ (= 0.77). Unfortunately, very little data points ($N = 10$) were left at the daily timescale at Trintelhaven, which is leading to many insignificant model fits (values are removed from those intersecting areas). A couple of models were found to be significant though, where the sum of wind speed and vertical gradient of vapour pressure again has the highest $R^2$ value (= 0.97). The relatively high adjusted $R^2$

values of these simple model fits, compared to models including more than two variables, indicate that the added value of using more than two variables is virtually nil. The results from the Venn diagrams form the base to create the data-driven models for which the data collected in 2019 is used. Both linear and quadratic regression models were considered as explained in section 2.5.

The results presented in the Venn diagrams is used to formulate the regression models. Both the 'simple' and 'best' fitted model were evaluated, based on hourly data and daily data, and are presented in the top panels ((**a**) and (**b**)) of figures 6 (Stavoren) and 7 (Trintelhaven). At both locations the simple models turned out the be the best models. For both locations the regression analysis based on hourly data has shown that the combination of wind speed and vertical gradient of vapour pressure explains most of the variability of $E_{\text{water}}$, leading to $R^2$ values of 0.74 and 0.70 at Stavoren and Trintelhaven, respectively. The

sum of wind speed and vertical gradient of vapour pressure were the most important ingredients to explain $E_{\text{water}}$ at a daily timescale at Trintelhaven ($R^2 = 0.98$). In Stavoren the sum of wind speed and water temperature results in $R^2 = 0.8$. Without predetermination of the variables, we found the same ingredients as used in the Dalton model as the most important drivers of $E_{\text{water}}$ at hourly timescales. To determine if the coefficients that were found for the hourly regression models of the two measurements locations significantly differ, an ANOVA statistical analysis was performed (see Appendix E). This analysis

shows that inclusion of the measurement site matters ($p < 0.05$). Therefore, we cannot rule out that the sites are different.

### 3.4.2    Validation

The models are validated using the data collected in 2020 and the results are presented in the lower panels ((**c**) and (**d**)) of the same figures (Fig. 6 and 7). The adequate $R^2$ values in the validation give confidence in the performance of the models. Validation of the hourly regression model in Stavoren has a higher $R^2$ value ($R^2 = 0.84$) compared to the calibration period

($R^2 = 0.74$). In an attempt to explain why a higher $R^2$ value occurs during the validation of the model, we have swapped the calibration (now summer of 2020) and validation (summer of 2019) period. The coefficients of the regression model were recalculated. The $R^2$ value of the validation was now found to be smaller than during the calibration. This provides an indication that the difference in $R^2$ values during calibration and validation seem to be related to the conditions during the two distinct summer periods, and it gives confidence that the model performs well. Additional to the data-driven models, the estimated

daily evaporation rates using Makkink's model are plotted for reference as well, because this model is currently used in the operational water management of Lake IJssel. Makkink's model fails to explain the dynamics of $E_{\text{water}}$ on a daily temporal resolution with $R^2$ values near zero.

### 3.4.3    Models based on routinely measured variables

The analysis as described above was repeated but now using only routinely measured observations of meteorological variables

at 2 m height by KNMI and observations of water temperatures by Rijkswaterstaat (Tables 2a and 2b). This was done to get to know the possibility of using these routine measurement to estimate $E_{\text{water}}$, instead of using the expensive and labour intensive eddy covariance instruments. The regression models found using these routine observations are, especially for the location Stavoren, well able to explain the dynamics of $E_{\text{water}}$, where wind speed and vertical gradient of vapour pressure are

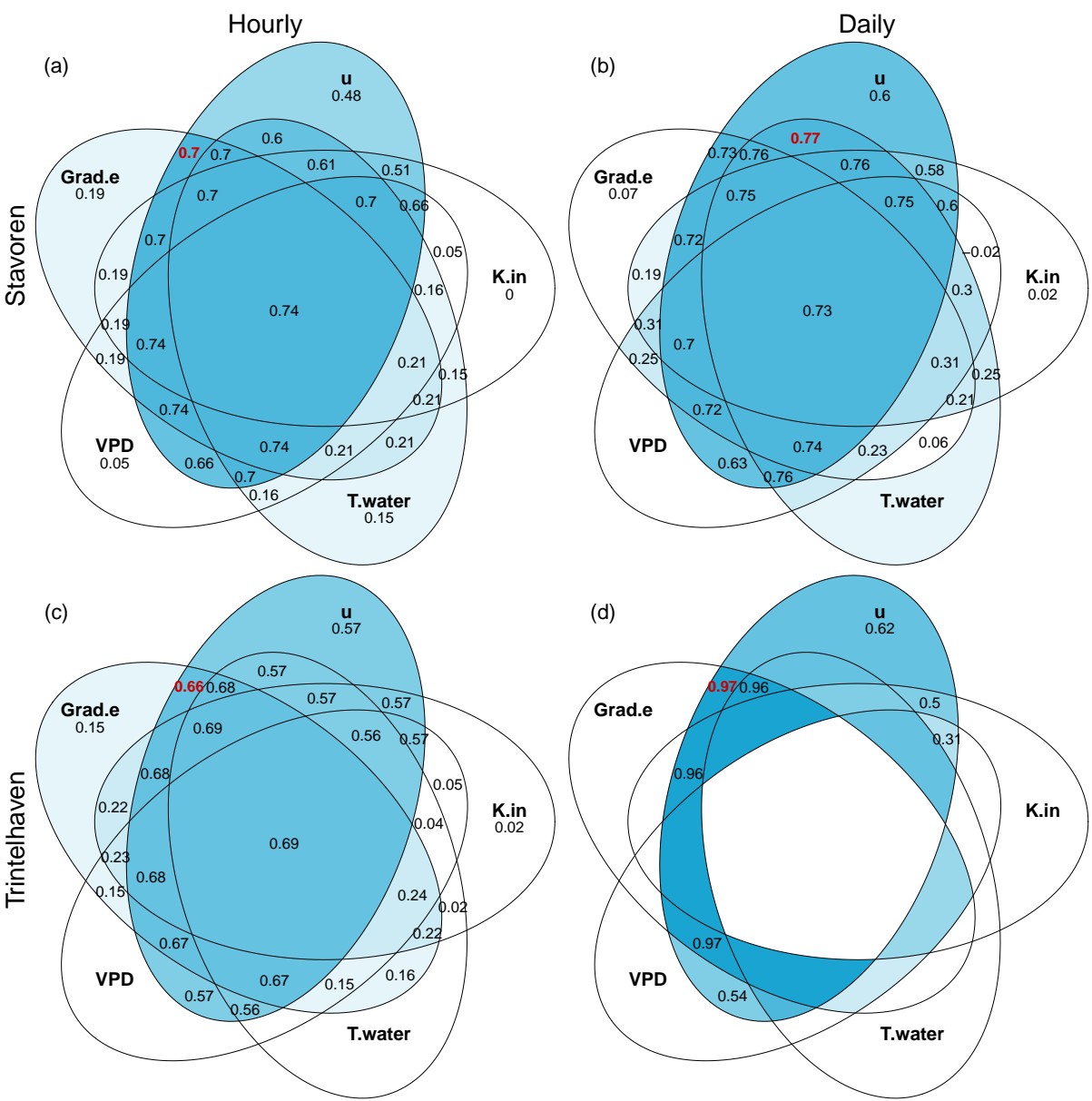

**Figure 5.** Systematic exploration which variable or combination of variables (sums) can best explain the dynamics of open water evaporation. The outer 'leaves' of the Venn diagram represent the model fit based on the single variables, while moving towards the centre of the diagram the summed combination of variables are represented. Within each leaf the adjusted $R^2$ value is depicted. The higher this value, the more blue the colour of the leaf. The red number indicates the highest $R^2$ value, indicating the best combination found for a maximum of two variables, i.e the best 'simple' model. Values of $R^2$ were removed if the model fit was found to be insignificant ($p < 0.05$). The analysis is based on data from the summer of 2019 and is performed at hourly timescale (left panels: Stavoren (**a**), Trintelhaven (**c**)) and daily timescale (right panels: Stavoren (**b**), Trintelhaven (**d**)).

the main ingredients for the simple model ($R^2$=0.83 using hourly data and $R^2 = 0.86$ using daily data). Validation using data from summer 2020 also yields satisfactory results, with high $R^2$ of 0.78 and 0.82 for hourly and daily data, respectively. The results of the simple model for the location of Trintelhaven fall short compared to Stavoren, with $R^2$ of 0.29 and 0.48 for hourly and daily data, respectively. An explanation for this may be that the location of the routine observations is situated at a larger

5    distance from the aimed location of Trintelhaven compared to Stavoren (Fig. 1a). Again at Trintelhaven, $R^2$ values during the validation period are found to be higher than during the calibration period, which seem to be related to the different conditions during the two summers.

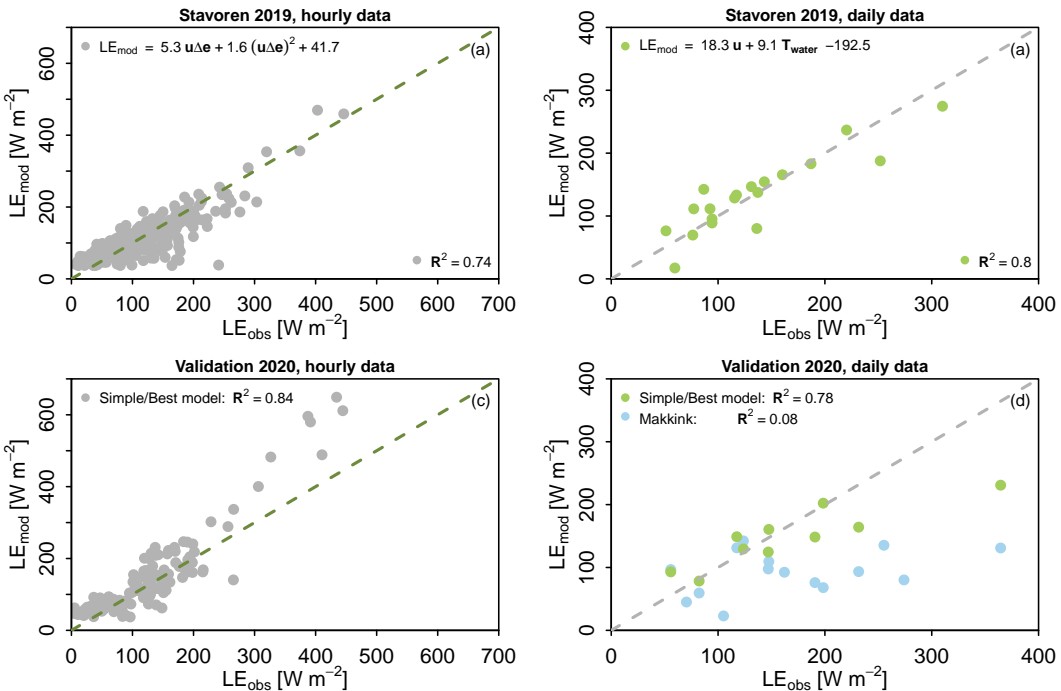

**Figure 6.** Evaluation of the developed 'simple' and 'best' data-driven models based on our own observations during the summer of 2019 (calibration) and 2020 (validation) to estimate open water evaporation in Stavoren at hourly (**a**) and daily (**b**) timescales. The model equations are shown in the top panels, which differ from the models found based on routinely measured observations (see Table 2a). Results of the validation of the models are presented in panels **c** and **d** for hourly and daily timescales, respectively. The simple model turned out to be the best model at the same time. Results of estimated evaporation using Makkink's model (light blue) is added to the validation plots as a reference. Model performance is indicated by the values of the coefficient of determination ($R^2$) shown in each panel.

## 4    Discussion

Our results have shown that the diurnal cycle of observed $E_{water}$ shows a distinctively different pattern compared to evaporation

10   estimated using the evaporation models of Penman (1948) and Makkink (1957). It should be reminded that in this study we have

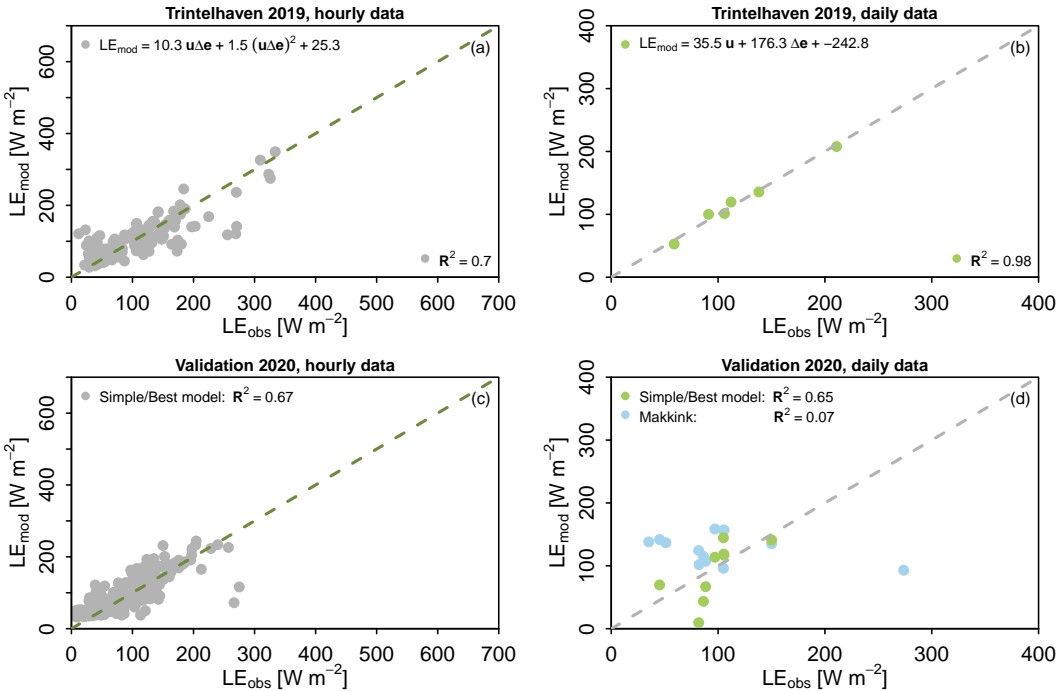

**Figure 7.** Evaluation of the developed 'simple' and 'best' data-driven models based on our own observations during the summer of 2019 (calibration) and 2020 (validation) to estimate open water evaporation in Trintelhaven at hourly (**a**) and daily (**b**) timescales. The simple model was found to be the best model and is given light coloured dots. The model equations are shown in the top panels, which differ from the models found based on routinely measured observations (see Table 2b). Results of the validation of the models are presented in panels **c** and **d** for hourly and daily timescales, respectively. Results of estimated evaporation using Makkink's model (light blue) is added to the validation plots as a reference. Model performance is indicated by the values of the coefficient of determination ($R^2$) shown in each panel.

omitted $G$ in Penman's model. The estimated evaporation by the models of Penman and Makkink resembles better the cycle that was observed at our station in Stavoren when we filtered on wind directions coming from the land surface, i.e. representing terrestrial evaporation (see Fig. 8). In contrast to the observed terrestrial evaporation, the observed $E_{water}$ is not directly coupled to global radiation at these time scales, which is demonstrated by the difference in diurnal variability between global radiation and observed LE (middle and upper panels of figure 3). Note that the relation between $E_{water}$ and other components of the radiation budget could not be studied, because of the lack of observations of these components. In combination with absent data on $G$, this prevented us from fully capturing the role of net radiation in the energy balance of the lake, and thus with the warming and cooling of the lake, which relates to evaporation through the water surface temperature. A better agreement of the observed diurnal cycle was found for Dalton's model, which is more constant throughout the day. Nevertheless, until now Makkink's model has been used as a base for calculating $E_{water}$ at Lake IJssel (Jansen and Teuling, 2020). We have shown that Makkink's model is not able to explain the dynamics of $E_{water}$ for the summer period at the daily timescale. Such a radiation-

**Table 2.** Evaluation of the developed 'simple' and 'best' data-driven models based on routinely measured observations (see Section 3.4) during the summer of 2019 (calibration) and 2020 (validation) to estimate open water evaporation in Stavoren (Table 2a) and Trintelhaven (Table 2b) at hourly and daily timescales. These models presented here are independent of the results found based on our own observations. Results of estimated evaporation using Makkink's model are provided as a reference. Model performance is indicated by the values of the coefficient of determination ($R^2$).

| (a) **Stavoren** | Model equation for calculating $LE_{mod}$ | $R^2_{calibration}$ | $R^2_{validation}$ |
|---|---|---|---|
| Hourly | | | |
|    Simple data-driven model | 29.4 $\mathbf{u}\Delta\mathbf{e}$ − 4.1 | 0.83 | 0.78 |
|    Best data-driven model | 1.4 $\mathbf{u}\Delta\mathbf{e}\mathbf{T_{water}}$ + 2.8 | 0.84 | 0.81 |
| Daily | | | |
|    Simple data-driven model | 30.8 $\mathbf{u}\Delta\mathbf{e}$ − 9.2 | 0.86 | 0.82 |
|    Best data-driven model | 1.5 $\mathbf{u}\Delta\mathbf{e}\mathbf{T_{water}}$ − 3.8 | 0.91 | 0.80 |
|    Makkink | 0.65 $\frac{s}{s+\gamma}$ $\mathbf{K_{in}}$ | – | -0.02 |

| (b) **Trintelhaven** | Model equation for calculating $LE_{mod}$ | $R^2_{calibration}$ | $R^2_{validation}$ |
|---|---|---|---|
| Hourly | | | |
|    Simple data-driven model | 13.0 $\mathbf{u}$ + 98.1 $\Delta\mathbf{e}$ − 21.8 | 0.29 | 0.51 |
|    Best data-driven model | 18.5 $\mathbf{u}$ + 128.6 $\Delta\mathbf{e}$ − 29.0 $\mathbf{VPD}$ − 42.0 | 0.42 | 0.54 |
| Daily | | | |
|    Simple/Best data-driven model | 28.9 $\mathbf{u}\Delta\mathbf{e}$ + 22.7 | 0.48 | 0.87 |
|    Makkink | 0.65 $\frac{s}{s+\gamma}$ $\mathbf{K_{in}}$ | – | 0.003 |

based approach (including a potential linear correction factor) might lead to the correct daily or monthly evaporation sums, but it will be for the wrong reason.

Resulting from the data-based modelling that was performed we found that not radiation, but a combination of wind speed and vapour pressure gradient are the most important ingredients to explain the variance of $E_{\text{water}}$ at short timescales. This is similar to what has been found by studies of for instance Blanken et al. (2011) and McGloin et al. (2014), and it was noticed that intraseasonal variations of $E_{\text{water}}$ can be linked to synoptic weather variations through these variables (Lenters et al., 2005a; MacIntyre et al., 2009; Liu et al., 2011; Woolway et al., 2020). The same ingredients of wind speed and vapour pressure gradient were used in the model by Dalton (1802). By combining and rearranging equations 6 and 7 we can write Dalton's model in the following form:

$$\text{LE}_{\text{Dalton}} = 37\Delta e + 40u_2\Delta e, \tag{13}$$

in which $\Delta e$ [kPa] is $e_s(T_0) - e_2$ [kPa]. This highlights the similarity of the functional form of Dalton's model and of our data-driven model that resulted from the regression analysis (see Fig. 6 and 7). When the exact functional form of equation 13, i.e. LE $= a\Delta e + bu_z\Delta e$, is fitted to our hourly observations of 2019, we find that coefficients $a$ and $b$ differ (not shown here). This difference is likely related to the height at which our measurements were done (10.8 and 7.5 metres above the surface

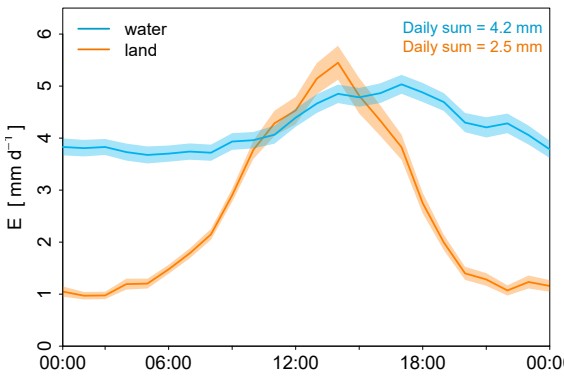

**Figure 8.** Comparison of the average diurnal cycle of open water evaporation (blue) and terrestrial evaporation (orange) observed during the summer period 2019 in Stavoren. The shaded area represents the uncertainty band, which is defined as the standard deviation divided by the square root of the number of observations.

in Stavoren and Trintelhaven, respectively) compared to the standard height of 2 metres. We have found that Penman's model seemed not suitable for estimating $E_{\text{water}}$ over the summer period in the form that we have employed it (i.e. with $G$ omitted). However, when we extended the time series from only the summer period to the whole year a clear yearly cycle was visible, with a peak in summer that is similar to the cycle of (available) radiation, and thus to estimates of evaporation using Penman's

model (see Fig. 9a). The benefit of Penman's model in this case is that it can easily be decomposed into an aerodynamic term and a radiation term. The individual terms are presented in figure 9b. Here a clear distinction between the yearly cycle of the two Penman terms is visible, where the radiation term has a distinct cycle with a peak in June, while the aerodynamic term is more constant over the year. This resembles the constancy of observed $E_{\text{water}}$ found in the diurnal cycle (see Fig. 3).

       Another phenomenon that could affect the yearly cycle of evaporation is lake water stability and thus mixing depth within

the lake. Seasonal changes in lake water stability affects the surface temperature and therefore evaporation rates through the vapour pressure gradient. Evaporation in turn has a cooling effect on the surface temperature, which increases potential mixing. Supported by a preliminary study where mixing depths were simulated using the model FLake (Voskamp, 2018), we assume that during 70% of the time Lake IJssel is fully mixed. This number is not surprising given the fact that during nighttime evaporation continues, and with wind speeds that are on average 5.8 m.s-1. In addition, the inflow of the river IJssel into the

lake is likely to support mixing as well. During summer it is more likely that stable conditions occur and we cannot directly assume fully mixed conditions. However, we considered this phenomenon beyond the scope of the current study.

       Linking back to the shorter time series spanning the summer months of 01/05 – 31/08, we can see that there is even a simple linear relationship present between observed $E_{\text{water}}$ and the aerodynamic term of Penman's model. In contrast, the variability of radiation is uncorrelated to observed $E_{\text{water}}$ (see Table 3). The high correlation between the aerodynamic term of Penman,

which includes similar variables as Dalton's model, and observed $E_{\text{water}}$ strengthens the finding that our data-driven model is embedded into the well-known theory. We are aware that using a statistical modelling approach has its limitations because it does not account for the actual physical processes the way that it might be included in physically-based models such as FLake

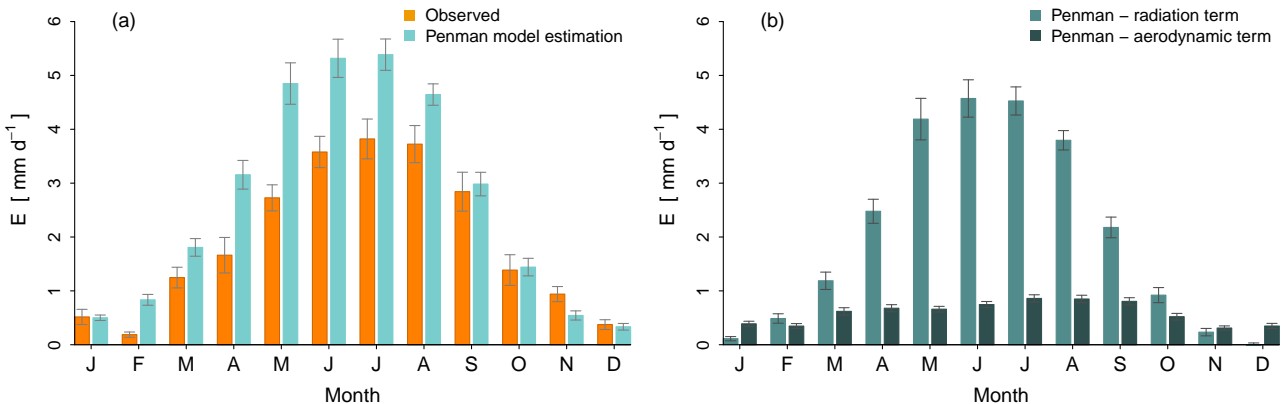

**Figure 9.** Comparison of the annual cycle of observed open water evaporation in Stavoren (orange) and estimated evaporation (blue) using Penman's model (**a**) based on daily data in 2019. The individual terms of Penman's model are displayed in the right panel (**b**), which shows the similarity of the annual cycle between observed open water evaporation and the radiation term of Penman's model. The bars indicate the monthly average evaporation and the whiskers represent the uncertainty, which is defined as the standard deviation divided by the square root of the number of observations.

(Mironov, 2008) for modelling lake evaporation. However, in such physically-based models empirical relations are included as well (e.g. the wind function in Dalton's and Penman's model) and parameters need to be statistically estimated. Furthermore, if drivers of open water evaporation appear to be a function of the temporal resolution, it should be concluded that models, including physical models, can only be properly used at the right temporal resolution. Considering this, we think that statistical modelling is a clean and simple approach that can provide a direct indication and insight of the most relevant input parameters involved in explaining the variation of evaporation, without making a priori assumptions on processes or relations that might be relevant. We argue therefore that our model is robust for applying to Lake IJssel and to other inland reservoirs of several metres depth in a similar climatic setting.

**Table 3.** Regression analysis between observed open water evaporation and estimated evaporation using Penman's model, also broken down to the two individual terms of Penman's model, i.e. the aerodynamic and radiation term. Analysis is performed for the summer period in 2019 using hourly observations of Stavoren.

| **Penman** | *Regression model* | $R^2$ |
|---|---|---|
| Full Penman model | 0.08 Penman + 93.5 | 0.03 |
| Aerodynamic term | 4.3 Penman$_{aerodynamic}$ − 13 | 0.75 |
| Radiation term | 0.05 Penman$_{radiation}$ + 99.1 | 0.01 |

The dynamics of the observed diurnal cycle of $E_{water}$ agrees to what has been found in studies of for example Tanny et al. (2008), Venäläinen et al. (1999), Granger and Hedstrom (2011), Nordbo et al. (2011), and Potes et al. (2017). Additionally, the

estimated diurnal cycle by lake model FLake (Mironov, 2008) resembles well our observed $E_{\text{water}}$ (Jansen and Teuling, 2020). All of the studies above have found the occurrence of nighttime evaporation as well. This indicates that heat which has been stored during the day, is being released during the night when the lake temperature exceeds the air temperature. The fluxes LE and $H$ are a function of surface and air temperature and through the outgoing and incoming long-wave radiation $R_n$ is a

function of surface and air temperature as well. As a consequence of the energy balance, this means that $G$ is also a function of temperature. The large heat capacity of a water body, controlled by the depth of the water column, provides the system with a 'memory'. As a result, the water temperature at the surface is not directly related to the instantaneous energy balance at the surface, where net radiation is divided over the turbulent fluxes and a water heat flux. Rather, water temperature is subject to a delay and results from the heat storage that is integrated over a longer timescale. We argue that the effect of delay also leads

to the different drivers that have been found at hourly and daily timescales in Stavoren. The volume of a water body of several metres deep with large heat capacity and three dimensional heat transfer through mixing, results in a fundamentally different system compared to a shallow water surface or land surface with different factors that drive evaporation (McMahon et al., 2013).

In other studies, the dynamics of $E_{\text{water}}$ have been found to spatially vary over an inland water body caused by advection

and fetch distance from the upwind shore (Weisman and Brutsaert, 1973). More specifically, Granger and Hedstrom (2011) have found that $E_{\text{water}}$ is a function of the lake-land contrast of temperature and vapour pressure. Another source for spatially varying $E_{\text{water}}$ is the water surface temperature, which can be affected by the spatial variability of water depth (Wang et al., 2014), or for instance by the supply of water of a different temperature through rivers. Given that our measurements sites are located (i) at the shore in the north of the lake (Stavoren), and (ii) on the dike in de middle of the lake (Trintelhaven), could

therefore potentially lead to differences in observed $E_{\text{water}}$ dynamics between the two measurement sites. The coefficients of the hourly regression models were found to be significantly different between the two locations (Sect. 3.4.1). This difference might be attributed to the difference in location (i.e. at the shore and in the middle of the lake, respectively). Other reasons might be the difference in measurement height or the inherently different meteorological conditions we measure, because the two measurement sites are located on opposite sides of the lake.

Not all the components of the energy balance could be measured during our field campaign. Therefore, the closure of the energy balance, which can be calculated as the ratio between the turbulent fluxes and available energy, could not be analysed. Other studies that were able to assess the energy balance closure (EBC) over lakes and/or reservoirs have found imbalances of the energy balance that were within a narrow range, and similar to those over land (Wilson et al., 2002). McGloin et al. (2014) found an average EBC value of 76% over a year, and found little variation over the seasons with a value of 77% for

the summer season. Similar values of 82% and 72% for the summer seasons of 2006 and 2007, respectively, were found by Nordbo et al. (2011). A reasonable EBC of 91% was found by Tanny et al. (2008), although this was for a short period of 14 days. The measured imbalance suggests a general underestimation of the turbulent fluxes. Factors that could contribute to this imbalance are: large-scale transport (advection) of heat and water vapour, a systematic instrument bias, mismatch between the frequency of sampling and the turbulent eddies, mismatch of the measurement footprint of the individual terms, and neglected

energy sources or sinks (Wilson et al., 2002; Foken, 2008; Mauder et al., 2013, 2020). Despite this likely underestimation of

observed $E_{\text{water}}$ following the imbalance of the energy budget, we believe that this bias will not influence the dynamics of $E_{\text{water}}$ and the correlations found with other meteorological variables.

## 5 Conclusions

In this study, we investigated the dynamics and drivers of open water evaporation of Lake IJssel in the Netherlands through the development of a data-driven model. To this end, open water evaporation was measured during two summer periods at two locations using eddy covariance instruments. Based on the results of regression analysis, it was found that at hourly timescales wind speed and the vertical vapour pressure gradient are the main drivers of open water evaporation during the observed summer periods of 2019 and 2020. These variables are the same as used in Dalton's model that is often used for estimating evaporation from deep water bodies. Using the data collected in 2019 simple data-driven models for both locations were developed. At the hourly timescales this resulted in $R^2 = 0.74$ and $R^2 = 0.70$ for Stavoren and Trintelhaven, respectively. Validation of these hourly simple models using the data collected during the summer of 2020 have shown that a simple data-driven model is able to explain large parts of the hourly dynamics of open water evaporation ($R^2 = 0.84$ and $R^2 = 0.67$ for Stavoren and Trintelhaven, respectively). The absent correlation between observed daily open water evaporation and estimated evaporation using Makkink's model has shown that this radiation-based model is unable to explain the dynamics of $E_{\text{water}}$, although this is current practice in the operational water management of Lake IJssel. Given the importance of $E_{\text{water}}$ in the large-scale water balance it is necessary that this process is incorporated correctly in hydrological models.

*Code and data availability.* The evaporation datasets of Stavoren and Trintelhaven are available on the 4TU data repository (doi: 10.4121/16601675). The code and accompanying data of the regression analysis is available on the 4TU data repository as well (doi: 10.4121/16913308). The KNMI datasets are available at https://www.knmi.nl/nederland-nu/klimatologie/uurgegevens (last access: 20 August 2021) (KNMI, 2021b). The water temperature datasets of Rijkswaterstaat are available at https://waterinfo.rws.nl/#!/kaart/watertemperatuur/ (last access: 20 August 2021) (Rijkswaterstaat, 2021). The sub-skin Sea Surface Temperature product of the Meteosat-11 satellite is available at https://osi-saf.eumetsat.int/products/sea-surface-temperature-products (last access: 20 August 2021) (EUMETSAT, 2021).

*Author contributions.* FAJ designed and carried out the study and field work under supervision of AJT, RU and CMJJ. All authors contributed to the interpretation of the results. FAJ wrote the manuscript and AJT, RU and CMJJ provided their feedback on the manuscript.

*Competing interests.* The authors declare that they have no conflict of interest.

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

## Appendix A

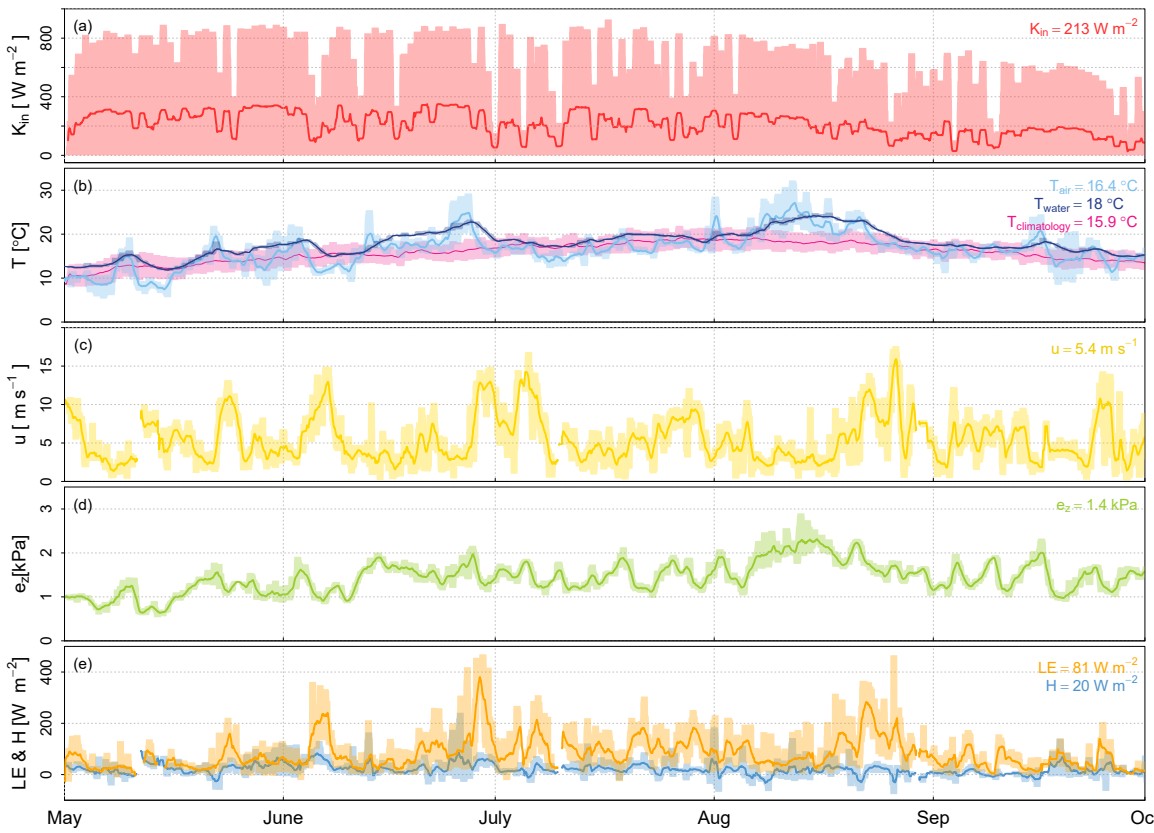

**Figure A1.** Meteorological conditions in Stavoren in 2020 showing running daily means of global radiation (**a**), air temperature (current and climatology) and water temperature (**b**), wind speed (**c**), vapour pressure (**d**), and turbulent fluxes (**e**). The shaded area represents the range between minimum and maximum observed values and the numbers reported at the top right of each panel provide the average values of the respective variables during the presented months.

# Appendix B

**Table B1.** Basic statistics of the latent heat flux during the summer period at both locations (i.e. Stavoren and Trintelhaven) at the hourly and daily timescale. The statistics given are mean, standard deviation, minimum, maximum, 25th and 75th quantile, and the number of observation points.

| (a) **Stavoren – LE [W m$^{-2}$]** | Mean | Stdev | min | max | Q25 | Q75 | _N_ |
|---|---|---|---|---|---|---|---|
| Hourly | | | | | | | |
| 2019 | 104.1 | 69.2 | -21.5 | 516.1 | 56.6 | 133.1 | 896 |
| 2020 | 128.4 | 87.6 | -27.7 | 444.6 | 66.1 | 146.5 | 687 |
| Daily | | | | | | | |
| 2019 | 130.7 | 67.3 | 51.2 | 310.1 | 84.3 | 147.5 | 20 |
| 2020 | 168.4 | 85.3 | 55.7 | 364.5 | 111.3 | 215.0 | 18 |

| (b) **Trintelhaven – LE [W m$^{-2}$]** | Mean | Stdev | min | max | Q25 | Q75 | _N_ |
|---|---|---|---|---|---|---|---|
| Hourly | | | | | | | |
| 2019 | 95.9 | 59.2 | 11.0 | 333.9 | 47.4 | 130.3 | 454 |
| 2020 | 91.6 | 60.2 | -34.8 | 351.8 | 52.1 | 113.2 | 663 |
| Daily | | | | | | | |
| 2019 | 122.3 | 45.3 | 58.9 | 210.9 | 94.8 | 147.3 | 10 |
| 2020 | 100.5 | 60.1 | 35.1 | 273.8 | 81.9 | 105.4 | 13 |

# Appendix C

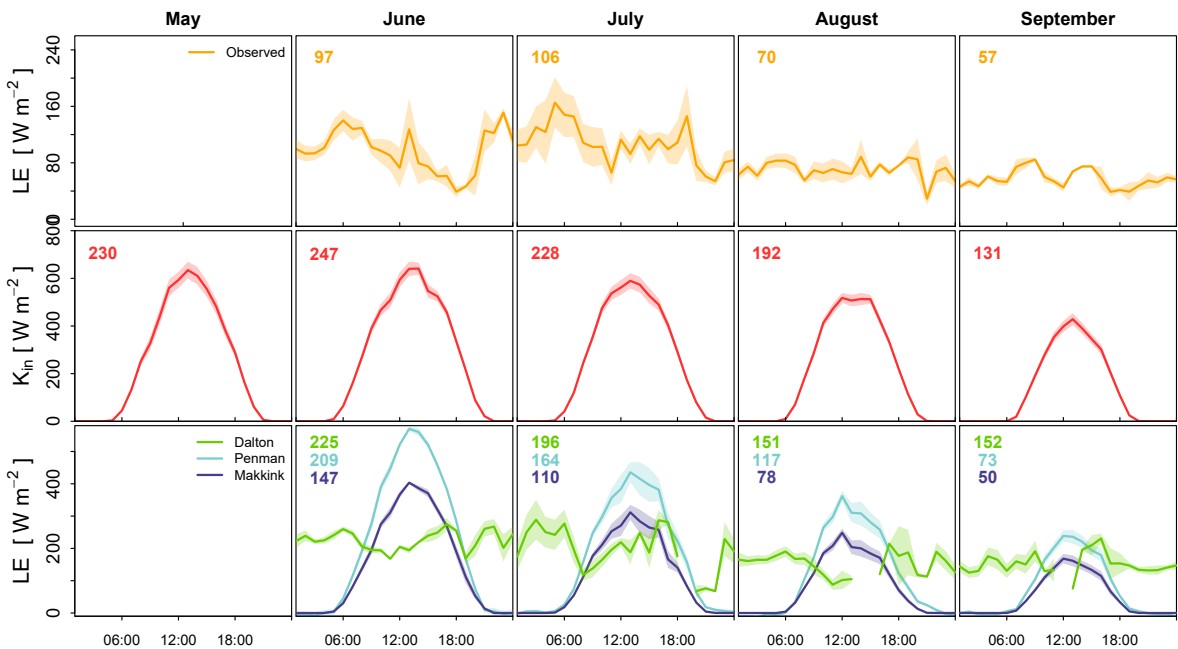

**Figure C1.** Illustration of the decoupling at Trintelhaven in 2019 between monthly average diurnal cycles of observed latent heat flux (top) and global radiation (middle), the latter forming the basis of the frequently used evaporation models of Penman (1948) and Makkink (1957). These models are shown together with the model of Dalton (1802) at the bottom panels. Note that some variables included in the evaporation models are measured at larger heights than the 2 m that are prescribed (see Eq. 6 – 12). Additionally, all three models are generally used on a daily basis, but they are presented here to show the underlying daily cycle. The shaded area represents the uncertainty, which is defined as the standard deviation divided by the square root of the number of observations. Average daily means of the respective variables are indicated by the number at the top left of each panel displaying the average course over the summer period.

# Appendix D

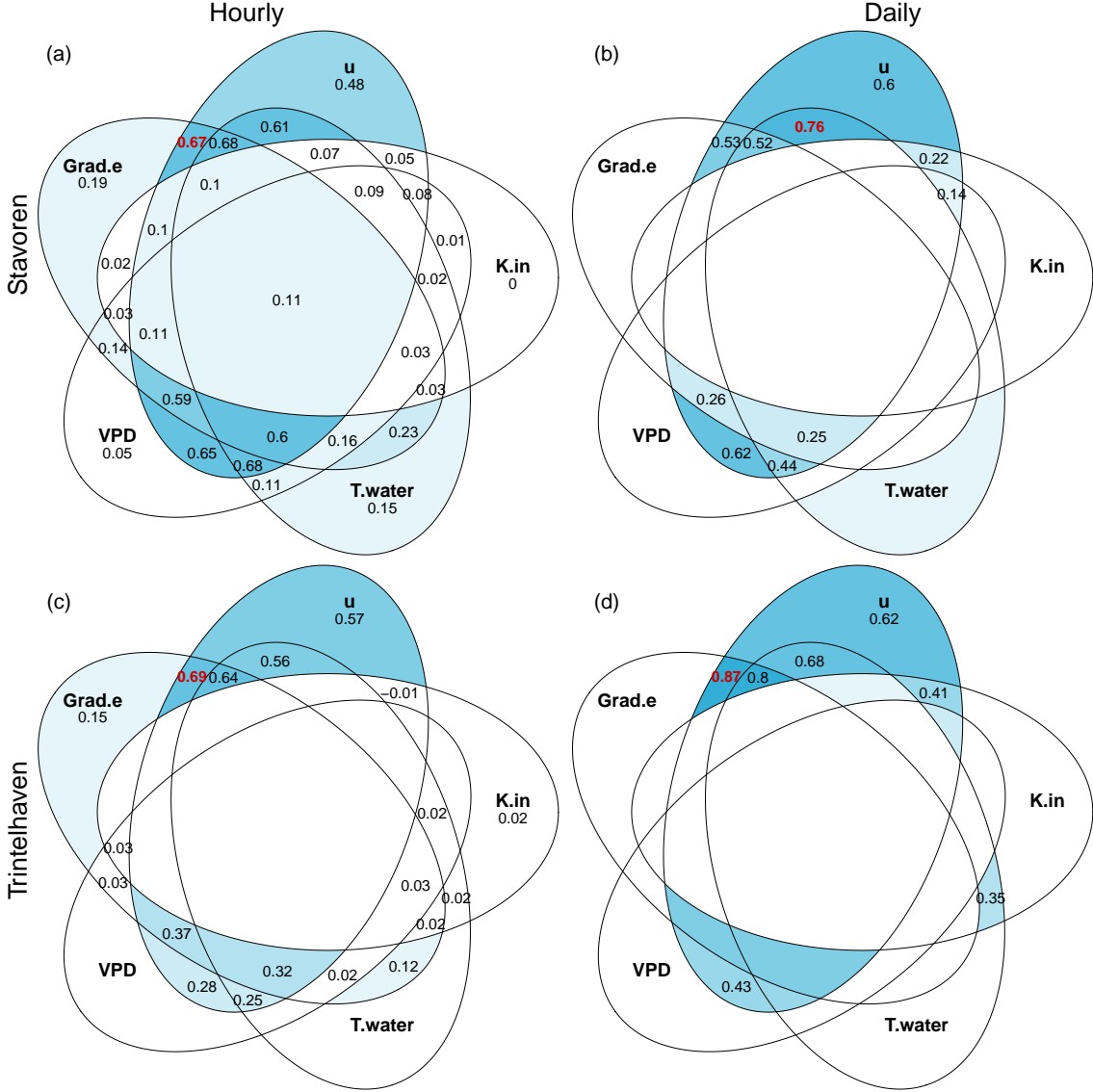

**Figure D1.** Systematic exploration which variable or combination of variables (product) can best explain the dynamics of open water evaporation. The outer 'leaves' of the Venn diagram represent the single variables, while moving towards the centre of the diagram the combination of products of variables are represented. Within each leaf the adjusted $R^2$ value is depicted. The higher this value, the more blue the colour of the leaf. The red number indicates the highest $R^2$ value indicating the best combination found for a maximum of two variables, i.e the best 'simple' model. Values of $R^2$ were removed if the model fit was found to be insignificant ($p < 0.05$). The analysis is based on data from the summer of 2019 and is performed at hourly timescale (left panels: Stavoren (**a**), Trintelhaven (**c**)) and daily timescale (right panels: Stavoren (**b**), Trintelhaven (**d**)).

## Appendix E

**Table E1.** ANOVA analysis indicating the significant difference (p < 0.05) of the hourly model coefficients found between the two measurement sites during the summer of 2019.

Analysis of Variance Table (ANOVA)

Model 1: LE ~ u$\Delta$e + (u$\Delta$e)$^2$
Model 2: LE ~ u$\Delta$e + (u$\Delta$e)$^2$ * Measurement site

| Model | Res.Df | RSS | Df | Sum of Sq | F | Pr(>F) |
|---|---|---|---|---|---|---|
| 1 | 559 | 664256 | | | | |
| 2 | 556 | 652804 | 3 | 11452 | 3.2512 | 0.02151* |

Signif. codes: 0 '****'  0.001 '**'  0.05 '.'  0.1 ' '  1