# Peer review of "Evaporation from a large lowland reservoir – observed dynamics and drivers during a warm summer"

_Hydrology and Earth System Sciences, 2021_

## Referee Comment (RC2)

**Review for « Evaporation from a large lowland reservoir – observed dynamics during a warm summer » by Jansen et al.**

**Paper summary :**

This paper presents a data-driven analysis of evaporation from Lake Ijssel (The Netherlands) over the summer period 2019 and 2020. The overall objective is, first, to determine the key variables driving the evaporation of the lake and, finally, to propose a new parametrisation for this specific water evaporation.
The study is interesting in its approach and aims to fill a gap in the short-term monitoring of the water resource. Based on *in situ* measurements estimated by the accurate and commonly used eddy covariance technique, a simple statistical model is developed and trained to accurately retrieve the summer evaporation of the lake on hourly and daily scales.

**General comments :**

The manuscript is well written and easy to read, although some sections need to be reworded throughout the paper to improve readability and highlight the scientific contribution of this study.
My main concern is with the overall presentation of the article. I found that the innovative aspect of this study was hidden by general facts and conclusions that have already been proven in the past. I understand the importance of such conclusions in justifying a new parametrisation but, in my opinion, focusing on this key point weakens the overall quality of the article. I suggest that the authors improve the paper to focus on the importance of using the developed parametrisation and to emphasise the need for such a statistical model. In this spirit, I recommend to better integrate this work into the current scientific literature.

Furthermore, some of the results discussed are not presented in detail and it is therefore impossible to review this information. You should either delete these results or provide the details.

All the comments do not call into question the study itself. The material for a good article is already there and just needs some rearrangement and minor revisions. I am convinced that the article will gain in precision and interest with additional information.

Specific comments are detailed below.

**Abstract**

General comments : In this section, the objective of the paper is not clearly pointed out. In my opinion, you should focus on the specific parametrisation you proposed for Lake Ijssel based on the field measurements. All the elements are already written and you just need to rearrange the section.

-P1.L1 : I would talk about a « sink » rather than a « large loss term ». It is more adequate to the scientific level of the journal.

-P1.L1 : « During summer seasons, which are projected to become warmer with more severe and prolonged periods of drought ».

This is a general sentence whereas your study focused on a specific location. Even if we are on a global climate change path, the consequences (not specifically warmer summer) are not the same worldwide. You should be more specific on the spectrum of warming on the studied area (or region) and put references.

- P1.L8 : « not available energy»
Be specific on the type of energy.

-P1.L11 : « main drivers »
Be specific. What type of phenomenon they are the drivers of?

- P1.L15 : « well performing simple data-driven models »
I would be less enthusiastic with a $R^2$ of 0.51 and 0.43. The model is adequate but does not perform well.

**Introduction**

- P2.L1 : There is a more up-to-date review paper you should include:
Woolway, R. I., Kraemer, B. M., Lenters, J. D., Merchant, C. J., O'Reilly, C. M., & Sharma, S. (2020). Global lake responses to climate change. *Nature Reviews Earth & Environment*, *1*(8), 388-403.

I would suggest to have a look at the paragraph about lake evaporation which gives essential materials for both your introduction and your discussion.

- P2.L2 : « evaporation is a large loss term of water bodies ... »
I would rephrase by saying this is « a sink in the lake water balance ». Also you should add a reference to justify this, even if it's a validated fact.

- P2.L7 : « Summers are projected to become warmer »
As mentioned for the abstract section, as you work on a specific location, you should be more specific on such fact as you are not working at global scale (mention the spatial scale). Moreover, I would recommend to give the climate reference on which the climatic comparison is made. You should also add a reference.

- P2.L12-13 : « In terms of thermodynamics,  ...»
This works on some lakes but this is not always correct. In terms of thermodynamics, big lakes (such as the American or African Great Lakes) could be approach by using either 3D ocean model or 1D model. It will depends on the presence or not of the hypolimnion and furthermore on the stratification dynamic, if there is one.

See :
Xue, P., Pal, J. S., Ye, X., Lenters, J. D., Huang, C., & Chu, P. Y. (2017). Improving the simulation of large lakes in regional climate modeling: Two-way lake–atmosphere coupling with a 3D hydrodynamic model of the Great Lakes. *Journal of Climate*, *30*(5), 1605-1627.

Gronewold, A. D., & Stow, C. A. (2014). Water loss from the Great Lakes. *Science*, *343*(6175), 1084-1085.

Thiery, W. I. M., et al. "LakeMIP Kivu: evaluating the representation of a large, deep tropical lake by a set of one-dimensional lake models." *Tellus A: Dynamic Meteorology and Oceanography* 66.1 (2014): 21390.

- P2.L17 :
This might be a detail but I would prefer to talk about a change of the amplitude (which suppose an increase of surface temperature during daytime but also a quicker decrease during night-time).

- P2.L21: Lake depth also controls the dynamical range of lake temperature amplitudes on diurnal timescale.

- P3.L2-6 : This paragraph would gain in readability if you reduce the description to its essential. Penman equation is well-known and its description can be shortened. Moreover, this description is redundant with the one on P9.

- P3.L7 : « Most studies ... »
Is this sentence linked with the reference list starting on L9 ? If so, you should movethe sentences « However, measurements of... » and « This can partly ... » elsewhere. Also, you said the contrary on L.26 "In the past, a number of studies reported ..."
Moreover, I would not be that direct by saying measurements of evaporation from inland water bodies are under-represented, numerous studies haven been published on the subject for the past 10 years:

Potes, M., Salgado, R., Costa, M. J., Morais, M., Bortoli, D., Kostadinov, I., & Mammarella, I. (2017). Lake–atmosphere interactions at Alqueva reservoir: a case study in the summer of 2014. *Tellus A: Dynamic Meteorology and Oceanography*, *69*(1), 1272787.

Pillco Zolá, R., Bengtsson, L., Berndtsson, R., Martí-Cardona, B., Satgé, F., Timouk, F., ... & Pasapera, J. (2019). Modelling Lake Titicaca's daily and monthly evaporation. *Hydrology and Earth System Sciences*, *23*(2), 657-668.

Moigne, P. L., Legain, D., Lagarde, F., Potes, M., Tzanos, D., Moulin, E. R. I. C., ... & Costa, M. J. (2013). Evaluation of the lake model FLake over a coastal lagoon during the THAUMEX field campaign. *Tellus A: Dynamic Meteorology and Oceanography*, *65*(1), 20951.

Blanken, P. D., Spence, C., Hedstrom, N., and Lenters, J. D.: Evaporation from Lake Superior: 1. Physical Controls and Processes, Journal  of Great Lakes Research, 37, 707–716, https://doi.org/10.1016/j.jglr.2011.08.009, 2011

- P3.L27 : I'm not convinced about the utility of the brackets. Moreover you could also include other important hydroclimate variables.

Zhou, W., Wang, L., Li, D., & Leung, L. R. (2021). Spatial pattern of lake evaporation increases under global warming linked to regional hydroclimate change. *Communications Earth & Environment*, *2*(1), 1-10.

- P3.L34 →P4.L2:
Woolway et al 2018 & Wang et al 2018 have addressed this issue. Moreover, even if I agree with your assumption, I'm not convinced about such parametrisation for use at longer timescale (for example, seasonal timescale). Lake temperature and evaporation are inter-dependent on such timescales and other hydroclimate variables should also be included.

Woolway, R. Iestyn, et al. "Geographic and temporal variations in turbulent heat loss from lakes: A global analysis across 45 lakes." *Limnology and Oceanography* 63.6 (2018): 2436-2449.

Wang, Wei, et al. "Global lake evaporation accelerated by changes in surface energy allocation in a warmer climate." *Nature Geoscience* 11.6 (2018): 410-414.

- P4.L3 : This sentence is the key point of your study. More generally, the paragraph from L3 to L14 should be the core of your introduction. I would reduce the presentation of the different equation (Penman, Makkink) and enrich this paragraph.
Add a reference for this : «  a crucial element in its water management system ». Also, you focus on the water management aspect however your paper does not specifically study the impact of the parametrisation on the lake hydrology. I would recommend to add other aspects of the evaporation as a component of the global energy and water cycle. For example, you can talk about the influence on the near-surface turbulence intensity, the stratification or the lake ecosystem.

Raymond, P. A., and others. 2013. Global carbon dioxideemissions from inland waters. Nature$503$:355–359. doi:10.1038/nature12760

Jenny, Jean-Philippe, et al. "Scientists' warning to humanity: rapid degradation of the world's large lakes." *Journal of Great Lakes Research* 46.4 (2020): 686-702.

- P4.L12-15 : In this sentence, you compare Makking's equation with Flake simulations. I'm wondering why you are not using FLake directly for lake IJssel if you consider Flake simulations as your reference ?

**Data, Material and Methods**

- P6.L8: It seems that the KNMI station only measure global radiation (as I see in the data provided), however would it be possible to have access to the four components of the global radiation? As shown in Wang et al 2018, the incoming radiation has an effect even if it's at longer timescale.

Wang, Wei, et al. "Global lake evaporation accelerated by changes in surface energy allocation in a warmer climate." *Nature Geoscience* 11.6 (2018): 410-414.

- P6.L30: Could you please rephrase this sentence to improve readability.

- P7.L30-32: "A regression analysis …" + "To develop ..."
Please rephrase to improve readability.

- P8.L11:
I found difficult to understand the justification of using such regression model and how the hypothesis of such model have been tested.
What are the type of estimator you used (I assume an ordinary least square estimator)? Did you perform a significativity test? It would be interesting to look at the result of the multiple linear regression model and specifically the p_value to include or exclude predictors.
Are the period chosen representative of the population?

- P8.L22: "surface temperature"
Are you talking about the Meteosat product? Hence, why do not use directly these field data? Are there representative of the surface temperature (the lake is shallow then it would be important to be sure the measurements are not performed in the thermocline).

**Results**

General comments: In this section, I would have a distinct paragraph presenting the results of the calibration, another for the validation and a final one for the result on the routinely measured variables. It would also improve the readability as I had hard time following this section.

- P11.L5; Fig2:
How did you choose the time period presented in the figure? Why do you present this period instead of either the training or validation time period? As I understand your paper is about summer and here you present a part of the autumn season. I admit I was a bit lost. Also I would suggest to be more specific on the time period (e.g: 01/05-31/08 instead of nouns (You write May-August most of the time and once May-September).

- Section 3.2:
In this section, you compare meteorological conditions. It seems you are comparing the air temperature and the wind speed that are are measured at different height. Measurements at Stavoren are made around 7m and around 10m at Trintelhaven. Did you adjust your measurement to an equivalent height? If not, this could explain some of the discrepancies. I have the same question for EC data.
Moreover, you compared these variables to the Dalton model which needs variables at 2m height. What was the procedure you used to adjust the measurements to this height?

- P10.L20: "the water temperature at" instead of "the water at".

- P10.L18: Please provide the correlation score to justify this is a strong correlation.

- P11. Fig2: Please put the graph corresponding to the 2020 summer period in Appendix.

- P12.L1: I would use the word "pattern" instead of "rhythm".

- P12.L5: Could you please add the graph in the Appendix?

- P13.L17: Could you please add the correlation score to justify if it's a strong correlation. Also, be careful not to mismatch between correlation and determination when you analyse your results and even more as you are studying a non-linear model. Please rephrase the sentence "Global radiation …" to account for this difference.

- P14.L4-6: I understand that you only exclude the global radiation from your model based on the $R^2$, however, in my opinion a $R^2(VPD)=0.05$ questioned the inclusion of VPD in your model. For example, adding VPD on the Stavoren hourly analysis as a limited impact which is not significant in Trintelhaven.
Does VPD has a significant impact on your score?

- P14.L22: Would it be possible to have some basics statistics on these data (mean, standard deviation, quantiles, min, max). It would also help to see if outliers are ejected from the analysis. A Table placed in the Appendix would be sufficient and would give a hint about the discrepancies between both summer seasons.

- P14.L15: $R^2$ explains 45% of the variance which is quite low. If you include the water temperature, it reaches 0.48, it is still low but better.
You limit the maximal number of variables for the simple model but in this case it would benefit to your model to add the water temperature.

- P14.L22: "This can be attributed …"
As noticed in Woolway et al 2021, lake evaporation is highly dependent on weather variability (through its dependence to the lake surface temperature). Your discussion need to stress this issue and not just focus on the comparison to a mean climate.
I would remove this sentence and discuss about this point in the adequate section.

- P14.L26: "this confirms …"
Rephrase this sentence. If the ingredients are the same than in the Dalton's model why do not use this model or use a calibrated version on your lake?

- P14.L28: "To determine if the coefficients …"
Without the results of the analysis it is impossible to assess the results. Either give the results or erase this sentence.

- P16.L5-9: "The results for the location of …"
This is a good analysis of your results. However be sure to be consistent. In your abstract you say that the model performs well.

**Discussions**

You need to improve your discussions and criticise your result in a more precise way. You could be more exhaustive and include limitations (e.g decomposition of the radiation term, looking at the influence of other hydroclimate variables on the variance).

- P17.L7: You should be more precise and discuss the fact that you do not analyse each term of the radiation budget.

- P18.L6: you can add Le Moigne et al 2016 as a reference.

- P19.L10-13: It is impossible to review this part of the discussion as you don't provide the results. You should either erase the sentence or give the correlation plots.

- As mentioned in the precedent comment (for P14.L22), you are working at short timescale and thus, the lake evaporation is dependant on the weather and the hydrological variability. Your discussion would be more complete by discussing these points.

**Conclusions**

- P21.L19: In my opinion, your main contribution is the development of the statistical model. I would suggest to rephrase your conclusion in order to account for this.

- P21.L20: Ok but this a general fact and this is not your main result.

- P21.L26-28: Rephrase the sentences to be more precise on the result you use (if it's hourly or daily timescale). It is hard time following which are the $R^2$ you are presenting.

**Editorial comments**

- Some sentences lack of consistency and readability. This is often the missing punctuation that is in cause. For example, look at P2.L12, P7.L17, P7.L24, P9.L24, P12.L3.

- Be attentive to have consistency in the form you write the units. The general form is to separate units with a point, ex : $m.s^{-1}$.

- « Focussing » should be written like this « focusing ».

- P2.L8 : parametrise/parametrize and not « parameterise ».

- P2.L26 : add a « to » → « and to represent ».

- P2.L33 and P7.L20: check the tense.

- P4.L25 : use the English structure : 1,100 km²

-P5. Figure1 :

Increase police size of your scale as it is not readable. Moreover, you should add the label on the contour lines. On the right figure you can erase the y-axis as it is the same than the center figure. Green and white colors for the center and right panels are not adapted to understand where are the land and the water.
Add labels to the windrose.

- P7.L8 : you should either use co-variance or covariance but not both throughout you paper.

- P8.L9 : « variable(s) » instead of « variable(x) ».

- P10 Table 1 : Be sure all the parameters are aligned in the first column.

- P12.L11: "are lacking"

- Please consider improving your Venn diagram in order to gain in readability (police size).

---

## Referee Comment (RC3)

**Title: Evaporation from a large lowland reservoir – observed dynamics during a warm summer**

The manuscript analyses evaporation measurements by EC towers and provides simple regression models based on the routine observations to describe evaporation dynamics in Lake IJsell (Netherlands). It is well written and organized. However, there are some aspects in the study that need clarification or should be addressed by the authors.

- There are many studies showing that evaporation dynamics vary between different parts of inland water bodies arising from, for example, spatial variability of water depth or meteorological inputs (boundary conditions). Considering this, it is fine to have two different measurement stations at the middle of the water body (Trintelhaven) and at the border (Stavoren). However, a discussion on the effect of EC towers location on the observed evaporation dynamics from the water body is missing (not its comparison with terrestrial evaporation as mentioned in section 2.2).

- Vapor pressure at the air-water interface was estimated based on the surface temperature obtained from satellite imagery that often show biases. Was this checked?

- From our own measurements of vertical water temperature in a shallow basin with 2 m depth, I can say water temperature at the surface, where evaporation takes place, is completely different with even 10 cm below. Thus I am not sure how water temperature at depth of 1.2 or 1.5 m could help for evaporation analysis, unless you have a temperature model to reproduce surface temperature.

- It is not surprising that neglecting thermal inertia of the water body (indicating the effect of radiation adsorption in depth) could make such considerable bias in the performance of models such as Penman's (see Friedrich et al. 2018: DOI:10.1175/BAMS-D-15-00224.1). Please see Section 2.4 of Zhao et al. 2020 (https://doi.org/10.1016/j.rse.2020.112104) accounting for the impact of $G$ on evaporation estimates by Penman-type approaches.

- Why air temperature is not included in the analysis of section 3.4?

- I believe radiation is a key component in shaping surface temperature that, in turn, defines vapor pressure gradient at the core of your simple regression model. In application of Eq. 10, how surface temperature is obtained? From measurements at depth of 1.5 m, satellite imagery, or solving for energy equation? Dalton-type models may look simple in representation but have difficulties associated with obtaining reliable surface temperatures, and of course wind function (especially in the context of the projected climate change).

---

## Author Comment (AC1)

**Response to referee comment Anonymous Referee #1**

We appreciate and would like to thank Anonymous Referee #1 for reading our manuscript thoroughly and for raising valuable points of feedback. This will be helpful to improve the manuscript and was used to identify and correct a mistake that we have found in one of the scripts. Please, find below our point-to-point response (comment of the referee in **black**, our response in **blue**).

Review of Jansen et al., Evaporation from a large lowland reservoir—Observed dynamics during a warm summer.

This manuscript is well-organized and generally well-written. The topic is of interest to HESSD readers. The gist of the study is to use eddy covariance sensible and latent heat fluxes from two towers located on a large shallow lake in the Netherlands, to determine which underlying variables drive shallow lake evaporation.

The abstract describes the work pretty well. One thing that caught my attention was the assertion that the Penman equation disagrees with the study results. After reading the rest of the paper, I think the authors are saying that the net radiation Rn and the ground heat flux G terms of Penman are extremely difficult to interpret in the context of a shallow lake. Deriving Rn-G directly from global radiation does not account for the heat capacity within the water column.

Thank you for raising this point. Please find our response to this general point at point 3 of the specific comments.

The experiment is well-designed. The flux footprint analysis is helpful in determining which eddy covariance flux data correspond entirely to lake evaporation and which time periods are contaminated with land fluxes as well. The figures are well-designed. I found Figure 5 (the Venn diagrams) especially informative.

Thank you for this positive feedback on these points!

Specific comments.

1. Page 6 line 18 says that the data were collected during the summer of 2019 and 2020. Section 3.1 describes the large fraction of time that lake fluxes were unavailable or of limited quality. My impression is that these statistics refer to 30-minute averaging periods. If I understand this correctly, it is not clear how daily, monthly, and yearly data were obtained from these data. How were gaps filled? In particular, how would long periods of time with flux footprints falling outside the lake be handled? While the paper is generally clearly written, more attention is needed to specify the time-scales for the various figures, tables, and in-text statistics. Finally on this point,

   We are sincerely thankful for your remark on how we have dealt with the low number of remaining data after filtering (on quality, wind direction). Your remark actually made us go back to the scripts, where we found a small mistake in one of the programming commands which let all daily averages be calculated even if just one hour would be present. This of course was not our intention. We now correctly calculate daily averages only if for at least 66% of the hours (two-third) in a day valid data are available. Of course this leads to less daily

data being available and therefore we had to redo some of the analyses, in particular the regression analyses. At a daily timescale the regression model is based on only a few data points (in the order of 12), but it is still found to be statistically significant (p-values < 0.01). The conclusion that in most cases a combination of wind speed and vertical gradient of vapour pressure forms the driving force of LE did not change. Finding the mistake thanks to the comment of the reviewer shows us that the review process is really helpful and needed. A good reminder and learning moment to me and us.

We intentionally did not filled large gaps (>1 hour) in the data. This would create synthetic results interfering with our aim to perform a process-oriented study into the role of evaporation in the surface-atmosphere coupling. We only performed linear interpolation to the data when gaps of at the most one hour occurred. We will add the following section to the manuscript at P7.L14: *"Hourly data was obtained by aggregating the half-hourly flux dataset, where at least one value per hour should be available. Within this hourly dataset large gaps (>1 hour) were intentionally not gap filled, because this would create synthetic results interfering with our aim to perform a process-oriented study. Only linear interpolation was performed to the data when gaps of at the most one hour occurred. Daily averages were calculated from this hourly dataset only if for at least 66% of the hours (two-third) in a day valid data are available."*

In the regression analysis we have only used hourly and daily data. Figures 4 and 9 show monthly averages based on hourly data. The uncertainty bars provide an indication of the ratio between the standard deviation of the hourly observations and the square root of the number of hourly observations taken into account in the calculation of the monthly mean, as a measure of the sampling uncertainty of the estimated mean.

2.  When I read that Penman's equation does not work for these data, I was surprised. I thought the authors were claiming that they physics behind the equation were incorrect. But I think the authors agree with the physics of the Penman equation. They simply cannot determine the available energy with any reasonable certainty for this water body. Do I understand this correctly? Read our response at point 3.

3.  Maybe this is just a re-wording of the previous comment, but the energy input, heating/cooling of the lake air stability, etc. do determine the fluxes. The temperatures and humidities at various heights and depths adjust according the these principles. The point of the paper is to determine which easily-measured variables give the best access to the fluxes.

    Thank you for your remarks on the disagreement of our results in relation to the Penman equation. In response to your general and specific comments that target the use of Penman's equation we would like to argue the following:
    Lake IJssel is a shallow lake of several metres deep and therefore most probably the heat capacity could indeed not be neglected. Penman developed his theory originally for land surfaces and (infinitely) shallow water surfaces. Therefore, he could make the assumption that the net radiation was divided over the turbulent fluxes and the ground heat flux exactly at the surface-atmosphere interface. This then determines the surface temperature and surface humidity, and eliminates surface temperature from the equation (P3.L2-3). However, in a water body part of the radiation penetrates the water, delivering relatively large amounts of energy below the air-water interface directly. This affects the physical processes

in a way that cannot be described with the PM equation. The large heat capacity of a water body provides the system with a "memory". As a result, the water temperature at the surface is not directly related to the instantaneous energy balance at the surface, which is how Penman's equation can be interpreted, but rather it is subject to a delay following the large heat capacity of the water body (P20+P21, L10-11+L1). Using a simple energy balance model of a water layer could help to solve the water temperature for the next time step (Equation 11 from Keijman (1974), and eq. 10 from De Bruin (1981)). However, in our analysis we have not measured $G$ based on temperature changes integrated over the volume of the water column. In other words, we have omitted the downward heat flux $G$ in the calculation of $Q^*$ to adhere to the original Penman theory (P9, L 17-19). All the results of the Penman model presented in this manuscript are therefore based on the original theory of Penman for land surfaces and (infinitely) shallow water surfaces where $G$ is neglected. We think that the Penman model does not present the full story in case of a water body of several meters deep.

We will adjust P20.L10-P21.L1 as follows: "*The large heat capacity of a water body provides the system with a "memory". As a result, the water temperature at the surface is not directly related to the instantaneous energy balance at the surface, which is how Penman's equation can be interpreted, but rather it is subject to a delay following the large heat capacity of the water body. We think that the Penman model does not present the full story in case of a water body of several meters deep.*"

4. In figure 5 in particular, are the diagrams only for the summer months?

   Thank you for this remark. Yes, this diagram is based on only the summer months. In the revised version of the manuscript this will be indicated in the caption of this figure, as well as for other figures where this applies.

5. Page 16 lines 4-6. Typically, if measurements have a restricted range of variability, this results in a smaller R2 value, because random fluctuations are large relative to the observed changes. It looks like the authors are claiming the opposite effect here. Please explain.

   R2 values are determined by the variances of both data series, and the covariance between those series. So we agree that our explanation: 'higher variance will lead to lower R2 values', is not covering the whole story. We agree with the reviewer that indeed measurements with a restricted range of variability would result in smaller R2 values given the measurement uncertainties, which are not correlated.

   In an attempt to understand why sometimes the R2 values are higher for the validation period and to check if we are dealing with a 'bad' model, we have merged all the data of both summers (2019 and 2020), and subsequently we have split the data randomly in two datasets (each 50% of the data to simulate two summers). The coefficients for the regression model, that has the same form as we originally used, were re-calculated. We now find that in case of hourly data for Stavoren the calibration results in R2=0.8, and validation using the test data results in R2=0.72. Similarly if we study the hourly routine data for Trintelhaven, we find $R2_{cal}$ = 0.48, and $R2_{val}$ = 0.35. This gives us confidence that the original model we have created is not 'bad'.

6.  In Figure 5 d, the central intersection has an R2 value smaller than some of the other intersections. How could adding a variable explain LESS variability than simply not including it?

    This is because the numbers that are depicted in the Venn diagrams indicate the adjusted R2 to study the fit of the model as is mentioned in the caption of the figure. The adjusted R2 takes into account the degrees of freedom and can therefore lead to a decrease in adjusted R2 if the added variable only slightly correlates with the dependent variable.

    We will add the following to P13.L11: "... *the colour is. Adding a variable will not always result in a higher adjusted R2 value, because the adjusted R2 takes into account the degrees of freedom. Therefore, it can lead to a decrease in adjusted R2 if the added variable only slightly correlates with the dependent variable. Venn diagrams (a) and (c) ....."*

7.  Throughout the paper I kept wondering why lake water stability was not included. Surely there must be seasonal changes in stability and thus of mixing depth within the lake. Did this have an impact on the data?

    We agree that lake water stability would be interesting to consider since it affects the surface temperature and therefore evaporation rates through the vapour pressure gradient. Evaporation in turn has a cooling effect on the surface temperature (which increases potential mixing). However, we unfortunately did not have the opportunity to measure water temperatures at several depths to study this. A preliminary study performed by one of our master students simulated mixing depths using the model FLake, which showed that in 70% of the time Lake IJssel is fully mixed. During summer it is of course more likely that stable conditions occur and we cannot directly assume fully mixed conditions. However, we considered this phenomenon beyond the scope of the current study.

    We suggest to add the following to P19.L7: "*Another phenomenon that could affect the yearly cycle of evaporation is lake water stability and thus mixing depth within the lake. Seasonal changes in lake water stability affects the surface temperature and therefore evaporation rates through the vapour pressure gradient. Evaporation in turn has a cooling effect on the surface temperature, which increases potential mixing. Supported by a preliminary study where mixing depths were simulated using the model FLake (Voskamp, 2018), we assume that during 70% of the time Lake IJssel is fully mixed. This number is not surprising given the fact that during night time evaporation continues, and with wind speeds that are on average 5.8 m.s$^{-1}$. In addition, the inflow of the river IJssel into the lake is likely to support mixing as well. During summer it is more likely that stable conditions occur and we cannot directly assume fully mixed conditions. However, we considered this phenomenon beyond the scope of the current study."*

    Voskamp, T.: The evaporation of Lake IJssel - Comparison of the FLake model with standard methods at multiple timescales for estimating evaporation rates, MSc thesis report, Wageningen University, Wageningen, the Netherlands, 2018.

---

## Author Comment (AC2)

**Response to referee comment Anonymous Referee #2**
We appreciate and would like to thank Anonymous Referee #2 for taking the time and effort to read our manuscript and expressing the generally positive impression of our work. We will use the constructive comments to improve our manuscript. Please, find below our point-to-point response (comment of the referee in **black**, our response in **blue**).

**General comments :**
The manuscript is well written and easy to read, although some sections need to be reworded throughout the paper to improve readability and highlight the scientific contribution of this study. My main concern is with the overall presentation of the article. I found that the innovative aspect of this study was hidden by general facts and conclusions that have already been proven in the past. I understand the importance of such conclusions in justifying a new parametrisation but, in my opinion, focusing on this key point weakens the overall quality of the article. I suggest that the authors improve the paper to focus on the importance of using the developed parametrisation and to emphasise the need for such a statistical model. In this spirit, I recommend to better integrate this work into the current scientific literature.

We would like to thank the referee for the well-supported specific comments. We think this has been of great help to us to make suggestions for changes. Through these suggested changes we think the manuscript will be more precise and focussed, and it will improve the overall quality of the manuscript.

Furthermore, some of the results discussed are not presented in detail and it is therefore impossible to review this information. You should either delete these results or provide the details.

We screened the manuscript regarding this issue, and in the specific comments you will find what we decided for each occurrence.

All the comments do not call into question the study itself. The material for a good article is already there and just needs some rearrangement and minor revisions. I am convinced that the article will gain in precision and interest with additional information.

We would like to thank the referee for this positive and supporting feedback. We appreciate that.

Specific comments are detailed below

**Abstract**

General comments : In this section, the objective of the paper is not clearly pointed out. In my opinion, you should focus on the specific parametrisation you proposed for Lake Ijssel based on the field measurements. All the elements are already written and you just need to rearrange the section.

We agree and we propose to change the abstract to:
*"We study the controls on open water evaporation of a large lowland reservoir in the Netherlands. To this end, we analyse the dynamics of open water evaporation at two locations, i.e. Stavoren and Trintelhaven, at the border of Lake IJssel (1100 km²) where eddy covariance systems were installed during the summer seasons of 2019 and 2020. These measurements were used to develop data-*

*driven models for both locations. Such a statistical model is a clean and simple approach that can provide a direct indication and insight of the most relevant input parameters involved in explaining the variance of open water evaporation, without making a prior assumptions on the process itself. This way, we find that a combination of wind speed and the vertical vapour pressure gradient can explain most of the variability of observed hourly open water evaporation. This is in agreement with Dalton's model which is a well-established model often used in oceanographic studies for calculating open water evaporation.*

*Validation of the data-driven models demonstrates that a simple model using only two variables yields satisfactory results at Stavoren, with $R^2$ values of 0.84 and 0.67 for hourly and daily data, respectively. However, the validation results for location Trintelhaven fall short ($R^2$ values of 0.65 and 0.44 for hourly and daily data, respectively). Using only routinely measured meteorological variables leads to adequate performing simple models at hourly ($R^2 = 0.79$ at Stavoren, and $R^2 = 0.51$ at Trintelhaven) and daily ($R^2 = 0.86$ at Stavoren, and $R^2 = 0.83$ at Trintelhaven) timescales. These results for the summer periods show that global radiation is not directly coupled to open water evaporation at the hourly or daily timescale, but it rather is a combination of wind speed and vertical gradient of vapour pressure. We would like to stress the importance of including the correct drivers of open water evaporation in the parametrization in hydrological models to adequately represent the role of evaporation in the surface-atmosphere coupling of inland water bodies.*

-P1.L1 : I would talk about a « sink » rather than a « large loss term ». It is more adequate to the scientific level of the journal.

This sentence will be removed in the newly proposed abstract. But we agree with the feedback and we will adjust this in the introduction section P2.L3 to '*Evaporation is a sink in the water balance of inland water bodies.*'

-P1.L1 : « During summer seasons, which are projected to become warmer with more severe and prolonged periods of drought ». This is a general sentence whereas your study focused on a specific location. Even if we are on a global climate change path, the consequences (not specifically warmer summer) are not the same worldwide. You should be more specific on the spectrum of warming on the studied area (or region) and put references.

This sentence will be removed in the newly proposed abstract, but in the introduction section we will be more specific on the region that we are referring to, i.e. the Netherlands. In addition to the already mentioned references in the introduction targeted to mid-latitude regions in Europe, we will refer to the KNMI'14 climate scenario's for the Netherlands specifically.
We will adjust accordingly: P2.L7 – '*Summer seasons are projected to become warmer in the Netherlands, with more severe and prolonged periods of drought (Seneviratne et al., 2006, 2012; KNMI, 2015; Teuling, 2018; Christidis and Stott, 2021).*'

KNMI, 2015: KNMI'14-klimaatscenario's voor Nederland; Leidraad voor professionals in klimaatadaptatie, KNMI, De Bilt, 34 pp

- P1.L8 : « not available energy» Be specific on the type of energy.

With available energy we mean $R_n – G$ (P9.L4), but we agree that in this context this may not have been clear. We will remove the term from the abstract as it does not support clarity of the sentence and the statement that we make there.

-P1.L11 : « main drivers » Be specific. What type of phenomenon they are the drivers of?

We agree we should be more specific here. However, in the proposed revised abstract this sentence has been removed.

- P1.L15 : « well performing simple data-driven models » I would be less enthusiastic with a $R^2$ of 0.51 and 0.43. The model is adequate but does not perform well.

We will add the required nuance of our statement. P1.L15 – '*Using only routinely measured meteorological variables leads to adequate performing simple models at hourly ($R^2 = 0.79$ at Stavoren, and $R^2 = 0.51$ at Trintelhaven) and daily ($R^2 = 0.86$ at Stavoren, and $R^2 = 0.83$[1] at Trintelhaven) timescales.*'
[1] Note that the numbers have changes after correcting a mistake (see our response to point 1 of anonymous reviewer #1)

**Introduction**

- P2.L1 : There is a more up-to-date review paper you should include: Woolway, R. I., Kraemer, B. M., Lenters, J. D., Merchant, C. J., O'Reilly, C. M., & Sharma, S. (2020). Global lake response to climate change. Nature Reviews Earth & Environment, 1(8), 388-403.
I would suggest to have a look at the paragraph about lake evaporation which gives essential materials for both your introduction and your discussion.

Thank you for the suggestion for this interesting paper. We will add the reference of Woolway et al. (2020) to P2.L2: '*Inland water bodies are known to interact with the local, regional and even global climate and are therefore highly sensitive to climate change (Adrian et al., 2009; Liu et al., 2009; Wang et al., 2018; Woolway et al., 2020).*

We will adjust and add the following to P2.L4-5: '*... how open water evaporation ($E_{water}$) will respond to these changing conditions. It is expected that changes in longwave radiation, Bowen ratio, ice cover, and stratification will affect the dynamics of $E_{water}$ at the long-term (Wang et al., 2018; Woolway et al., 2020). Whereas at the shorter decadal timescale, a contribution to trends and variations in $E_{water}$ is expected resulting from changes in wind speed, humidity, and also through global and regional solar dimming and brightening and its effect on water surface temperature (Desai et al., 2009; McVicar et al., 2012; Schmid and Köster, 2016; Wang et al., 2018; Woolway et al., 2020). During the summer season evaporation rates are...*'

Desai, A.R., Austin, J.A., Bennington, V. & McKinley, G.A. Stronger winds over a large lake in response to weakening air-to-lake temperature gradient. Nat. Geosci. 2, 855-858 (2009)
McVicar, T.R. et al. Global review and synthesis of trends in observed terrestrial near surface wind speeds: Implications for evaporation. J. Hydrol. 416–417, 182–205 (2012)
Schmid, M. & Köster, O. Excess warming of a Central European lake by solar brightening. Water Resour. Res. 52, 8103–8116 (2016)
Wang, W. et al. Global lake evaporation accelerated by changes in surface energy allocation in a warmer climate. Nat. Geosci. 11, 410–414 (2018)
Woolway, R. I., Kraemer, B. M., Lenters, J. D., Merchant, C. J., O'Reilly, C. M., & Sharma, S. (2020). Global lake response to climate change. Nature Reviews Earth & Environment, 1(8), 388-403.

We will add a sentence to P4.L4 (see specific comment on P4.L3): *'Adequate estimations of $E_{water}$ are important in this context because there is a strong coupling between $E_{water}$ and for instance lake level and extent, the lake ecosystem, and lake stratification and mixing regimes (Woolway et al., 2020; Jenny et al., 2020).'*

In addition, we will add the following to P18.L6 (see specific comment to P14.L22): *'This is similar to what has been found by studies of for instance Blanken et al. (2011) and McGloin et al. (2014), and it was noticed that intraseasonal variations of $E_{water}$ can be linked to synoptic weather variations through these variables (Lenters et al., 2005, MacIntyre et al. 2009, Liu et al., 2011, Woolway et al. 2020).'*

- P2.L2 : « evaporation is a large loss term of water bodies … » I would rephrase by saying this is « a sink in the lake water balance ». Also you should add a reference to justify this, even if it's a validated fact.

Thank you for the suggestion. We will change the terminology. We would like to argue that the fact that evaporation is a sink in the water balance of a lake can be referred to as common knowledge, and therefore does not need a reference.

- P2.L7 : « Summers are projected to become warmer » As mentioned for the abstract section, as you work on a specific location, you should be more specific on such fact as you are not working at global scale (mention the spatial scale). Moreover, I would recommend to give the climate reference on which the climatic comparison is made. You should also add a reference.

We agree that we should be more specific on the region. Therefore, we suggest to change the sentence to: *'Summer seasons are projected to become warmer in the Netherlands, with more severe and prolonged periods of drought (Seneviratne et al., 2006, 2012; KNMI, 2015; Teuling, 2018; Christidis and Stott, 2021).* This should make the statement more precise.

KNMI, 2015: KNMI'14-klimaatscenario's voor Nederland; Leidraad voor professionals in klimaatadaptatie, KNMI, De Bilt, 34 pp

- P2.L12-13 : « In terms of thermodynamics, …» This works on some lakes but this is not always correct. In terms of thermodynamics, big lakes (such as the American or African Great Lakes) could be approach by using either 3D ocean model or 1D model. It will depends on the presence or not of the hypolimnion and furthermore on the stratification dynamic, if there is one.
See :
*Xue, P., Pal, J. S., Ye, X., Lenters, J. D., Huang, C., & Chu, P. Y. (2017). Improving the simulation of large lakes in regional climate modeling: Two-way lake–atmosphere coupling with a 3D hydrodynamic model of the Great Lakes. Journal of Climate, 30(5), 1605-1627.*
*Gronewold, A. D., & Stow, C. A. (2014). Water loss from the Great Lakes. Science, 343(6175), 1084-1085.*
*Thiery, W. I. M., et al. "LakeMIP Kivu: evaluating the representation of a large, deep tropical lake by a set of onedimensional lake models." Tellus A: Dynamic Meteorology and Oceanography 66.1 (2014): 21390.*

We agree that the statement we make here is not applicable to all inland water bodies and does depend on stratification and should be better focused on our target water body.

*Therefore, we suggest to change it to: 'In terms of thermodynamics a shallow inland water body of only a few meters deep'*

- P2.L17 : This might be a detail but I would prefer to talk about a change of the amplitude (which suppose an increase of surface temperature during daytime but also a quicker decrease during night-time).

*Yes, we are referring to the same process, using other words. We suggest to change the sentence to: P2.L16 '...., where heat is stored in the lower atmosphere, vegetation and the upper soil layers. This leads to larger temperature amplitudes in sunny conditions, with strongly increasing surface temperatures and warming of the lower atmosphere during daytime, and strong decreases during night-time.'*

- P2.L21: Lake depth also controls the dynamical range of lake temperature amplitudes on diurnal timescale.

*See also our response to point 3 made by anonymous reviewer #1 on the simulation of water temperature using a simple energy balance model of a water layer.*
*We would like to suggest to clarify the text at P2.L23 into: 'The subsurface energy budget implies that Lake depth controls the dynamical range of lake temperature amplitudes on diurnal timescale. Thus, instead of focussing at the surface only, rather the whole volume of the system should be considered.'*

- P3.L2-6 : This paragraph would gain in readability if you reduce the description to its essential. Penman equation is well-known and its description can be shortened. Moreover, this description is redundant with the one on P9.

*We agree that the Penman equation is well-known, but we also think that it is important to stress the assumption made by Penman (i.e. assuming energy storage below the surface to be neglected, resulting in an instantaneous response of the surface temperature), which makes it less straightforward to use it for water bodies of a few meters deep. Moreover, it specifies the essential difference between the Penman and the Dalton equation. Overall, we think the balance in the introduction is good (i.e. 1 paragraph dedicated to the description of the Penman and the Dalton model). Therefore, we prefer to keep the text as is.*

- P3.L7 : « Most studies ... » Is this sentence linked with the reference list starting on L9 ? If so, you should move the sentences « However, measurements of... » and « This can partly ... » elsewhere. Also, you said the contrary on L.26 "In the past, a number of studies reported ..." Moreover, I would not be that direct by saying measurements of evaporation from inland water bodies are under-represented, numerous studies have been published on the subject for the past 10 years:

*Potes, M., Salgado, R., Costa, M. J., Morais, M., Bortoli, D., Kostadinov, I., & Mammarella, I. (2017). Lake–atmosphere interactions at Alqueva reservoir: a case study in the summer of 2014. Tellus A: Dynamic Meteorology and Oceanography, 69(1), 1272787.*
*Pillco Zolá, R., Bengtsson, L., Berndtsson, R., Martí-Cardona, B., Satgé, F., Timouk, F., ... & Pasapera, J. (2019). Modelling Lake Titicaca's daily and monthly evaporation. Hydrology and Earth System Sciences, 23(2), 657-668.*
*Moigne, P. L., Legain, D., Lagarde, F., Potes, M., Tzanos, D., Moulin, E. R. I. C., ... & Costa, M. J. (2013). Evaluation of the lake model FLake over a coastal lagoon during the THAUMEX field campaign. Tellus A: Dynamic Meteorology and Oceanography, 65(1), 20951.*

*Blanken, P. D., Spence, C., Hedstrom, N., and Lenters, J. D.: Evaporation from Lake Superior: 1. Physical Controls and Processes, Journal of Great Lakes Research, 37, 707–716, https://doi.org/10.1016/j.jglr.2011.08.009, 2011*

Thank you for bringing up these references, we will include these in the study to integrate our work more into the current scientific literature.

Assuming that there is referred to the reference list at L12, it is not correct that these belong to our statement in L7 '*Most studies...*'. We therefore prefer to keep that as is.

However, we agree that our statements in L7 and L26 could read as *contradictory*, but what we mean here is that comparably there have been a lot more studies focussing on the understanding of terrestrial evaporation, and much less studies focussed on open water evaporation. Fortunately, there have definitely been studies that measured and modelled open water evaporation (also shown by the references provided by you). To avoid confusion, we suggest to change the sentence (P3.L7/8) to: '*However, comparably significantly less studies performed measurements of $E_{water}$ from inland water bodies.*'

- P3.L27 : I'm not convinced about the utility of the brackets. Moreover you could also include other important hydroclimate variables.
*Zhou, W., Wang, L., Li, D., & Leung, L. R. (2021). Spatial pattern of lake evaporation increases under global warming linked to regional hydroclimate change. Communications Earth & Environment, 2(1), 1-10*

Agreed on the brackets; we will remove them.

The coupling between lake evaporation and hydroclimate (P-E) as referred to in Zhou et al. (2021), is found on larger timescales (decades), and a link is made to changes in lake evaporation under different climate scenarios. It therefore provides another concept to describe lake evaporation, but at another timescale than the other studies that we are referring to in these lines (P3.L29-31). We suggest to add the following (P3.L31): '*At larger timescales a spatial coupling was found between $E_{water}$ and P-$E_{terrestrial}$ (Zhou et al., 2021). Jansen and Teuling (2020) studied the performance of a number of concepts that are commonly used to describe open water evaporation.*'.

- P3.L34 →P4.L2: Woolway et al 2018 & Wang et al 2018 have addressed this issue. Moreover, even if I agree with your assumption, I'm not convinced about such parametrisation for use at longer timescale (for example, seasonal timescale). Lake temperature and evaporation are interdependent on such timescales and other hydroclimate variables should also be included.

*Woolway, R. Iestyn, et al. "Geographic and temporal variations in turbulent heat loss from lakes: A global analysis across 45 lakes." Limnology and Oceanography 63.6 (2018): 2436-2449.*
*Wang, Wei, et al. "Global lake evaporation accelerated by changes in surface energy allocation in a warmer climate." Nature Geoscience 11.6 (2018): 410-414.*

Thank you for the additional references. Our statement about the disagreement of the methods on the average increasing historical trend of the evaporation rate, as well as for the projected future trends is based on the findings in our previous study (Jansen and Teuling, 2020). That study showed that the choice of method, with different representations of the evaporation process, can lead to significantly different projected trends. We do agree with the point you raise here that lake temperature and evaporation are interdependent on these longer timescales, which requires the water body energy balance to be represented correctly. That observation actually supports our statement that it is important to find a way to correctly represent the evaporation process for the timescale that is studied, i.e. hourly and daily.

We suggest to make the following adjustment P3.L1-2:
*'At longer timescales (i.e. seasonal and yearly timescales) it is important to include the interdependency between lake temperature and evaporation. This requires a concept in which the water body energy balance to be represented adequately, for the correct modelling of the $E_{water}$ process.'*

- P4.L3 : This sentence is the key point of your study. More generally, the paragraph from L3 to L14 should be the core of your introduction. I would reduce the presentation of the different equation (Penman, Makkink) and enrich this paragraph.
Add a reference for this : « a crucial element in its water management system ». Also, you focus on the water management aspect however your paper does not specifically study the impact of the parametrisation on the lake hydrology. I would recommend to add other aspects of the evaporation as a component of the global energy and water cycle. For example, you can talk about the influence on the near-surface turbulence intensity, the stratification or the lake ecosystem.

*Raymond, P. A., and others. 2013. Global carbon dioxide emissions from inland waters. Nature503:355–359. doi:10.1038/nature12760*
*Jenny, Jean-Philippe, et al. "Scientists' warning to humanity: rapid degradation of the world's large lakes." Journal of Great Lakes Research 46.4 (2020): 686-702.*

It is correct that this last paragraph of our introduction includes the aim of our study. We brought it into context of previous studies and core concepts that have been used, and are still frequently used, in calculating evaporation (Penman, Makkink, Dalton). So we think it is important to introduce that as well in this section. We agree to add a reference to 'a crucial element in its water management system' (*Buitelaar et al., 2015*). In addition, this study was performed to bring the importance of the correct parametrization of $E_{water}$ in context of lake hydrology. Especially because, as you mention, there is a strong connection between $E_{water}$ and e.g. lake level and extent, lake ecosystem, and lake stratification and mixing regimes (Woolway et al., 2020). Through these connections, $E_{water}$ affects the water management of the lake in terms of drinking water services and water availability for agricultural land (Jenny et al., 2020).
We would like to suggest to add the following text at P4.L4:
*'....in its water management system (Buitelaar et al., 2015). Adequate estimations of $E_{water}$ are important in this context because there is a strong coupling between $E_{water}$ and for instance lake level and extent, the lake ecosystem, and lake stratification and mixing regimes (Woolway et al., 2020; Jenny et al., 2020).'*

Woolway, R. I., Kraemer, B. M., Lenters, J. D., Merchant, C. J., O'Reilly, C. M., & Sharma, S. (2020). Global lake response to climate change. Nature Reviews Earth & Environment, 1(8), 388-403.
Jenny, et al. "Scientists' warning to humanity: rapid degradation of the world's large lakes." Journal of Great Lakes Research 46.4 (2020): 686-702.
Buitelaar, R., Kollen, J., Leerlooijer, C. (2015). Rapport Operationeel waterbeheer IJsselmeergebied - Inventarisatie huidige waterbeheer IJsselmeergebied door Rijkswaterstaat en Waterschappen. Report. 112 pp. Grontmij. Alkmaar.

- P4.L12-15 : In this sentence, you compare Makkink's equation with Flake simulations. I'm wondering why you are not using FLake directly for lake IJssel if you consider Flake simulations as your reference ?

We agree that in this sentence it seems that we treat Flake as a reference. However, the main focus of our paper is to develop a simple statistical model from measurements, which can provide a simple and clean solution. So we use the measurements as a reference. To our knowledge Flake was never tested before against EC measurements for Lake IJssel. The comparison is made because Flake is a physically-based model, which is also integrated in the ECMWF model for instance, and we therefore assume it will perform better than the Makkink equation at these timescales, while the latter is currently used for estimating evaporation from Lake IJssel, so that is where our interest lies.

**Data, Material and Methods**

- P6.L8: It seems that the KNMI station only measure global radiation (as I see in the data provided), however would it be possible to have access to the four components of the global radiation? As shown in Wang et al 2018, the incoming radiation has an effect even if it's at longer timescale.
*Wang, Wei, et al. "Global lake evaporation accelerated by changes in surface energy allocation in a warmer climate." Nature Geoscience 11.6 (2018): 410-414.*

The assertion is correct: the operational KNMI stations only provide global radiation data (i.e. incoming solar radiation). Thanks to your remark here however, we did notice we have not explained how we obtained net radiation from the meteorological data. This will be added to section 2.5, where the Penman equation is explained. At longer timescales global radiation indeed will affect evaporation rates, that is why we also included this variable in the regression analysis.

P9.L6: '... *from the water surface. Net longwave radiation was calculated according to the equations* $L_{in} = \varepsilon_a \sigma T_a^4$ *and* $L_{out} = L_{e,\,out} + (1 - \varepsilon_s)L_{in}$, *(equations 2.24 and 2.28 in Moene and Van Dam, 2014) and net shortwave radiation as* $K^* = (1-\alpha)K_{in}$ *(Allen et al., 1998) with average monthly albedo values calculated as function of latitude (Cogley, 1979).*'

Moene, A. F. and van Dam, J. C.: Transport in the Atmosphere-Vegetation-Soil Continuum, Cambridge University Press, Cambridge, 2014.
Allen, R. G., Pereira, L. S., Raes, D., and Smith, M.: Crop Evapotranspiration - Guidelines for Computing Crop Water Requirements - FAO Irrigation and Drainage Paper 56, United Nations-Food and Agricultural Organization, 1998.
Cogley, J. Graham. "The Albedo of Water as a Function of Latitude." *Monthly Weather Review* 107 (1979): 775-781.

- P6.L30: Could you please rephrase this sentence to improve readability.

We suggest to change the sentence into: 'Firstly, the raw data were quality-controlled using several criteria in order to remove faulty or corrupted data.'

- P7.L30-32: "A regression analysis …" + "To develop …" Please rephrase to improve readability.

We suggest to change the sentence into: 'A regression analysis was performed to explore which variable, or combination of variables, can best explain the dynamics of $E_{water}$. Variables included in this analysis were wind speed, VPD, global radiation, vertical vapour pressure gradient, air temperature and water temperature. From the regression analysis a data-driven model was developed to estimate $E_{water}$ of Lake IJssel. This was done for both locations, Stavoren and Trintelhaven.'

- P8.L11: I found difficult to understand the justification of using such regression model and how the hypothesis of such model have been tested.

What are the type of estimator you used (I assume an ordinary least square estimator)? Did you perform a significativity test? It would be interesting to look at the result of the multiple linear regression model and specifically the p_value to include or exclude predictors. Are the period chosen representative of the population?

The functional form of the regression models was chosen to be a simple combination (sum or product) of variables only considering linear regression, multiple linear regression and quadratic regression models, which was a data-driven decision rather than a process-based decision.
Yes, statistical significance was tested on the used models (p-values < 0.05; this will be added at P8.L10 '… a single variable. Statistical significance (p < 0.05) was tested. From the multitude…'), meaning the best and simple models that we continued working with: Fig 6+7 and Table 2. The Venn diagrams however show the model fit of all combinations, without an indication of significance. We will adjust this to the Venn diagrams by removing those values where the model fit was found insignificant. We will add to the caption of the Venn diagrams: 'Values were removed if the model fit was found to be insignificant (p < 0.05).'
Since our aim is to study the surface-atmosphere coupling, we did not perform gap filling, resulting in less data points, but avoiding the use of 'artificial' data. That is the trade-off to be made and as a consequence the summer period chosen for calibration is all data that we have. The fact that the chosen models that were fit on this period are significant, provides us confidence, as well as the relatively good validation results.

- P8.L22: "surface temperature" Are you talking about the Meteosat product? Hence, why do not use directly these field data? Are there representative of the surface temperature (the lake is shallow then it would be important to be sure the measurements are not performed in the thermocline).

What we are trying to convey here is that there are no routine observations of the surface water temperature, or the skin temperature, measured with for instance thermal infrared cameras. Due to these lacking camera's we were curious to find out if water temperature measured routinely at 1.2 to 1.5 m deep would suffice in estimating $E_{water}$. To clarify this we suggest to replace the text at P8.L20-22 by: 'There are no routine observations available of the skin water temperature of the lake. As an alternative, the use of water temperature data routinely measured by Rijkswaterstaat at depths ranging from 1.2 to 1.5 m was explored.'

**Results**

General comments: In this section, I would have a distinct paragraph presenting the results of the calibration, another for the validation and a final one for the result on the routinely measured variables. It would also improve the readability as I had hard time following this section.

Thank you for this feedback. In response to this valuable suggestion, what we will do to improve readability is to make this distinction in presenting the results of 3.4.1) the calibration, 3.4.2) the validation, and 3.4.3) the routinely measured variables, as subsections of Section 3.4.

- P11.L5; Fig2: How did you choose the time period presented in the figure? Why do you present this period instead of either the training or validation time period? As I understand your paper is about summer and here you present a part of the autumn season. I admit I was a bit lost. Also I would

suggest to be more specific on the time period (e.g: 01/05-31/08 instead of nouns (You write May-August most of the time and once May-September).

The idea behind this figure, as well as figure 3, was to explore how the dynamics and trends of the meteorological variables, and especially the heat fluxes, would evolve before, during and after the summer, to explore if any lags for instance would occur. That is the reason why we presented here the period 01/05 – 30/09.

To clarify this we suggest to add the following to P10.L2: '*This figures illustrates the dynamics and trends of the meteorological variables and the heat fluxes before, during and after the summer period to explore if any lags for instance would occur at this timescale.*'

And at P11.L2 '*The monthly average diurnal variability of observed LE, based on hourly data, are shown in the top panels of figure 3 for location Stavoren for the same period as in figure 2 (i.e. 01/05/2019 – 30/09/2019).*'

- Section 3.2: In this section, you compare meteorological conditions. It seems you are comparing the air temperature and the wind speed that are are measured at different height. Measurements at Stavoren are made around 7m and around 10m at Trintelhaven. Did you adjust your measurement to an equivalent height? If not, this could explain some of the discrepancies. I have the same question for EC data. Moreover, you compared these variables to the Dalton model which needs variables at 2m height. What was the procedure you used to adjust the measurements to this height?

You are right that we have not adjusted our measurements to an equivalent height. We will make a remark on this at P6.L6: '*The measurement height at the two locations Stavoren and Trintelhaven differ. In our analysis we have not adjusted the measurements to an equivalent height. In theory, the small height difference will not affect the heat fluxes under the assumption of a constant turbulent flux layer.*'
Furthermore, the difference in wind speed found between the two locations cannot be explained by the difference in measuring height, as the resulting average wind speed actually is higher in Stavoren which is measured closer to the surface. The regression coefficients found might differ slightly because of the difference in measurement height. However, we do think that the variables that were found to be most important to explain the variance of $E_{water}$ (i.e. wind speed and vapour pressure gradient) will not change.
Considering the remark about comparison with the Dalton model – there are two notes on this: 1) in the caption of figure 3 ('*Note that some variables included in the evaporation models are measured at larger heights than the 2 m that are prescribed (see Eq. 6 – 9)*', and 2) in the discussion at P18.L12-14. This should be taken into account. However, we do think using variables measured at greater heights will not change the diurnal trends that were found.

- P10.L20: "the water temperature at" instead of "the water at".

P.10.L10: Correct, will be adjusted.

- P10.L18: Please provide the correlation score to justify this is a strong correlation.

We will indicate the correlation score here, which is $R^2$=0.61. But, moreover, what we would like to indicate here is that from looking at time series only, one could see that latent heat flux and wind

speed are showing similarity in their trend, suggesting a good correlation between the two variables. This gives an indication for further analysis. Therefore, we will change the sentence into: '*The latent heat flux displays similar trends as the measured wind speed, indicating that the two variables are correlated ($R^2$ = 0.61).*'

- P11. Fig2: Please put the graph corresponding to the 2020 summer period in Appendix.

Agreed. We will do this.

- P12.L1: I would use the word "pattern" instead of "rhythm".

Agreed. Will be changed.

- P12.L5: Could you please add the graph in the Appendix?

We will do that indeed.

- P13.L17: Could you please add the correlation score to justify if it's a strong correlation. Also, be careful not to mismatch between correlation and determination when you analyse your results and even more as you are studying a non-linear model. Please rephrase the sentence "Global radiation …" to account for this difference.

The square of the correlation coefficient between wind speed and latent heat flux following from figure 2 is $R^2$=0.61. To match this with previous wording we will change it into '… with the adequate correlation ($R^2$=0.61) visible in figure 2.'. Furthermore, we will adjust P13.L17 to 'Global radiation and VPD have the lowest adjusted coefficient of determination, which agrees with our findings in figures 2 and 3.'

- P14.L4-6: I understand that you only exclude the global radiation from your model based on the $R^2$, however, in my opinion a $R^2$(VPD)=0.05 questioned the inclusion of VPD in your model. For example, adding VPD on the Stavoren hourly analysis has a limited impact which is not significant in Trintelhaven. Does VPD has a significant impact on your score?

We have checked the addition of VPD to the model again, and indeed it appeared to be significant. However, due to a mistake that we found as a result of one of the feedback points from anonymous reviewer #1, the Venn diagrams changed: only slightly for the hourly diagrams, but quite significantly for the daily diagrams. As a result, the best regression model is the same as the simple regression model: $LE_{mod}$ = 5.1 $u\Delta e$ + 1.6 $(u\Delta e)^2$ + 42 ($R^2$ = 0.74). Adjustment of the figures and accompanying text will be changed accordingly.

- P14.L22: Would it be possible to have some basics statistics on these data (mean, standard deviation, quantiles, min, max). It would also help to see if outliers are ejected from the analysis. A Table placed in the Appendix would be sufficient and would give a hint about the discrepancies between both summer seasons.

We will add the following table to the Appendix:

| (a) **Stavoren** | Mean | Stdev | min | max | Q25 | Q75 | N |
|---|---|---|---|---|---|---|---|
| Hourly | | | | | | | |
| 2019 | 103.6 | 69.1 | -24.4 | 516.1 | 56.8 | 132.3 | 935 |
| 2020 | 128.0 | 87.9 | -27.7 | 444.6 | 66.1 | 164.5 | 716 |
| Daily | | | | | | | |
| 2019 | 117.0 | 63.9 | 51.3 | 296.3 | 71.7 | 139.1 | 25 |
| 2020 | 154.6 | 82.8 | 56.5 | 365.1 | 102.6 | 194.2 | 18 |

| (b) **Trintelhaven** | Mean | Stdev | min | max | Q5 | Q95 | N |
|---|---|---|---|---|---|---|---|
| Hourly | | | | | | | |
| 2019 | 93.9 | 59.2 | 11.0 | 333.9 | 46.2 | 128.9 | 473 |
| 2020 | 90.9 | 59.8 | -34.8 | 351.8 | 51.6 | 112.9 | 697 |
| Daily | | | | | | | |
| 2019 | 109.8 | 49.4 | 35.2 | 213.8 | 74.1 | 140.8 | 12 |
| 2020 | 97.4 | 49.8 | 37.0 | 261.6 | 69.4 | 104.7 | 19 |

- P14.L15: $R^2$ explains 45% of the variance which is quite low. If you include the water temperature, it reaches 0.48, it is still low but better. You limit the maximal number of variables for the simple model but in this case it would benefit to your model to add the water temperature.

Due to solving the mistake that we mentioned earlier the numbers have changed as well as the combination of variables leading to the best and the simple model. The Venn diagram of Trintelhaven where the sum of variables is given for daily timescales (Fig. 5d) now gives the highest $R^2$ for wind speed and $\Delta e$. Additionally, it is simply a choice to explore both 1) the best model which can include as many variables as possible, and 2) to limit the simple model to two variables at the most, as explained in P8.L12. Adjustment of the figures and accompanying text will be changed accordingly.

- P14.L22: "This can be attributed …" As noticed in Woolway et al 2021, lake evaporation is highly dependent on weather variability (through its dependence to the lake surface temperature). Your discussion need to stress this issue and not just focus on the comparison to a mean climate. I would remove this sentence and discuss about this point in the adequate section.

Thank you for this suggestion. We will add some words on this topic to the discussion section for completeness. However, we think that the dependence of lake evaporation to synoptic weather variations through the dependence with lake surface temperature is actually included in the regression analysis, where $T_{water}$ and vapour pressure gradient (which combines wind speed and humidity, both also related to synoptic weather variations) are two of the variables that were included.

We will adjust the following to P18.L6: '*This is similar to what has been found by studies of for instance Blanken et al. (2011) and McGloin et al. (2014), and it was noticed that intraseasonal variations of $E_{water}$ can be linked to synoptic weather variations through these variables (Lenters et al., 2005, MacIntyre et al. 2009, Liu et al., 2011, Woolway et al. 2020). The same ingredients of wind speed and vapour pressure gradient were used in the model by Dalton (1802).*

Lenters, J. D., Kratz, T. K., and Bowser, C. J.: Effects of Climate Variability on Lake Evaporation: Results from a Long-Term Energy Budget Study of Sparkling Lake, Northern Wisconsin (USA), Journal of Hydrology, 308, 168–195, https://doi.org/10.1016/j.jhydrol.2004.10.028, (2005)

MacIntyre, S., Fram, J.P., Kushner, P.J., Bettez, N.D., O'Brien, W.J., Hobbie, J.E. & Kling, G.W. Climate-related variations in mixing dynamics in an Alaskan arctic lake. Limnol. Oceanogr. 54(6, part 2), 2401-2417 (2009)

Liu, H., Blanken, P.D., Weidinger, T., Nordbo, A. & Vesala, T. Variability in cold front activities modulating cool-season evaporation from a southern inland water in the USA. Environ. Res. Lett. 6(024022) (2011)

Woolway, R. I., Kraemer, B. M., Lenters, J. D., Merchant, C. J., O'Reilly, C. M., & Sharma, S. (2020). Global lake response to climate change. Nature Reviews Earth & Environment, 1(8), 388-403.

- P14.L26: "this confirms …" Rephrase this sentence. If the ingredients are the same than in the Dalton's model why do not use this model or use a calibrated version on your lake?

The aim of our study is to find the drivers of $E_{water}$ based on observations, without predetermining the variables to be included. The observations were used to develop the regression models, which confirmed a very similar relation to what was found by Dalton. We think this only helps to gain confidence to the fact that indeed wind speed and $\Delta e$ are the most important drivers of $E_{water}$. We will replace P14.L26 with the following text: '*Without predetermination of the variables, we found the same ingredients as used in the Dalton model as the most important drivers of $E_{water}$ at hourly and daily timescales.*'

- P14.L28: "To determine if the coefficients …" Without the results of the analysis it is impossible to assess the results. Either give the results or erase this sentence.

The result from the ANOVA analysis showed that with a p-value of 0.02, that the inclusion of the station (i.e. Stavoren or Trintelhaven) matters. So we cannot use the same model coefficients for both locations, and in other words: we cannot rule out that the sites are different (P14.L29). We will add the ANOVA table as Appendix, and adjust P14.L29: '*…, an ANOVA statistical analysis was performed (see Appendix ...). This analysis shows that inclusion of the station matters (p < 0.05). Therefore, we cannot rule out that the sites are different.*'

```
> anova(fit0, fit1)
Analysis of Variance Table

Model 1: LE ~ u_De + I(u_De^2)
Model 2: LE ~ u_De * station + I(u_De^2) * station
  Res.Df    RSS Df Sum of Sq      F  Pr(>F)
1    559 664256
2    556 652804  3     11452 3.2513 0.02151 *
* * *
Signif. codes:  0 '***' 0.001 '**' 0.01 '*' 0.05 '.' 0.1 ' ' 1
```

- P16.L5-9: "The results for the location of …" This is a good analysis of your results. However be sure to be consistent. In your abstract you say that the model performs well.

Agreed. We checked for consistency and adjusted the abstract: '*Validation of the data-driven models demonstrates that a simple model using only two variables yields satisfactory results at Stavoren, with $R^2$ values of 0.84 and 0.67 for hourly and daily data, respectively. However, the validation results for location Trintelhaven fall short ($R^2$ values of 0.65 and 0.44 for hourly and daily data, respectively). Using only routinely measured meteorological variables leads to adequate performing simple models at hourly ($R^2 = 0.79$ at Stavoren, and $R^2 = 0.51$ at Trintelhaven) and daily ($R^2 = 0.86$ at Stavoren, and $R^2 = 0.83$ at Trintelhaven) timescales.*'

**Discussions**

You need to improve your discussions and criticise your result in a more precise way. You could be more exhaustive and include limitations (e.g decomposition of the radiation term, looking at the influence of other hydroclimate variables on the variance).

- P17.L7: You should be more precise and discuss the fact that you do not analyse each term of the radiation budget.
Agreed. We will remove the brackets around <<global>> at P17.L7, and we will add the following to P17.L9: '... figure 3). '*Note that the relation between $E_{water}$ and other components of the radiation budget could not be studied, because of the lack of observations of these components. In combination with absent data on G, this prevented us from fully capturing the role of net radiation in the energy balance of the lake, and thus with the warming and cooling of the lake, which relates to evaporation through the water surface temperature.'*

- P18.L6: you can add Le Moigne et al 2016 as a reference.

Thank you for this suggestion. As far as we can see the reference mostly focusses on the impact of lakes on climate using the Flake lake scheme. It does not directly support the statement we make here about the combination of wind speed and vapour pressure gradient to estimate $E_{water}$. However, we will include this interesting reference to the introduction at P2.L3.

*Patrick Le Moigne, Jeanne Colin & Bertrand Decharme (2016) Impact of lake surface temperatures simulated by the FLake scheme in the CNRM-CM5 climate model, Tellus A: Dynamic Meteorology and Oceanography, 68:1, DOI: 10.3402/tellusa.v68.31274*

- P19.L10-13: It is impossible to review this part of the discussion as you don't provide the results. You should either erase the sentence or give the correlation plots.

Agreed. After consideration, we decided to erase these sentences as it does not contribute substantially to our storyline to add more graphs.

- As mentioned in the precedent comment (for P14.L22), you are working at short timescale and thus, the lake evaporation is dependent on the weather and the hydrological variability. Your discussion would be more complete by discussing these points.

We agree that this point should be noticed in the discussion section. In the precedent comment for P14.L22 we have added the suggested change to the text.

**Conclusions**

- P21.L19: In my opinion, your main contribution is the development of the statistical model. I would suggest to rephrase your conclusion in order to account for this.

Indeed the development of the statistical model forms a significant part of this study. However, the statistical model is used as a tool to explore the drivers of open water evaporation. We suggest to make the following adjustment at P21.L19: '*In this study, we investigated the dynamics and drivers of*

*open water evaporation of Lake IJssel in the Netherlands through the development of a data-driven model.'*

- P21.L20: Ok but this a general fact and this is not your main result.

We stated it there to provide a context for readers, but in order to kick-off with our main conclusions we will remove the following sentences: P21.L20 'We have shown....' till P21.L23 '....of open water evaporation'.

- P21.L26-28: Rephrase the sentences to be more precise on the result you use (if it's hourly or daily timescale). It is hard time following which are the $R^2$ you are presenting.

We agree with that and we will be more specific in our referencing to the timescales: P21.L26-28: *'Using the data collected in 2019 regression models for both location were developed. At the hourly timescale this resulted in $R^2 = 0.74$ and $R^2 = 0.71$ for Stavoren and Trintelhaven, respectively. Validation of these hourly models using the data collected during the summer of 2020 have shown that a simple data-driven model is able to explain large part of the hourly dynamics of open water evaporation ($R^2 = 0.84$ and $R^2 = 0.65$ for Stavoren and Trintelhaven, respectively).'*.

**Editorial comments**

- Some sentences lack of consistency and readability. This is often the missing punctuation that is in cause. For example, look at P2.L12, P7.L17, P7.L24, P9.L24, P12.L3.
We will correct this.

- Be attentive to have consistency in the form you write the units. The general form is to separate units with a point, ex : m.s-1 .
HESS prescribes to have the units written with a space and exponent, e.g. $W\ m^{-2}$.  We will check for consistency throughout the manuscript.

- « Focussing » should be written like this « focusing ».
Noted.

- P2.L8 : parametrise/parametrize and not « parameterise ».
Parameterize is widely used and also accepted by HESS. We will change to parameterize.

- P2.L26 : add a « to » → « and to represent ».
We will add this.

- P2.L33 and P7.L20: check the tense.
P2.L33: will change into: 'entered'
P7.L20: will change this into: 'we used'

- P4.L25 : use the English structure : 1,100 km²
Agreed.

-P5. Figure1 : Increase police size of your scale as it is not readable. Moreover, you should add the label on the contour lines. On the right figure you can erase the y-axis as it is the same than the center figure. Green and white colors for the center and right panels are not adapted to understand where are the land and the water. Add labels to the windrose.
Thank you for this feedback. We will adjust this figure according to your feedback.

- P7.L8 : you should either use co-variance or covariance but not both throughout you paper.
Noted.

- P8.L9 : « variable(s) » instead of « variable(x) ».
Thank you for spotting this. Will be adjusted.

- P10 Table 1 : Be sure all the parameters are aligned in the first column.
We placed the indent there as $T_{air}$, $T_{air,climatology}$, and $T_{water}$ are all belonging to the variable T.

- P12.L11: "are lacking"
Noted.

- Please consider improving your Venn diagram in order to gain in readability (police size).
The arrangement of all the intersections of the Venn diagram leaves less space for increasing the font size. We will do our best to gain readability.

---

## Author Comment (AC3)

**Response to referee comment Anonymous Referee #3**

We appreciate and would like to thank Anonymous Referee #3 for reading our manuscript and taking the time and effort to provide us with generally positive and constructive feedback. We will use this feedback to improve our manuscript. Please, find below our point-to-point response (comment of the referee in **black**, our response in **blue**).

**Review of HESS manuscript #hess-2021-549**
**Title: Evaporation from a large lowland reservoir – observed dynamics during a warm summer**

The manuscript analyses evaporation measurements by EC towers and provides simple regression models based on the routine observations to describe evaporation dynamics in Lake IJsell (Netherlands). It is well written and organized. However, there are some aspects in the study that need clarification or should be addressed by the authors.

• There are many studies showing that evaporation dynamics vary between different parts of inland water bodies arising from, for example, spatial variability of water depth or meteorological inputs (boundary conditions). Considering this, it is fine to have two different measurement stations at the middle of the water body (Trintelhaven) and at the border (Stavoren). However, a discussion on the effect of EC towers location on the observed evaporation dynamics from the water body is missing (not its comparison with terrestrial evaporation as mentioned in section 2.2).

This is a good point that is raised. We will add this to the discussion section of the manuscript.

• Vapor pressure at the air-water interface was estimated based on the surface temperature obtained from satellite imagery that often show biases. Was this checked?

Specifications of the Meteosat sub-skin water temperature product describe a bias of 0.5 Kelvin, where validation takes place with temperature measurements from drifting buoys. We argue that this small bias will not change our findings of the most important drivers (i.e. wind speed and vapour pressure gradient) of $E_{water}$. If at all, the bias in the satellite product might lead to slightly different observed dynamics in water temperature and thus vapour pressure, which potentially could lead to changes in the regression coefficients. However, we think this effect will be insignificant.

• From our own measurements of vertical water temperature in a shallow basin with 2 m depth, I can say water temperature at the surface, where evaporation takes place, is completely different with even 10 cm below. Thus I am not sure how water temperature at depth of 1.2 or 1.5 m could help for evaporation analysis, unless you have a temperature model to reproduce surface temperature.

We agree that indeed preferably the skin temperature of the water surface is used in the estimation of evaporation rates. That is why we have used the satellite product in the first part of our regression analysis. Additional to that, we were interested if other routinely observations in the Netherlands which are easily accessible would provide another reasonable source from which evaporation rates could be estimated. The good correlation ($R^2 = 0.71$ and $R^2 = 0.94$; based on summer 2019 at Stavoren and Trintelhaven, respectively) between sub-skin water temperature from the satellite product and water temperature measured at depth of 1.2 or 1.5 m gave rise to that.

• It is not surprising that neglecting thermal inertia of the water body (indicating the effect of radiation adsorption in depth) could make such considerable bias in the performance of models such as Penman's (see Friedrich et al. 2018: DOI:10.1175/BAMS-D-15-00224.1). Please see Section 2.4 of Zhao et al. 2020 (https://doi.org/10.1016/j.rse.2020.112104) accounting for the impact of G on evaporation estimates by Penman-type approaches.

As a response to the role of G on evaporation, we would like to refer to our response to specific comment 3 of referee #1.

• Why air temperature is not included in the analysis of section 3.4?

We did not include air temperature as a separate variable here because we chose to include its effect in the vertical vapour pressure gradient, where air temperature is integrated in.

• I believe radiation is a key component in shaping surface temperature that, in turn, defines vapor pressure gradient at the core of your simple regression model. In application of Eq. 10, how surface temperature is obtained? From measurements at depth of 1.5 m, satellite imagery, or solving for energy equation? Dalton-type models may look simple in representation but have difficulties associated with obtaining reliable surface temperatures, and of course wind function (especially in the context of the projected climate change).

We agree that radiation is shaping surface temperature, and thus affects the vapour pressure gradient. In turn, evaporation decreases surface temperature and affects vapour pressure gradient as well. They are interdependent. To answer to your question about equation 10: We did not include the Dalton model in our analysis in the form presented as equation 10. We have placed it her in the discussion section to make a comparison between the functional form of the Dalton model and the regression model that we have found. To be complete, in figure 3 where we did present the monthly average diurnal cycle of, amongst others, the Dalton model. There, we have used the temperature data originating from the satellite product to calculate the vapour pressure gradient. We will add about the difficulties associated with the Dalton model to the discussion section.

---

## Author Response (AR1)

**Response to comments from Anonymous Referee #1, #2, and #3, and editor combined**

We appreciate and would like to thank again all anonymous referees for providing their valuable feedback. In addition, we would like to thank the response of the editor to our rebuttal. In this document we compiled all of our final responses with our response to the editor's feedback woven into it. It includes references in green to where changes have been made in the revised manuscript (note that the feedback of referee #2 is pointing to the previous version of the manuscript before revision). A track-change document is added as well. Note that in the revised manuscript some of the figures and tables have been changed as well but is not always visible from the track-change document.

Please, find below our point-to-point response (comment of the referee/editor in **black**, our response in **blue**).

**Specific comments by anonymous referee #1**

1. Page 6 line 18 says that the data were collected during the summer of 2019 and 2020. Section 3.1 describes the large fraction of time that lake fluxes were unavailable or of limited quality. My impression is that these statistics refer to 30-minute averaging periods. If I understand this correctly, it is not clear how daily, monthly, and yearly data were obtained from these data. How were gaps filled? In particular, how would long periods of time with flux footprints falling outside the lake be handled? While the paper is generally clearly written, more attention is needed to specify the time-scales for the various figures, tables, and in-text statistics.

   We are sincerely thankful for the remark of the referee on how we have dealt with the low number of remaining data after filtering (on quality, wind direction). Your remark actually made us go back to the scripts, where we found a small mistake in one of the programming commands which let all daily averages be calculated even if just one hour would be present. This of course was not our intention. We now correctly calculate daily averages only if for at least 66% of the hours (two-third) in a day valid data are available. Of course this leads to less daily data being available and therefore we had to redo some of the analyses, in particular the regression analyses. At a daily timescale the regression model is based on only a few data points (in the order of 12), but it is still found to be statistically significant (p-values < 0.01). The conclusion that in most cases a combination of wind speed and vertical gradient of vapour pressure forms the driving force of LE did not change. Finding the mistake thanks to the comment of the reviewer shows us that the review process is really helpful and needed. A good reminder and learning moment to me and us.

   We intentionally did not filled large gaps (>1 hour) in the data. This would create synthetic results interfering with our aim to perform a process-oriented study into the role of evaporation in the surface-atmosphere coupling. We only performed linear interpolation to the data when gaps of at the most one hour occurred.

   **Additional comment by editor:**

   In your response to Comment (1) of Referee #1, you state that in the hourly data, data gaps of at most one hour were linearly interpolated. For a half-hourly dataset, this means that gaps of a maximum of two consecutive data points were linearly gap filled, right? Perhaps this would be a clearer way of describing the process. On the other hand, I am still wondering why hours with no data at all (i.e. one hour data gap) were gap-filled at all. Doesn't this create the synthetic data

you wanted to avoid? For the daily averages, would it not be better to calculate them directly from the half hourly data set and describe how you dealt with gaps separately from the generation of hourly data? The criterion of at least 66% valid hourly data could otherwise be reached with in fact 33% valid half-hourly data. In addition, is the criterion of 66% valid diurnal data sufficient to calculate reliable daily averages? For example, if the 66% of the data was measured predominantly during daylight, the average would be quite different to a day where 66% of the data was measured predominantly during the night, even if the total daily evaporation was the same. It might be good to check if the data gaps were random or clustered.

We agree with you that it is better to not fill the gaps larger than half an hour and to aggregate the daily data directly from the half-hourly dataset. We have adjusted our analysis accordingly and now only performed linear interpolation to the data when gaps of at the most half an hour occurred.
We will add the following section to the manuscript at **P7.L26**: "*Within this dataset, gaps of maximum one data point (i.e. half an hour) were linearly interpolated. Larger gaps were intentionally not gap filled, because this would create synthetic results interfering with our aim to perform a process-oriented study. Hourly data were obtained by aggregating the half-hourly dataset with no further gap filling actions taken. Daily averages were calculated from the half-hourly dataset only if valid data were available for at least 66% of the time.*"

We have added the following to **P11.L3**: "*No clustering has been found in the availability of latent heat flux data during daylight hours (6hr – 21hr) compared to the night (21hr – 6hr). In the final dataset in Stavoren, latent heat flux data were available during 56% of the total daytime half-hours in the summer of 2019, and 49% during night-time. In the summer of 2020 this was 49% and 40% for daytime and night-time, respectively. For location Trintelhaven the corresponding fractions were 10% and 18% during daytime and night-time in the summer of 2019, respectively. The difference between daytime and night-time was smaller during the summer of 2020, with 18% and 21% of data available, respectively.*"

In the regression analysis we have only used hourly and daily data. Figures 4 and 9 show monthly averages based on hourly data. The uncertainty bars provide an indication of the ratio between the standard deviation of the hourly observations and the square root of the number of hourly observations taken into account in the calculation of the monthly mean, as a measure of the sampling uncertainty of the estimated mean.

2. When I read that Penman's equation does not work for these data, I was surprised. I thought the authors were claiming that they physics behind the equation were incorrect. But I think the authors agree with the physics of the Penman equation. They simply cannot determine the available energy with any reasonable certainty for this water body. Do I understand this correctly? Read our response at point 3 of referee #1.

3. Maybe this is just a re-wording of the previous comment, but the energy input, heating/cooling of the lake air stability, etc. do determine the fluxes. The temperatures and humidities at various heights and depths adjust according the these principles. The point of the paper is to determine which easily-measured variables give the best access to the fluxes.

Thank you for your remarks on the disagreement of our results in relation to the Penman equation. In response to your general and specific comments that target the use of Penman's equation we would like to argue the following:
Lake IJssel is a shallow lake of several metres deep and therefore most probably the heat capacity could indeed not be neglected. Penman developed his theory originally for land surfaces

and (infinitely) shallow water surfaces. Therefore, he could make the assumption that the net radiation was divided over the turbulent fluxes and the ground heat flux exactly at the surface-atmosphere interface. This then determines the surface temperature and surface humidity, and eliminates surface temperature from the equation (P3.L2-3). However, in a water body part of the radiation penetrates the water, delivering relatively large amounts of energy below the air-water interface directly. This affects the physical processes in a way that cannot be described with the PM equation. The large heat capacity of a water body provides the system with a "memory". As a result, the water temperature at the surface is not directly related to the instantaneous energy balance at the surface, which is how Penman's equation can be interpreted, but rather it is subject to a delay following the large heat capacity of the water body (P20+P21, L10-11+L1). Using a simple energy balance model of a water layer could help to solve the water temperature for the next time step (Equation 11 from Keijman (1974), and eq. 10 from De Bruin (1981)). However, in our analysis we have not measured $G$ based on temperature changes integrated over the volume of the water column. In other words, we have omitted the downward heat flux $G$ in the calculation of $Q*$ to adhere to the original Penman theory (P9, L 17-19). All the results of the Penman model presented in this manuscript are therefore based on the original theory of Penman for land surfaces and (infinitely) shallow water surfaces where $G$ is neglected. We think that the Penman model does not present the full story in case of a water body of several meters deep.

**Additional comment by editor:**

In your response to Comment (3) of Referee #1, you propose to write: "As a result, the water temperature at the surface is not directly related to the instantaneous energy balance at the surface, which is how Penman's equation can be interpreted, but rather it is subject to a delay following the large heat capacity of the water body. We think that the Penman model does not present the full story in case of a water body of several meters deep." I think this is misleading, as Penman (1948) did explicitly address the problem of heat storage, stating: "Over a period of several days, and frequently over a single day, the change in the stored heat, S, is negligible". This means that he did not claim that his model should necessarily be useful at the daily scale, and certainly not at the hourly scale. This is omitted on P9L15 in the manuscript where you stated that Penman's derivation assumes negligible G for shallow water surfaces. As the reviewer stated, the time scales of integration are paramount here and it is not clear enough in the article at what time scales each model can be used. Please clarify.

We agree that the effect of heat storage takes place over larger timescales (rather than hourly or daily). This is in agreement with the quote of Penman (1948) that you mention here. In addition, the effect of heat storage is visible in figure 3 of our manuscript, which shows a lag of one month between global radiation and observed $E_{water}$. This lag means that the change in heat storage should be integrated over larger timescales. However, this seems to contradict how the Penman equation is typically employed, assuming instantaneous energy balance at the surface.

We have changed the text into (**P23.L3**):
'*The large heat capacity of a water body, controlled by the depth of the water column, provides the system with a 'memory'. As a result, the water temperature at the surface is not directly related to the instantaneous energy balance at the surface, where net radiation is divided over the turbulent fluxes and a water heat flux. Rather, water temperature is subject to a delay and results from heat storage integrated over longer timescales.*'

And (**P10.L13**):

*'According to Penman's derivation G is assumed to be negligible for short timescales of a day to several days for shallow water surfaces, similar to land surfaces, and the term is often ignored because of the difficulty of measuring it. However, for water bodies of several metres deep the impact on the energy balance by neglecting G can be considerable (Keijman, 1974; de Bruin, 1982; Tanny et al., 2008; van Emmerik et al., 2013). For these water bodies G should be considered as a result of temperature changes integrated over the volume of the water column in contrast to a land surface where the impact of G is more superficial. It should be clearly noted that, although Lake Ijssel is a lake of several metres deep, we have neglected G in the following analyses because (1) we have not been able to measure it, and (2) in order to adhere to how Penman's equation is typically employed for shallow water surfaces.*

4. In figure 5 in particular, are the diagrams only for the summer months?

   Thank you for this remark. Yes, this diagram is based on only the summer months. In the revised version of the manuscript this is indicated in the caption of this figure, as well as for other figures where this applies.

5. Page 16 lines 4-6. Typically, if measurements have a restricted range of variability, this results in a smaller R2 value, because random fluctuations are large relative to the observed changes. It looks like the authors are claiming the opposite effect here. Please explain.

   R2 values are determined by the variances of both data series, and the covariance between those series. So we agree that our explanation: 'higher variance will lead to lower R2 values', is not covering the whole story. We agree with the reviewer that indeed measurements with a restricted range of variability would result in smaller R2 values given the measurement uncertainties, which are not correlated.

   In an attempt to understand why sometimes the R2 values are higher for the validation period, we have swapped the calibration (now summer of 2020) and validation (summer of 2019) period. The coefficients of the regression model were re-calculated. Now, we find R2 values of the validation to be smaller than the calibration period. This provides an indication that the difference in R2 values seem partly related to the conditions during the two distinct summer periods. Randomly selecting a calibration and validation period from the two summers combined, results in lower R2 values during the validation period as well.

   **P17.L16**: *'In an attempt to explain why a higher $R^2$ value occurs during the validation of the model, we have swapped the calibration (now summer of 2020) and validation (summer of 2019) period. The coefficients of the regression model were re-calculated. The $R^2$ value of the validation was now found to be smaller than during the calibration. This provides an indication that the difference in $R^2$ values during calibration and validation seem partly related to the conditions during the two distinct summer periods, and it gives confidence that the model performs well.'*

6. In Figure 5 d, the central intersection has an R2 value smaller than some of the other intersections. How could adding a variable explain LESS variability than simply not including it?

   This is because the numbers that are depicted in the Venn diagrams indicate the adjusted R2 to study the fit of the model as is mentioned in the caption of the figure. The adjusted R2 takes into account the degrees of freedom and can therefore lead to a decrease in adjusted R2 if the added variable only slightly correlates with the dependent variable.

We added the following to **P14.L3**: "... *the colour is. Adding a variable will not always result in a higher adjusted R2 value, because the adjusted R2 takes into account the degrees of freedom. Therefore, it can lead to a decrease in adjusted R2 if the added variable only slightly correlates with the dependent variable. Venn diagrams (a) and (c) .....*"

**Additional comment by editor:**

In your response to comment (6) of Referee #1, you explain the meaning of the adjusted coefficient of determination. When reading this, I realised that it is not mentioned in Section 2.5 how the regression analysis was performed. Could you please add this information, referring to the R-packages used in this analysis and explaining what metrics were used and how they were calculated? On P8L13, you mention that you aim to "find the best simple model that uses maximum two variables, while still able to explain the dynamics of E water well." This might be a good place to add details about the metrics and how they were used to achieve this goal.

We will add the following to **P8.L18**: *'The 'leaps' package in R has been used to identify the best regression model, where the residual sum of squares was used as a metric to find the best model given the predictors.'*

7.  Throughout the paper I kept wondering why lake water stability was not included. Surely there must be seasonal changes in stability and thus of mixing depth within the lake. Did this have an impact on the data?

We agree that lake water stability would be interesting to consider since it affects the surface temperature and therefore evaporation rates through the vapour pressure gradient. Evaporation in turn has a cooling effect on the surface temperature (which increases potential mixing). However, we unfortunately did not have the opportunity to measure water temperatures at several depths to study this. A preliminary study performed by one of our master students simulated mixing depths using the model FLake, which showed that in 70% of the time Lake IJssel is fully mixed. During summer it is of course more likely that stable conditions occur and we cannot directly assume fully mixed conditions. However, we considered this phenomenon beyond the scope of the current study.

We added the following to **P21.L18**: "*Another phenomenon that could affect the yearly cycle of evaporation is lake water stability and thus mixing depth within the lake. Seasonal changes in lake water stability affects the surface temperature and therefore evaporation rates through the vapour pressure gradient. Evaporation in turn has a cooling effect on the surface temperature, which increases potential mixing. Supported by a preliminary study where mixing depths were simulated using the model FLake (Voskamp, 2018), we assume that during 70% of the time Lake IJssel is fully mixed. This number is not surprising given the fact that during night time evaporation continues, and with wind speeds that are on average 5.8 m.s$^{-1}$. In addition, the inflow of the river IJssel into the lake is likely to support mixing as well. During summer it is more likely that stable conditions occur and we cannot directly assume fully mixed conditions. However, we considered this phenomenon beyond the scope of the current study.*"

Voskamp, T.: The evaporation of Lake IJssel - Comparison of the FLake model with standard methods at multiple timescales for estimating evaporation rates, MSc thesis report, Wageningen University, Wageningen, the Netherlands, 2018.

**Specific comments by anonymous referee #2**

**Abstract**

General comments : In this section, the objective of the paper is not clearly pointed out. In my opinion, you should focus on the specific parametrisation you proposed for Lake Ijssel based on the field measurements. All the elements are already written and you just need to rearrange the section.

We agree and have changed the abstract as follows **(P1.L1)**:
*"We study the controls on open water evaporation of a large lowland reservoir in the Netherlands. To this end, we analyse the dynamics of open water evaporation at two locations, i.e. Stavoren and Trintelhaven, at the border of Lake IJssel (1,100 km$^2$) where eddy covariance systems were installed during the summer seasons of 2019 and 2020. These measurements were used to develop data-driven models for both locations. Such a statistical model is a clean and simple approach that can provide a direct indication and insight of the most relevant input parameters involved in explaining the variance of open water evaporation, without making a prior assumptions on the process itself. This way, we find that a combination of wind speed and the vertical vapour pressure gradient can explain most of the variability of observed hourly open water evaporation. This is in agreement with Dalton's model which is a well-established model often used in oceanographic studies for calculating open water evaporation.*
*Validation of the data-driven models demonstrates that a simple model using only two variables yields satisfactory results at Stavoren, with R$^2$ values of 0.84 and 0.78 for hourly and daily data, respectively. However, the validation results for location Trintelhaven fall short (R$^2$ values of 0.67 and 0.65 for hourly and daily data, respectively). Using only routinely measured meteorological variables leads to adequate performing simple models at hourly (R$^2$ = 0.78 at Stavoren, and R$^2$ = 0.51 at Trintelhaven) and daily (R$^2$ = 0.82 at Stavoren, and R$^2$ = 0.87 at Trintelhaven) timescales. These results for the summer periods show that global radiation is not directly coupled to open water evaporation at the hourly or daily timescale, but it rather is a combination of wind speed and vertical gradient of vapour pressure. We would like to stress the importance of including the correct drivers of open water evaporation in the parametrization in hydrological models to adequately represent the role of evaporation in the surface-atmosphere coupling of inland water bodies.*

-P1.L1 : I would talk about a « sink » rather than a « large loss term ». It is more adequate to the scientific level of the journal.

This sentence is removed in the newly proposed abstract. But we agree with the feedback and we adjusted this in the introduction section **P1.L22** to '*Evaporation is a sink in the water balance of inland water bodies.*'

-P1.L1 : « During summer seasons, which are projected to become warmer with more severe and prolonged periods of drought ». This is a general sentence whereas your study focused on a specific location. Even if we are on a global climate change path, the consequences (not specifically warmer summer) are not the same worldwide. You should be more specific on the spectrum of warming on the studied area (or region) and put references.

This sentence has been removed in the newly proposed abstract, but in the introduction section we are now more specific on the region that we are referring to, i.e. the Netherlands. In addition to the

already mentioned references in the introduction targeted to mid-latitude regions in Europe, we refer to the KNMI'14 climate scenario's for the Netherlands specifically.

**P2.L5**: *'Summer seasons are projected to become warmer in the Netherlands, with more severe and prolonged periods of drought (Seneviratne et al., 2006, 2012; KNMI, 2015; Teuling, 2018; Christidis and Stott, 2021).'*

KNMI, 2015: KNMI'14-klimaatscenario's voor Nederland; Leidraad voor professionals in klimaatadaptatie, KNMI, De Bilt, 34 pp

- P1.L8 : « not available energy» Be specific on the type of energy.

With available energy we mean $R_n$ – G (P9.L4), but we agree that in this context this may not have been clear. We have removed the term from the abstract as it does not support clarity of the sentence and the statement that we make there.

-P1.L11 : « main drivers » Be specific. What type of phenomenon they are the drivers of?

We agree we should be more specific here. However, in the revised abstract this sentence has been removed.

- P1.L15 : « well performing simple data-driven models » I would be less enthusiastic with a R² of 0.51 and 0.43. The model is adequate but does not perform well.

We have added the required nuance of our statement. **P1.L12** – *'Validation of the simple models that are using only routinely measured meteorological variables perform adequately at hourly ($R^2$ = 0.78 at Stavoren, and $R^2$ = 0.51 at Trintelhaven) and daily ($R^2$ = 0.82 at Stavoren, and $R^2$ = 0.87[1] at Trintelhaven) timescales.'*
[1] Note that the numbers have changed after correcting a mistake (see our response to point 1 of referee #1)

**Introduction**

- P2.L1 : There is a more up-to-date review paper you should include: Woolway, R. I., Kraemer, B. M., Lenters, J. D., Merchant, C. J., O'Reilly, C. M., & Sharma, S. (2020). Global lake response to climate change. Nature Reviews Earth & Environment, 1(8), 388-403.
I would suggest to have a look at the paragraph about lake evaporation which gives essential materials for both your introduction and your discussion.

Thank you for the suggestion for this interesting paper. We have added the reference of Woolway et al. (2020) to **P1.L20**: *'Inland water bodies are known to interact with the local, regional and even global climate and are therefore highly sensitive to climate change (Adrian et al., 2009; Liu et al., 2009; Wang et al., 2018; Woolway et al., 2020).*

We have adjusted and added the following to **P1.L23**: *'... how open water evaporation ($E_{water}$) will respond to these changing conditions. It is expected that changes in longwave radiation, Bowen ratio, ice cover, and stratification will affect the dynamics of $E_{water}$ at the long-term (Wang et al., 2018; Woolway et al., 2020). Whereas at the shorter decadal timescale, a contribution to trends and variations in $E_{water}$ is expected resulting from changes in wind speed, humidity, and also through global and regional solar dimming and brightening and its effect on water surface temperature (Desai*

*et al., 2009; McVicar et al., 2012; Schmid and Köster, 2016; Wang et al., 2018; Woolway et al., 2020). During the summer season evaporation rates are...'*

Desai, A.R., Austin, J.A., Bennington, V. & McKinley, G.A. Stronger winds over a large lake in response to weakening air-to-lake temperature gradient. Nat. Geosci. 2, 855-858 (2009)

McVicar, T.R. et al. Global review and synthesis of trends in observed terrestrial near surface wind speeds: Implications for evaporation. J. Hydrol. 416–417, 182–205 (2012)

Schmid, M. & Köster, O. Excess warming of a Central European lake by solar brightening. Water Resour. Res. 52, 8103–8116 (2016)

Wang, W. et al. Global lake evaporation accelerated by changes in surface energy allocation in a warmer climate. Nat. Geosci. 11, 410–414 (2018)

Woolway, R. I., Kraemer, B. M., Lenters, J. D., Merchant, C. J., O'Reilly, C. M., & Sharma, S. (2020). Global lake response to climate change. Nature Reviews Earth & Environment, 1(8), 388-403.

We have added a sentence at **P4.L9** (see specific comment on P4.L3): '*Adequate estimations of $E_{water}$ are important in this context because there is a strong coupling between $E_{water}$ and for instance lake level and extent, the lake ecosystem, and lake stratification and mixing regimes (Woolway et al., 2020; Jenny et al., 2020).*'

In addition, we have added the following at **P19.L11** (see specific comment to P14.L22): '*This is similar to what has been found by studies of for instance Blanken et al. (2011) and McGloin et al. (2014), and it was noticed that intraseasonal variations of $E_{water}$ can be linked to synoptic weather variations through these variables (Lenters et al., 2005, MacIntyre et al. 2009, Liu et al., 2011, Woolway et al. 2020).*'

- P2.L2 : « evaporation is a large loss term of water bodies ... » I would rephrase by saying this is « a sink in the lake water balance ». Also you should add a reference to justify this, even if it's a validated fact.

Thank you for the suggestion. We have changed the terminology (**P1.L22**). We would like to argue that the fact that evaporation is a sink in the water balance of a lake can be referred to as common knowledge, and therefore does not need a reference.

- P2.L7 : « Summers are projected to become warmer » As mentioned for the abstract section, as you work on a specific location, you should be more specific on such fact as you are not working at global scale (mention the spatial scale). Moreover, I would recommend to give the climate reference on which the climatic comparison is made. You should also add a reference.

We agree that we should be more specific on the region. Therefore, we have changed the sentence to (**P2.L5**): '*Summer seasons are projected to become warmer in the Netherlands, with more severe and prolonged periods of drought (Seneviratne et al., 2006, 2012; KNMI, 2015; Teuling, 2018; Christidis and Stott, 2021).*

KNMI, 2015: KNMI'14-klimaatscenario's voor Nederland; Leidraad voor professionals in klimaatadaptatie, KNMI, De Bilt, 34 pp

- P2.L12-13 : « In terms of thermodynamics, ...» This works on some lakes but this is not always correct. In terms of thermodynamics, big lakes (such as the American or African Great Lakes) could

be approach by using either 3D ocean model or 1D model. It will depends on the presence or not of the hypolimnion and furthermore on the stratification dynamic, if there is one.
See :

*Xue, P., Pal, J. S., Ye, X., Lenters, J. D., Huang, C., & Chu, P. Y. (2017). Improving the simulation of large lakes in regional climate modeling: Two-way lake–atmosphere coupling with a 3D hydrodynamic model of the Great Lakes. Journal of Climate, 30(5), 1605-1627.*
*Gronewold, A. D., & Stow, C. A. (2014). Water loss from the Great Lakes. Science, 343(6175), 1084-1085.*
*Thiery, W. I. M., et al. "LakeMIP Kivu: evaluating the representation of a large, deep tropical lake by a set of onedimensional lake models." Tellus A: Dynamic Meteorology and Oceanography 66.1 (2014): 21390.*

We agree that the statement we make here is not applicable to all inland water bodies and does depend on stratification and should be better focused on our target water body.
Therefore, we have changed it to (**P2.L11**): '*In terms of thermodynamics a shallow inland water body of only a few meters deep*'

- P2.L17 : This might be a detail but I would prefer to talk about a change of the amplitude (which suppose an increase of surface temperature during daytime but also a quicker decrease during night-time).

Yes, we are referring to the same process, using other words. We have changed the sentence to (P.L): '*…., where heat is stored in the lower atmosphere, vegetation and the upper soil layers. This leads to larger temperature amplitudes in sunny conditions, with strongly increasing surface temperatures and warming of the lower atmosphere during daytime, and strong decreases during night-time.*'

- P2.L21: Lake depth also controls the dynamical range of lake temperature amplitudes on diurnal timescale.

See also our response to point 3 made by referee #1 on the simulation of water temperature using a simple energy balance model of a water layer.
We have changed the text at **P2.L24** into: '*The subsurface energy budget implies that Lake depth controls the dynamical range of lake temperature amplitudes on diurnal timescale. Thus, instead of focussing at the surface only, rather the whole volume of the system should be considered.*'

- P3.L2-6 : This paragraph would gain in readability if you reduce the description to its essential. Penman equation is well-known and its description can be shortened. Moreover, this description is redundant with the one on P9.

We agree that the Penman equation is well-known, but we also think that it is important to stress the assumptions that are made by Penman (i.e. assuming energy storage below the surface to be neglected over a period of a several days, resulting in an instantaneous response of the surface temperature), which makes it less straightforward to use it for water bodies of a few meters deep. Moreover, it specifies the essential difference between the Penman and the Dalton equation. Overall, we think the balance in the introduction is good (i.e. 1 paragraph dedicated to the description of the Penman and the Dalton model). Therefore, we prefer to keep the text as is.

**Additional comment by the editor**
Your response to Reviewer #2's comment on P3.L2-6 prompted me to re-read that paragraph and I believe that the following sentence is inaccurate: "This assumption results in the essential difference between the models of Dalton and Penman, where Dalton uses the vertical gradient of vapour

pressure, while Penman uses the vapour pressure deficit at 2 metres height." Firstly, Dalton (1802) did not speak about gradients, only differences. Secondly, the main difference introduced by Penman was that the surface temperature needed to calculate saturation vapour pressure at the surface was replaced by the need to account for the net radiation and heat exchange between the surface and the water body. The assumption that the latter is negligible is NOT integral to the Penman equation, as the equation can equally be applied if that component is measured. I agree with you, though, that the explanation of the Penman equation should not be left out, as suggested by the referee, as the background and underlying assumptions are not immediately present in the readers' minds, even if the equation is well known.

We agree that Dalton speaks about differences instead of gradients. We have corrected that throughout our manuscript.

In response to the second point you have raised: Penman assumed that temperature and vapour pressure at a reference height could replace the surface temperature through linearisation of the vapour pressure equation (Brutsaert, 1982). That is the main assumption made by Penman. If observations of surface temperature are available, one would end up with Dalton's equation. If the net radiation and heat exchange between the surface and the water body ($R_n$ – G) could be measured, it would not replace observations of surface temperature.

- P3.L7 : « Most studies ... » Is this sentence linked with the reference list starting on L9 ? If so, you should move the sentences « However, measurements of... » and « This can partly ... » elsewhere. Also, you said the contrary on L.26 "In the past, a number of studies reported ..." Moreover, I would not be that direct by saying measurements of evaporation from inland water bodies are under-represented, numerous studies have been published on the subject for the past 10 years:

*Potes, M., Salgado, R., Costa, M. J., Morais, M., Bortoli, D., Kostadinov, I., & Mammarella, I. (2017). Lake–atmosphere interactions at Alqueva reservoir: a case study in the summer of 2014. Tellus A: Dynamic Meteorology and Oceanography, 69(1), 1272787.*
*Pillco Zolá, R., Bengtsson, L., Berndtsson, R., Martí-Cardona, B., Satgé, F., Timouk, F., ... & Pasapera, J. (2019). Modelling Lake Titicaca's daily and monthly evaporation. Hydrology and Earth System Sciences, 23(2), 657-668.*
*Moigne, P. L., Legain, D., Lagarde, F., Potes, M., Tzanos, D., Moulin, E. R. I. C., ... & Costa, M. J. (2013). Evaluation of the lake model FLake over a coastal lagoon during the THAUMEX field campaign. Tellus A: Dynamic Meteorology and Oceanography, 65(1), 20951.*
*Blanken, P. D., Spence, C., Hedstrom, N., and Lenters, J. D.: Evaporation from Lake Superior: 1. Physical Controls and Processes, Journal of Great Lakes Research, 37, 707–716, https://doi.org/10.1016/j.jglr.2011.08.009, 2011*

Thank you for bringing up these references, we have included some of these in the study to integrate our work more into the current scientific literature (**P3.L31; P22.L7**).
Assuming that there is referred to the reference list at L12, it is not correct that these belong to our statement in L7 '*Most studies...*'. We therefore preferred to keep that as is.
However, we agree that our statements in L7 and L26 could read as *contradictory*, but what we mean here is that comparably there have been a lot more studies focussing on the understanding of terrestrial evaporation, and much less studies focussed on open water evaporation. Fortunately, there have definitely been studies that measured and modelled open water evaporation (also shown by the references provided by you). To avoid confusion, we have changed the sentence (**P3.L9**) to: '*However, comparably significantly less studies performed measurements of E$_{water}$ from inland water bodies*.'

- P3.L27 : I'm not convinced about the utility of the brackets. Moreover you could also include other important hydroclimate variables.
*Zhou, W., Wang, L., Li, D., & Leung, L. R. (2021). Spatial pattern of lake evaporation increases under global warming linked to regional hydroclimate change. Communications Earth & Environment, 2(1), 1-10*

Agreed on the brackets; we have removed them.
The coupling between lake evaporation and hydroclimate (P-E) as referred to in Zhou et al. (2021), is found on larger timescales (decades), and a link is made to changes in lake evaporation under different climate scenarios. It therefore provides another concept to describe lake evaporation, but at another timescale than the other studies that we are referring to in these lines (P3.L29-31). We have added the following (**P3.L33**): *'At larger timescales a spatial coupling was found between $E_{water}$ and $P-E_{terrestrial}$ (Zhou et al., 2021). Jansen and Teuling (2020) studied the performance of a number of concepts that are commonly used to describe open water evaporation.'*.

- P3.L34 →P4.L2: Woolway et al 2018 & Wang et al 2018 have addressed this issue. Moreover, even if I agree with your assumption, I'm not convinced about such parametrisation for use at longer timescale (for example, seasonal timescale). Lake temperature and evaporation are interdependent on such timescales and other hydroclimate variables should also be included.

*Woolway, R. Iestyn, et al. "Geographic and temporal variations in turbulent heat loss from lakes: A global analysis across 45 lakes." Limnology and Oceanography 63.6 (2018): 2436-2449.*
*Wang, Wei, et al. "Global lake evaporation accelerated by changes in surface energy allocation in a warmer climate." Nature Geoscience 11.6 (2018): 410-414.*

Thank you for the additional references. Our statement about the disagreement of the methods on the average increasing historical trend of the evaporation rate, as well as for the projected future trends is based on the findings in our previous study (Jansen and Teuling, 2020). That study showed that the choice of method, with different representations of the evaporation process, can lead to significantly different projected trends. We do agree with the point you raise here that lake temperature and evaporation are interdependent on these longer timescales, which requires the water body energy balance to be represented correctly. That observation actually supports our statement that it is important to find a way to correctly represent the evaporation process for the timescale that is studied, i.e. hourly and daily.

We have made the following adjustment **P4.L3**:
*'At longer timescales (i.e. seasonal and yearly timescales) it is important to include the interdependency between lake temperature and evaporation. This requires a concept in which the water body energy balance to be represented adequately, for the correct modelling of the $E_{water}$ process.'*

- P4.L3 : This sentence is the key point of your study. More generally, the paragraph from L3 to L14 should be the core of your introduction. I would reduce the presentation of the different equation (Penman, Makkink) and enrich this paragraph.
Add a reference for this : « a crucial element in its water management system ». Also, you focus on the water management aspect however your paper does not specifically study the impact of the parametrisation on the lake hydrology. I would recommend to add other aspects of the evaporation as a component of the global energy and water cycle. For example, you can talk about the influence on the near-surface turbulence intensity, the stratification or the lake ecosystem.

*Raymond, P. A., and others. 2013. Global carbon dioxide emissions from inland waters. Nature503:355–359. doi:10.1038/nature12760*
*Jenny, Jean-Philippe, et al. "Scientists' warning to humanity: rapid degradation of the world's large lakes." Journal of Great Lakes Research 46.4 (2020): 686-702.*

It is correct that this last paragraph of our introduction includes the aim of our study. We brought it into context of previous studies and core concepts that have been used, and are still frequently used, in calculating evaporation (Penman, Makkink, Dalton). So we think it is important to introduce that as well in this section. We agree to add a reference to 'a crucial element in its water management system' (*Buitelaar et al., 2015*). In addition, this study was performed to bring the importance of the correct parametrization of $E_{water}$ in context of lake hydrology. Especially because, as you mention, there is a strong connection between $E_{water}$ and e.g. lake level and extent, lake ecosystem, and lake stratification and mixing regimes (Woolway et al., 2020). Through these connections, $E_{water}$ affects the water management of the lake in terms of drinking water services and water availability for agricultural land (Jenny et al., 2020).

We have added the following text to **P4.L8**:

*'....in its water management system (Buitelaar et al., 2015). Adequate estimations of $E_{water}$ are important in this context because there is a strong coupling between $E_{water}$ and for instance lake level and extent, the lake ecosystem, and lake stratification and mixing regimes (Woolway et al., 2020; Jenny et al., 2020).'*

Woolway, R. I., Kraemer, B. M., Lenters, J. D., Merchant, C. J., O'Reilly, C. M., & Sharma, S. (2020). Global lake response to climate change. Nature Reviews Earth & Environment, 1(8), 388-403.
Jenny, et al. "Scientists' warning to humanity: rapid degradation of the world's large lakes." Journal of Great Lakes Research 46.4 (2020): 686-702.
Buitelaar, R., Kollen, J., Leerlooijer, C. (2015). Rapport Operationeel waterbeheer IJsselmeergebied - Inventarisatie huidige waterbeheer IJsselmeergebied door Rijkswaterstaat en Waterschappen. Report. 112 pp. Grontmij. Alkmaar.

- P4.L12-15 : In this sentence, you compare Makkink's equation with Flake simulations. I'm wondering why you are not using FLake directly for lake IJssel if you consider Flake simulations as your reference ?

We agree that in this sentence it seems that we treat Flake as a reference. However, the main focus of our paper is to develop a simple statistical model from measurements, which can provide a simple and clean solution. So we use the measurements as a reference. To our knowledge Flake was never tested before against EC measurements for Lake IJssel. The comparison is made because Flake is a physically-based model, which is also integrated in the ECMWF model for instance, and we therefore assume it will perform better than the Makkink equation at these timescales, while the latter is currently used for estimating evaporation from Lake IJssel, so that is where our interest lies.

We have changed the sentence into (**P4.L19**)*: 'Makkink's equation is not able to capture the dynamics of $E_{water}$ when compared to what is found by aforementioned observational studies on $E_{water}$ and compared to estimations with physically-based lake models such as FLake (Jansen and Teuling, 2020)'*

**Data, Material and Methods**

- P6.L8: It seems that the KNMI station only measure global radiation (as I see in the data provided), however would it be possible to have access to the four components of the global radiation? As shown in Wang et al 2018, the incoming radiation has an effect even if it's at longer timescale.
*Wang, Wei, et al. "Global lake evaporation accelerated by changes in surface energy allocation in a warmer climate." Nature Geoscience 11.6 (2018): 410-414.*

The assertion is correct: the operational KNMI stations only provide global radiation data (i.e. incoming solar radiation). Thanks to your remark here however, we did notice we have not explained how we obtained net radiation from the meteorological data. We have added this to section 2.5, where the Penman equation is explained. At longer timescales global radiation indeed will affect evaporation rates, that is why we also included this variable in the regression analysis.

**P9.L29**: '*... from the water surface. Net longwave radiation was calculated following the equations 2.24 and 2.28 from Moene and Van Dam (2014):*
$L_{in} = \varepsilon_a \sigma T_a^4$,
$L_{out} = L_{e,\,out} + (1 - \varepsilon_s) L_{in}$,
*in which $L_{in}$ [W m-2] is the incoming longwave radiation, $L_{out}$ [W m-2] is the outgoing longwave radiation, $\varepsilon_a$ [–] is the apparent emissivity that is a function of the fraction of cloud cover, $\sigma$ [= 5.67_10-8 W m-2 K-4] the Stefan-Boltzmann constant, $T_a$ [K] the air temperature, $L_{e,\,out}$ [W m-2] is the emitted longwave radiation, and $\varepsilon_s$ the surface emissivity. Net shortwave radiation was calculated as (Allen et al., 1998):*
$K^* = (1-\alpha) K_{in}$
*in which $K^*$ [W m-2] is the net shortwave radiation, $K_{in}$ [W m-2] the global radiation, and $\alpha$ [–] the albedo values for which monthly values were calculated as a function of latitude (Cogley, 1979).'*

Moene, A. F. and van Dam, J. C.: Transport in the Atmosphere-Vegetation-Soil Continuum, Cambridge University Press, Cambridge, 2014.
Allen, R. G., Pereira, L. S., Raes, D., and Smith, M.: Crop Evapotranspiration - Guidelines for Computing Crop Water Requirements - FAO Irrigation and Drainage Paper 56, United Nations-Food and Agricultural Organization, 1998.
Cogley, J. Graham. "The Albedo of Water as a Function of Latitude." *Monthly Weather Review* 107 (1979): 775-781.

- P6.L30: Could you please rephrase this sentence to improve readability.

We have changed the sentence into (**P7.L10**): '*Firstly, the raw data were quality-controlled using several criteria in order to remove faulty or corrupted data.*'

- P7.L30-32: "A regression analysis …" + "To develop …" Please rephrase to improve readability.

We have changed the sentence into (**P8.L17**): '*A regression analysis was performed to explore which variable, or combination of variables, can best explain the dynamics of $E_{water}$. Variables included in this analysis were wind speed, VPD, global radiation, vertical vapour pressure gradient, air temperature and water temperature. From the regression analysis a data-driven model was developed to estimate $E_{water}$ of Lake IJssel. This was done for both locations, Stavoren and Trintelhaven.*'

- P8.L11: I found difficult to understand the justification of using such regression model and how the hypothesis of such model have been tested.
What are the type of estimator you used (I assume an ordinary least square estimator)? Did you perform a significativity test? It would be interesting to look at the result of the multiple linear regression model and specifically the p_value to include or exclude predictors. Are the period chosen representative of the population?

The functional form of the regression models was chosen to be a simple combination (sum or product) of variables only considering linear regression, multiple linear regression and quadratic regression models, which was a data-driven decision rather than a process-based decision.
Yes, statistical significance was tested on the used models (p-values < 0.05; this has been added at **P9.L1** *'... a single variable. Statistical significance (p < 0.05) was tested. From the multitude...'*), meaning the best and simple models that we continued working with: Fig 6+7 and Table 2. The Venn diagrams however showed the model fit of all combinations, without an indication of significance. We have adjusted this to the Venn diagrams by removing those values where the model fit was found insignificant. We added to the caption of the Venn diagrams: *'Values were removed if the model fit was found to be insignificant (p < 0.05).'*
Since our aim is to study the surface-atmosphere coupling, we did not perform gap filling, resulting in less data points, but avoiding the use of 'artificial' data. That is the trade-off to be made and as a consequence the summer period chosen for calibration is all data that we have. The fact that the chosen models that were fit on this period are significant, provides us confidence, as well as the relatively good validation results.

- P8.L22: "surface temperature" Are you talking about the Meteosat product? Hence, why do not use directly these field data? Are there representative of the surface temperature (the lake is shallow then it would be important to be sure the measurements are not performed in the thermocline).

What we are trying to convey here is that there are no routine observations of the surface water temperature, or the skin temperature, measured with for instance thermal infrared cameras. Due to these lacking observations we were curious to find out if water temperature measured routinely at 1.2 to 1.5 m deep would suffice in estimating $E_{water}$. To clarify this we have replaced the text at **P9.L11** by: *'There are no routine observations available of the skin water temperature of the lake. As an alternative, the use of water temperature data routinely measured by Rijkswaterstaat at depths ranging from 1.2 to 1.5 m was explored.'*

**Results**

General comments: In this section, I would have a distinct paragraph presenting the results of the calibration, another for the validation and a final one for the result on the routinely measured variables. It would also improve the readability as I had hard time following this section.

Thank you for this feedback. In response to this valuable suggestion, we have improved readability by making this distinction in presenting the results of 3.4.1) Calibration, 3.4.2) Validation, and 3.4.3) Models based on routinely measured variables, as subsections of Section 3.4.

- P11.L5; Fig2: How did you choose the time period presented in the figure? Why do you present this period instead of either the training or validation time period? As I understand your paper is about summer and here you present a part of the autumn season. I admit I was a bit lost. Also I would

suggest to be more specific on the time period (e.g: 01/05-31/08 instead of nouns (You write May-August most of the time and once May-September).

*The idea behind this figure, as well as figure 3, was to explore how the dynamics and trends of the meteorological variables, and especially the heat fluxes, would evolve before, during and after the summer, to explore if any lags for instance would occur. That is the reason why we presented here the period 01/05 – 30/09.*

*To clarify this we have added the following to* **P11.L12**: *'This figure illustrates the dynamics and trends of the meteorological variables and the heat fluxes before, during and after the summer period to explore if any lags for instance would occur at this timescale.'*

*And at* **P12.L2** *'The monthly average diurnal variability of observed LE, based on hourly data, are shown in the top panels of figure 3 for location Stavoren for the same period as in figure 2 (i.e. 01/05/2019 – 30/09/2019).'*

- Section 3.2: In this section, you compare meteorological conditions. It seems you are comparing the air temperature and the wind speed that are are measured at different height. Measurements at Stavoren are made around 7m and around 10m at Trintelhaven. Did you adjust your measurement to an equivalent height? If not, this could explain some of the discrepancies. I have the same question for EC data. Moreover, you compared these variables to the Dalton model which needs variables at 2m height. What was the procedure you used to adjust the measurements to this height?

*You are right that we have not adjusted our measurements to an equivalent height. We have made a remark on this at* **P6.L12**: *'The measurement height at the two locations Stavoren and Trintelhaven differ. In our analysis we have not adjusted the measurements to an equivalent height. In theory, the small height difference will not affect the heat fluxes under the assumption of a constant turbulent flux layer.'*
*Furthermore, the difference in wind speed found between the two locations cannot be explained by the difference in measuring height, as the resulting average wind speed actually is higher in Stavoren which is measured closer to the surface. The regression coefficients found might differ slightly because of the difference in measurement height. However, we do think that the variables that were found to be most important to explain the variance of* $E_{water}$ *(i.e. wind speed and vapour pressure gradient) will not change.*
*Considering the remark about comparison with the Dalton model – there are two notes on this: 1) in the caption of figure 3 ('Note that some variables included in the evaporation models are measured at larger heights than the 2 m that are prescribed (see Eq. 6 – 9)', and 2) in the discussion at* **P21.L9**.
*This should be taken into account. However, we do think using variables measured at greater heights will not change the diurnal trends that were found.*

- P10.L20: "the water temperature at" instead of "the water at".

*Correct, has been adjusted.*

- P10.L18: Please provide the correlation score to justify this is a strong correlation.

*We will indicate the correlation score here, which is* $R^2=0.61$. *But, moreover, what we would like to indicate here is that from looking at time series only, one could see that latent heat flux and wind*

speed are showing similarity in their trend, suggesting a good correlation between the two variables. This gives an indication for further analysis. Therefore, we have changed the sentence into **(P11.L29)**: '*The latent heat flux displays similar trends as the measured wind speed, indicating that the two variables are correlated ($R^2$ = 0.61).*'

- P11. Fig2: Please put the graph corresponding to the 2020 summer period in Appendix.

Agreed and done.

- P12.L1: I would use the word "pattern" instead of "rhythm".

Agreed and changed.

- P12.L5: Could you please add the graph in the Appendix?

This has been added (**App C**).

- P13.L17: Could you please add the correlation score to justify if it's a strong correlation. Also, be careful not to mismatch between correlation and determination when you analyse your results and even more as you are studying a non-linear model. Please rephrase the sentence "Global radiation …" to account for this difference.

The square of the correlation coefficient between wind speed and latent heat flux following from figure 2 is $R^2$=0.61. To match this with previous wording we will change it into '… *with the adequate correlation ($R^2$=0.61) visible in figure 2.*'. Furthermore, we have adjusted **P15.L1** to '*Global radiation has the lowest adjusted coefficient of determination, which agrees with our findings in figures 2 and 3.*'

- P14.L4-6: I understand that you only exclude the global radiation from your model based on the $R^2$, however, in my opinion a $R^2$(VPD)=0.05 questioned the inclusion of VPD in your model. For example, adding VPD on the Stavoren hourly analysis has a limited impact which is not significant in Trintelhaven. Does VPD has a significant impact on your score?

We have checked the addition of VPD to the model again, and indeed it appeared to be significant. However, due to a mistake that we found as a result of one of the feedback points from anonymous reviewer #1, the Venn diagrams changed: only slightly for the hourly diagrams, but quite significantly for the daily diagrams. As a result, the best regression model is the same as the simple regression model: $LE_{mod}$ = 5.3 $u\Delta e$ + 1.6 $(u\Delta e)^2$ + 41.7 ($R^2$ = 0.74). The figures and accompanying text have been changed accordingly.

- P14.L22: Would it be possible to have some basics statistics on these data (mean, standard deviation, quantiles, min, max). It would also help to see if outliers are ejected from the analysis. A Table placed in the Appendix would be sufficient and would give a hint about the discrepancies between both summer seasons.

We have added the following table to the Appendix:

| (a) Stavoren – LE [W m$^{-2}$] | Mean | Stdev | min | max | Q25 | Q75 | N |
|---|---|---|---|---|---|---|---|
| Hourly | | | | | | | |
| 2019 | 104.1 | 69.2 | -21.5 | 516.1 | 56.6 | 133.1 | 896 |
| 2020 | 128.4 | 87.6 | -27.7 | 444.6 | 66.1 | 146.5 | 687 |
| Daily | | | | | | | |
| 2019 | 130.7 | 67.3 | 51.2 | 310.1 | 84.3 | 147.5 | 20 |
| 2020 | 168.4 | 85.3 | 55.7 | 364.5 | 111.3 | 215.0 | 18 |

| (b) Trintelhaven – LE [W m$^{-2}$] | Mean | Stdev | min | max | Q25 | Q75 | N |
|---|---|---|---|---|---|---|---|
| Hourly | | | | | | | |
| 2019 | 95.9 | 59.2 | 11.0 | 333.9 | 47.4 | 130.3 | 454 |
| 2020 | 91.6 | 60.2 | -34.8 | 351.8 | 52.1 | 113.2 | 663 |
| Daily | | | | | | | |
| 2019 | 122.3 | 45.3 | 58.9 | 210.9 | 94.8 | 147.3 | 10 |
| 2020 | 100.5 | 60.1 | 35.1 | 273.8 | 81.9 | 105.4 | 13 |

- P14.L15: $R^2$ explains 45% of the variance which is quite low. If you include the water temperature, it reaches 0.48, it is still low but better. You limit the maximal number of variables for the simple model but in this case it would benefit to your model to add the water temperature.

Due to solving the mistake that we mentioned earlier the numbers have changed as well as the combination of variables leading to the best and the simple model. The Venn diagram of Trintelhaven where the sum of variables is given for daily timescales (Fig. 5d) now gives the highest adjusted $R^2$ for the combination of wind speed and Δe. The figures and accompanying text have been adjusted accordingly.  Additionally, it is simply a choice to explore both 1) the best model which can include as many variables as possible, and 2) to limit the simple model to two variables at the most, as explained in **P9.L3**.

- P14.L22: "This can be attributed …" As noticed in Woolway et al 2021, lake evaporation is highly dependent on weather variability (through its dependence to the lake surface temperature). Your discussion need to stress this issue and not just focus on the comparison to a mean climate. I would remove this sentence and discuss about this point in the adequate section.

Thank you for this suggestion. We have added some words on this topic to the discussion section for completeness. However, we think that the dependence of lake evaporation to synoptic weather variations through the dependence with lake surface temperature is actually included in the regression analysis, where $T_{water}$ and vapour pressure gradient (which combines wind speed and humidity, both also related to synoptic weather variations) are two of the variables that were included.

We have adjusted the following to **P19.L11**: '*This is similar to what has been found by studies of for instance Blanken et al. (2011) and McGloin et al. (2014), and it was noticed that intraseasonal variations of $E_{water}$ can be linked to synoptic weather variations through these variables (Lenters et al., 2005, MacIntyre et al. 2009, Liu et al., 2011, Woolway et al. 2020). The same ingredients of wind speed and vapour pressure gradient were used in the model by Dalton (1802).*

Lenters, J. D., Kratz, T. K., and Bowser, C. J.: Effects of Climate Variability on Lake Evaporation: Results from a Long-Term Energy Budget Study of Sparkling Lake, Northern Wisconsin (USA), Journal of Hydrology, 308, 168–195, https://doi.org/10.1016/j.jhydrol.2004.10.028, (2005)

MacIntyre, S., Fram, J.P., Kushner, P.J., Bettez, N.D., O'Brien, W.J., Hobbie, J.E. & Kling, G.W. Climate-related variations in mixing dynamics in an Alaskan arctic lake. Limnol. Oceanogr. 54(6, part 2), 2401-2417 (2009)

Liu, H., Blanken, P.D., Weidinger, T., Nordbo, A. & Vesala, T. Variability in cold front activities modulating cool-season evaporation from a southern inland water in the USA. Environ. Res. Lett. 6(024022) (2011)

Woolway, R. I., Kraemer, B. M., Lenters, J. D., Merchant, C. J., O'Reilly, C. M., & Sharma, S. (2020). Global lake response to climate change. Nature Reviews Earth & Environment, 1(8), 388-403.

- P14.L26: "this confirms ..." Rephrase this sentence. If the ingredients are the same than in the Dalton's model why do not use this model or use a calibrated version on your lake?

The aim of our study is to find the drivers of $E_{water}$ based on observations, without predetermining the variables to be included. The observations were used to develop the regression models, which confirmed a very similar relation to what was found by Dalton. We think this only helps to gain confidence to the fact that indeed wind speed and $\Delta e$ are the most important drivers of $E_{water}$. We have replaced **P17.L7** with the following text: '*Without predetermination of the variables, we found the same ingredients as used in the Dalton model as the most important drivers of $E_{water}$ at hourly and daily timescales.*'

- P14.L28: "To determine if the coefficients ..." Without the results of the analysis it is impossible to assess the results. Either give the results or erase this sentence.

The result from the ANOVA analysis showed that with a p-value of 0.02, that the inclusion of the station (i.e. Stavoren or Trintelhaven) matters. So we cannot use the same model coefficients for both locations, and in other words: we cannot rule out that the sites are different (**P17.L11**). We have added the ANOVA table as Appendix (**App E**), and adjusted **P17.L10**: '*..., an ANOVA statistical analysis was performed (see Appendix ...). This analysis shows that inclusion of the station matters (p < 0.05). Therefore, we cannot rule out that the sites are different.*'

```
> anova(fit0, fit1)
Analysis of Variance Table

Model 1: LE ~ u_De + I(u_De^2)
Model 2: LE ~ u_De * station + I(u_De^2) * station
  Res.Df    RSS Df Sum of Sq      F  Pr(>F)
1    559 664256
2    556 652804  3     11452 3.2513 0.02151 *
* * *
Signif. codes:  0 '***' 0.001 '**' 0.01 '*' 0.05 '.' 0.1 ' ' 1
```

- P16.L5-9: "The results for the location of ..." This is a good analysis of your results. However be sure to be consistent. In your abstract you say that the model performs well.

Agreed. We checked for consistency and adjusted the abstract **(P1.L10)**: '*Validation of the data-driven models demonstrates that a simple model using only two variables yields satisfactory results at Stavoren, with $R^2$ values of 0.84 and 0.78 for hourly and daily data, respectively. However, the validation results for location Trintelhaven fall short ($R^2$ values of 0.67 and 0.65 for hourly and daily data, respectively). Using only routinely measured meteorological variables leads to adequate performing simple models at hourly ($R^2 = 0.78$ at Stavoren, and $R^2 = 0.51$ at Trintelhaven) and daily ($R^2 = 0.82$ at Stavoren, and $R^2 = 0.87$ at Trintelhaven) timescales.*'

**Discussions**

You need to improve your discussions and criticise your result in a more precise way. You could be more exhaustive and include limitations (e.g decomposition of the radiation term, looking at the influence of other hydroclimate variables on the variance).

- P17.L7: You should be more precise and discuss the fact that you do not analyse each term of the radiation budget.
Agreed. We removed the brackets around <<global>> at **P18.L10**, and we have added the following to **P19.L1**: '... figure 3). '*Note that the relation between $E_{water}$ and other components of the radiation budget could not be studied, because of the lack of observations of these components. In combination with absent data on G, this prevented us from fully capturing the role of net radiation in the energy balance of the lake, and thus with the warming and cooling of the lake, which relates to evaporation through the water surface temperature.*'

- P18.L6: you can add Le Moigne et al 2016 as a reference.

Thank you for this suggestion. As far as we can see the reference mostly focusses on the impact of lakes on climate using the Flake lake scheme. It does not directly support the statement we make here about the combination of wind speed and vapour pressure gradient to estimate $E_{water}$. However, we have included this interesting reference to the introduction at **P1.L21**.

*Patrick Le Moigne, Jeanne Colin & Bertrand Decharme (2016) Impact of lake surface temperatures simulated by the FLake scheme in the CNRM-CM5 climate model, Tellus A: Dynamic Meteorology and Oceanography, 68:1, DOI: 10.3402/tellusa.v68.31274*

- P19.L10-13: It is impossible to review this part of the discussion as you don't provide the results. You should either erase the sentence or give the correlation plots.

Agreed. After consideration, we decided to erase these sentences as it does not contribute substantially to our storyline to add more graphs.

- As mentioned in the precedent comment (for P14.L22), you are working at short timescale and thus, the lake evaporation is dependent on the weather and the hydrological variability. Your discussion would be more complete by discussing these points.

We agree that this point should be noticed in the discussion section. In the precedent comment about P14.L22 we have added the suggested change to the text.

**Conclusions**

- P21.L19: In my opinion, your main contribution is the development of the statistical model. I would suggest to rephrase your conclusion in order to account for this.

Indeed the development of the statistical model forms a significant part of this study. However, the statistical model is used as a tool to explore the drivers of open water evaporation. We have made the following adjustment at **P24.L2**: '*In this study, we investigated the dynamics and drivers of open water evaporation of Lake IJssel in the Netherlands through the development of a data-driven model.*'

- P21.L20: Ok but this a general fact and this is not your main result.

We stated it there to provide a context for readers, but in order to kick-off with our main conclusions we have removed the following sentences: P21.L20 'We have shown....' till P21.L23 '....of open water evaporation'.

- P21.L26-28: Rephrase the sentences to be more precise on the result you use (if it's hourly or daily timescale). It is hard time following which are the R² you are presenting.

We agree with that and we are now more specific in our referencing to the timescales: **P24.L7**: '*Using the data collected in 2019 regression models for both location were developed. At the hourly timescale this resulted in $R^2 = 0.74$ and $R^2 = 0.70$ for Stavoren and Trintelhaven, respectively. Validation of these hourly models using the data collected during the summer of 2020 have shown that a simple data-driven model is able to explain large part of the hourly dynamics of open water evaporation ($R^2 = 0.84$ and $R^2 = 0.67$ for Stavoren and Trintelhaven, respectively).*'.

**Editorial comments**

- Some sentences lack of consistency and readability. This is often the missing punctuation that is in cause. For example, look at P2.L12, P7.L17, P7.L24, P9.L24, P12.L3.
Corrected.

- Be attentive to have consistency in the form you write the units. The general form is to separate units with a point, ex : m.s-1 .
HESS prescribes to have the units written with a space and exponent, e.g. W m$^{-2}$.  We have checked for consistency throughout the manuscript.

- « Focussing » should be written like this « focusing ».
Noted.

- P2.L8 : parametrise/parametrize and not « parameterise ».
Parameterize is widely used and also accepted by HESS. We have changed it to *parameterize*.

- P2.L26 : add a « to » → « and to represent ».
Done.

- P2.L33 and P7.L20: check the tense.
P2.L33: was changed into: 'entered'
P7.L20: was changed this into: 'we used'

- P4.L25 : use the English structure : 1,100 km²
Agreed.

-P5. Figure1 : Increase police size of your scale as it is not readable. Moreover, you should add the label on the contour lines. On the right figure you can erase the y-axis as it is the same than the center figure. Green and white colors for the center and right panels are not adapted to understand where are the land and the water. Add labels to the windrose.

Thank you for this feedback. We have adjusted this figure according to your feedback. However, adding labels to the contour lines of the footprint analysis was not possible in terms of readability. But we do think that the text (P.L) and caption will help the reader to understand the contour lines.

- P7.L8 : you should either use co-variance or covariance but not both throughout your paper.

Noted.

- P8.L9 : « variable(s) » instead of « variable(x) ».

Thank you for spotting this. Adjusted.

- P10 Table 1 : Be sure all the parameters are aligned in the first column.

We placed the indent there as $T_{air}$, $T_{air,climatology}$, and $T_{water}$ are all belonging to the variable T.

- P12.L11: "are lacking"

Noted.

- Please consider improving your Venn diagram in order to gain in readability (police size).

The arrangement of all the intersections of the Venn diagram leaves less space for increasing the font size. We have done our best to gain readability.

**Specific comments by anonymous referee #3**

• There are many studies showing that evaporation dynamics vary between different parts of inland water bodies arising from, for example, spatial variability of water depth or meteorological inputs (boundary conditions). Considering this, it is fine to have two different measurement stations at the middle of the water body (Trintelhaven) and at the border (Stavoren). However, a discussion on the effect of EC towers location on the observed evaporation dynamics from the water body is missing (not its comparison with terrestrial evaporation as mentioned in section 2.2).

This is a good point that is raised. We have added the following to the discussion section of the manuscript (**P23.L11**): '*In other studies, the dynamics of $E_{water}$ have been found to spatially vary over an inland water body caused by advection and fetch distance from the upwind shore (Weisman and Brutsaert, 1973). More specifically, Granger and Hedstrom (2011) have found that $E_{water}$ is a function of the lake-land contrast of temperature and vapour pressure. Another source for spatially varying $E_{water}$ is the water surface temperature, which can be affected by the spatial variability of water depth (Wang et al., 2014), or for instance by the supply of water of a different temperature through rivers. Given that our measurements sites are located (i) at the shore in the north of the lake (Stavoren), and (ii) on the dike in de middle of the lake (Trintelhaven), could therefore potentially lead to differences in observed $E_{water}$ dynamics between the two measurement sites. The coefficients of the hourly regression models were found to be significantly different between the two locations (Sect. 3.4.1). This difference might be attributed to the difference in location (i.e. at the shore and in the middle of*

*the lake, respectively). Other reasons might be the difference in measurement height or the inherently different meteorological conditions we measure, because the two measurement sites are located on opposite sides of the lake.'*

• Vapor pressure at the air-water interface was estimated based on the surface temperature obtained from satellite imagery that often show biases. Was this checked?

Specifications of the Meteosat sub-skin water temperature product describe a bias of 0.5 Kelvin, where validation takes place with temperature measurements from drifting buoys. We argue that this small bias will not change our findings of the most important drivers (i.e. wind speed and vapour pressure gradient) of $E_{water}$. If at all, the bias in the satellite product might lead to slightly different observed dynamics in water temperature and thus vapour pressure, which potentially could lead to changes in the regression coefficients. However, we think this effect will be insignificant.

**Additional comment by editor:**

Your response to Reviewer #3 regarding bias in remotely sensed sub-skin water temperature: Would it not be relatively easy to propagate the reported uncertainty through your analysis to confirm your assertion that the effect is not significant?

We believe that propagating the reported uncertainty of the satellite product would not help to learn more about which variable(s) are driving $E_{water}$. Propagating the bias would not change the correlation that is found and propagation of the uncertainty (standard deviation is reported to be 1°C) would just add noise and as a result reduce the correlation. We therefore argue that reporting the product specifications will be sufficient, taking into account the aim of the study.
*'Product specification describes a target accuracy with a bias of 0.5°C and a standard deviation of 1.0°C.'* **(P6.L20)**

• From our own measurements of vertical water temperature in a shallow basin with 2 m depth, I can say water temperature at the surface, where evaporation takes place, is completely different with even 10 cm below. Thus I am not sure how water temperature at depth of 1.2 or 1.5 m could help for evaporation analysis, unless you have a temperature model to reproduce surface temperature.

**Additional comment by editor:**

Could you also mention the strong temperature gradients near the surface mentioned by Reviewer #3 to alert the reader to the fact that the correlation found here may be an exceptional case?

We agree that indeed preferably the skin temperature of the water surface is used in the estimation of evaporation rates. That is why we have used the satellite product in the first part of our regression analysis. Additional to that, we were interested if other routinely observations in the Netherlands which are easily accessible would provide another reasonable source from which evaporation rates could be estimated. The good correlation ($R^2$ = 0.71 and $R^2$ = 0.94; based on summer 2019 at Stavoren and Trintelhaven, respectively) between sub-skin water temperature from the satellite product and water temperature measured at depth of 1.2 or 1.5 m gave rise to that.

Concerning the strong temperature gradients near the water surface we have added **(P6.L25)**:
*' Although typically a strong vertical water temperature gradient exists near the water surface, a good correlation (R2 = 0.71 and R2 = 0.94, for the summer period of 2019 at Stavoren and Trintelhaven, respectively)*

*was found between the sub-skin water temperature from the satellite product and the water temperature measured at larger depths.'*

• It is not surprising that neglecting thermal inertia of the water body (indicating the effect of radiation adsorption in depth) could make such considerable bias in the performance of models such as Penman's (see Friedrich et al. 2018: DOI:10.1175/BAMS-D-15-00224.1). Please see Section 2.4 of Zhao et al. 2020 (https://doi.org/10.1016/j.rse.2020.112104) accounting for the impact of G on evaporation estimates by Penman-type approaches.

As a response to the role of G on evaporation, we would like to refer to our response to specific comment 3 of referee #1.

• Why air temperature is not included in the analysis of section 3.4?

We did not include air temperature as a separate variable here because we chose to include its effect in the vertical vapour pressure gradient, where air temperature is integrated in.

We have added the following to **P8.L21**: '*Air temperature was not included as a separate variable because it was chosen to include its effect through the vertical gradient of vapour pressure.'*

• I believe radiation is a key component in shaping surface temperature that, in turn, defines vapor pressure gradient at the core of your simple regression model. In application of Eq. 10, how surface temperature is obtained? From measurements at depth of 1.5 m, satellite imagery, or solving for energy equation? Dalton-type models may look simple in representation but have difficulties associated with obtaining reliable surface temperatures, and of course wind function (especially in the context of the projected climate change).

We agree that radiation is shaping surface temperature, and thus affects the vapour pressure gradient. In turn, evaporation decreases surface temperature and affects vapour pressure gradient as well. They are interdependent. To answer to your question about equation 10: We did not include the Dalton model in our analysis in the form presented as equation 10. We have placed it here in the discussion section to make a comparison between the functional form of the Dalton model and the regression model that we have found. To be complete, in figure 3 where we did present the monthly average diurnal cycle of, amongst others, the Dalton model. There, we have used the temperature data originating from the satellite product to calculate the vapour pressure gradient. We have added some words about the difficulties associated with the Dalton model to section 2.5 (**P9.L22**). '*Although the representation of the Dalton models may seem simple, obtaining reliable measurements of surface temperatures are challenging.'*

---

## Author Response (AR2)

**Response to feedback from Anonymous Referee #1, #3, and the editor**

We would like to thank and express our appreciation to the positive feedback from the reviewers and the editor to our revised manuscript that we have submitted March 30, 2022. Below, we would like to respond to the remaining point of suggestion for revision raised by anonymous referee #3 (comment of the referee in **black**, our response in **blue**). It includes references in **green** to where changes have been made in the newly revised manuscript.

**Comment by anonymous referee #3**
The authors have mostly addressed my comments and questions; however, I am not convinced with the answer to my question about the lack of air temperature in the analysis of section 3.4.
The authors think that the effect of air temperature is included in the vertical vapor pressure gradient. This is correct but air temperature is a key parameter directly affecting sensible heat exchanges of the surface with overlying air and thus surface temperature and energy balance of the water body. I would not thus limit the role of air temperature to VPD and vertical gradients.

**Additional comment by editor:**
Referee #3 pointed out one remaining issue about not including air temperature in the analysis and I agree with him that your explanation is not clear enough. It would be good to clarify to the reader that the fundamental driver for evaporation is considered to be the water-air vapour pressure gradient, whereas air temperature affects the water temperature through sensible heat flux, and hence affects the evaporation indirectly through its effect on the vapour pressure gradient. Since VPD is a function of vapour pressure and air temperature, the reader may wonder why it was included as a separate variable in addition to the vapour pressure gradient (P8L19), while air temperature was not.

In this study we have not included air temperature as a separate variable in the regression analysis. Air temperature affects the surface temperature through the sensible heat flux. In turn, the surface temperature affects the vapour pressure gradient and thus evaporation. For a land surface we expect that air temperature indeed affects surface temperature in a direct way through the sensible heat flux. However, due to the large thermal buffer we expect that for a water body there is a less direct coupling between sensible heat flux and latent heat fluxes at short timescales, meaning that there is little direct influence of air temperature on open water evaporation. For instance, if you would have very dry air that has the exact same temperature as the water temperature evaporation would take place, but the sensible heat flux would be zero. Additionally, variables that are included in the regression analysis should ideally be as different as possible. In this case, air temperature correlates to an extent with both the surface temperature and VPD. Therefore we chose to not include air temperature additional to the water temperature and VPD.

We have changed/added the following texts in the manuscript:

**P8.L21:** *"Air temperature was not included as a separate variable in the regression analysis. Air temperature affects the surface temperature through the sensible heat flux. In turn, the surface temperature affects the vapour pressure gradient and thus evaporation. For a land surface we expect that air temperature indeed affects surface temperature in a direct way through the sensible heat flux. However, due to the large thermal buffer we expect that for a water body there is a less direct coupling between the sensible heat flux and latent heat flux at short timescales. Additionally, variables that are included in the regression analysis should ideally be as different as possible. Here, air temperature correlates to an extent with both the surface temperature and VPD. Therefore we chose to not include air temperature additional to the water temperature and VPD."*

**P14.L9:** *"As explained in Section 2.5 air temperature was not included in the regression analysis because we expect that the large thermal buffer of a water body results in a less direct coupling between the sensible heat flux and the latent heat flux at short timescales."*

---

## Author Response (AR3)

**Response to feedback from the editor**

Thank you for your quick response and your suggestions for improving our argumentation. We see the point that you bring up here and agree with you that to base our argumentation on an assumption (i.e. the lack of response to Ta as a result of the thermal buffer) is not very objective and therefore less strong. Therefore, we agree to strengthen our motivation based on your suggestion where we argue that using VPD and vertical vapour pressure gradient, rather than using Ta, is based on the explicit use of these variables by Dalton and Penman in their equations.

We therefore have adjusted the following text:

**P8.L21:** "*To be specific, the choice of including VPD and vertical vapour pressure gradient in the regression analysis was motivated by the apparent drivers of the Dalton and Penman equations. It was decided to give preference to the use of VPD over air temperature as dependent variable in the regression analysis due to its explicit mention in the Penman equation, whereas air temperature only features implicitly in the definition of the slope of the vapour pressure gradient (s), and VPD.*"

**P14.L1:** "*It should be reminded that air temperature was not explicitly included in the regression analysis as explained in Section 2.5.*"

---

## Author Response (AR4)

**Response to feedback from the editor**

Thank you for your quick response again. We indeed would like to elaborate a bit more on what the consequence could have been by including air temperature explicitly in the regression analysis.

We therefore have adjusted the following text:

**P14.L1:** "*It should be reminded that air temperature was not explicitly included in the regression analysis as explained in Section 2.5. We expect that including air temperature as a separate dependent variable might have explained a part of the evaporation dynamics, since air temperature affects surface temperature through the sensible heat flux. In turn, the surface temperature affects the vapour pressure gradient and thus evaporation. However, due to the large thermal buffer of a water body we expect that there is a less direct coupling between the sensible heat flux and the latent heat flux at short timescales.*"